# EVENT-T2M: EVENT-LEVEL CONDITIONING FOR COMPLEX TEXT-TO-MOTION SYNTHESIS

**Seong-Eun Hong**[1]*, **JaeYoung Seon**[2]*, **JuYeong Hwang**[1], **JongHwan Shin**[1], **HyeongYeop Kang**[1]†

[1]Department of Computer Science and Engineering, Korea University
[2]Department of Artificial Intelligence, Kyung Hee University
{seong_eun,05judy02,citi1006,siamiz_hkang}@korea.ac.kr[1], cogongnam@khu.ac.kr[2]

## ABSTRACT

Text-to-motion generation has advanced with diffusion models, yet existing systems often collapse complex multi-action prompts into a single embedding, leading to omissions, reordering, or unnatural transitions. In this work, we shift perspective by introducing a principled definition of an *event* as the smallest semantically self-contained action or state change in a text prompt that can be temporally aligned with a motion segment. Building on this definition, we propose Event-T2M, a diffusion-based framework that decomposes prompts into events, encodes each with a motion-aware retrieval model, and integrates them through event-based cross-attention in Conformer blocks. Existing benchmarks mix simple and multi-event prompts, making it unclear whether models that succeed on single actions generalize to multi-action cases. To address this, we construct HumanML3D-E, the first benchmark stratified by event count. Experiments on HumanML3D, KIT-ML, and HumanML3D-E show that Event-T2M matches state-of-the-art baselines on standard tests while outperforming them as event complexity increases. Human studies validate the plausibility of our event definition, the reliability of HumanML3D-E, and the superiority of Event-T2M in generating multi-event motions that preserve order and naturalness close to ground-truth. These results establish event-level conditioning as a generalizable principle for advancing text-to-motion generation beyond single-action prompts. Code and data are available at https://tjswodud.github.io/EventT2M.

## 1 INTRODUCTION

Text-to-motion generation has recently achieved striking numerical results on benchmarks such as HumanML3D (Guo et al., 2022) and KIT-ML (Plappert et al., 2016), with state-of-the-art models pushing *Fréchet Inception Distance* (FID) to the second decimal place. However, these numbers obscure a critical limitation rooted in the benchmarks themselves. HumanML3D, for example, primarily consists of simple, easy-to-generate motions; as a result, most research has focused on refining performance on trivial motions rather than tackling the challenge of hard-to-generate. In essence, the field has become adept at making simple motions slightly better while ignoring the complex, temporally ordered behaviors where text-to-motion could truly matter. Consequently, when a description such as "run forward, then stop, then wave" is given, leading systems frequently merge, skip, or reorder actions. This misalignment between benchmark success and the demands of structured real-world motion remains a key obstacle to deploying text-to-motion techniques in practical applications such as animation pipelines (Kappel et al., 2021), video production (Majoe et al., 2009; Yeasin et al., 2004), and embodied agents (Yoshida et al., 2025).

To move beyond this impasse, it is necessary first to characterize compositional complexity rather than treat all prompts as equally difficult. Existing datasets and evaluation protocols do not distinguish between simple single-action descriptions and complex multi-action sequences, making it impossible to assess whether improvements on low-complexity motions carry over to scenarios requiring higher temporal and structural complexity.

---

*These authors contributed equally to this work.
†Corresponding author.

In this work, we make three main contributions. (1) We reframe text-to-motion generation around the notion of an *event*, introducing a principled definition of an *event* as the smallest semantically self-contained action or state change described in a text prompt whose execution can be temporally isolated and mapped to a contiguous motion segment. (2) Building on this definition, we propose Event-T2M. This diffusion-based model that injects *event* tokens through a novel event-based cross-attention module (ECA), enabling the generation of complex multi-action sequences and achieving state-of-the-art performance on HumanML3D and KIT-ML. (3) To rigorously assess whether gains on simple motions generalize to compositional cases, we construct and release *HumanML3D-E*, the first benchmark that systematically stratifies text-to-motion prompts by *event* count, thereby introducing a reproducible evaluation protocol for event-level complexity and demonstrating the advantages of Event-T2M under increasing compositional demands.

## 2   RELATED WORKS

### 2.1   COMPLEX TEXT-TO-MOTION SYNTHESIS

Text-to-motion generation has made remarkable progress since the introduction of the HumanML3D (Guo et al., 2022). Research has mainly diverged into two directions: Vector Quantized-Variational AutoEncoder (VQ-VAE) (Van Den Oord et al., 2017)-based models and diffusion (Ho et al., 2020)-based models, pioneered respectively by T2M-GPT (Zhang et al., 2023a) and MotionDiffuse (Zhang et al., 2024a). VQ-VAE work has primarily focused on reducing quantization loss (Guo et al., 2024). Meanwhile, diffusion-based approaches have concentrated on improving model performance while simultaneously reducing the inference time of the diffusion process (Chen et al., 2023; Zeng et al., 2025). Across both, the main objective has been higher scores on HumanML3D, with many works achieving state-of-the-art results.

Beyond these benchmark-driven gains, some diffusion-based studies explicitly target more complex behaviors. GraphMotion (Jin et al., 2023) enriches text with semantic graphs to encourage compositional generation, though its evaluation is limited. MotionMamba (Zhang et al., 2024b) defines "complex" motions merely by filtering longer HumanML3D sequences, offering limited insight into true compositionality.

### 2.2   TOKEN-LEVEL CONDITION FOR FINE-GRAINED ALIGNMENT

Recent advances in text-to-motion increasingly adopt token-level cross-attention for fine-grained alignment between text and motion. A representative example is AttT2M (Zhong et al., 2023), which combines body-part attention and global-local motion-text attention. Motion is first encoded into a discrete latent space using a VQ-VAE whose encoder preserves body-part structure, ensuring token interactions reflect part-level dependencies. During generation, local cross-attention links motion tokens with individual words, while global attention via sentence embeddings provides holistic guidance. This design improves interpretability and motion quality, yielding strong results on HumanML3D (Guo et al., 2022) and KIT-ML (Plappert et al., 2016).

MMM (Pinyoanuntapong et al., 2024) extends this line with masked motion modeling, reconstructing motion tokens from masked segments conditioned on text. By jointly encoding text and motion in a single transformer, MMM enables bidirectional attention, reinforcing token-level alignment. Together, AttT2M and MMM demonstrate the benefit of token-level conditioning for richer text–motion correspondences.

However, much of the literature still relies on CLIP (Radford et al., 2021), whose text encoder represents an entire prompt with a single global embedding (e.g., the [EOS] token) when performing image–text matching. This design obscures the temporal order of multi-step descriptions. For example, "run forward, then stop, then wave" may collapse into one undifferentiated vector, leading to merged or reordered actions. In addition, CLIP's pretraining on broad image–text corpora provides weak supervision for motion, overlooking temporal continuity and event transitions that are critical for compositional generation.

To address these issues, we leverage TMR (Text-to-Motion Retrieval) (Petrovich et al., 2023), trained explicitly for motion-language alignment, injecting domain expertise absent in CLIP. Furthermore, instead of collapsing the entire prompt into one token, we introduce event-level tokeniza-

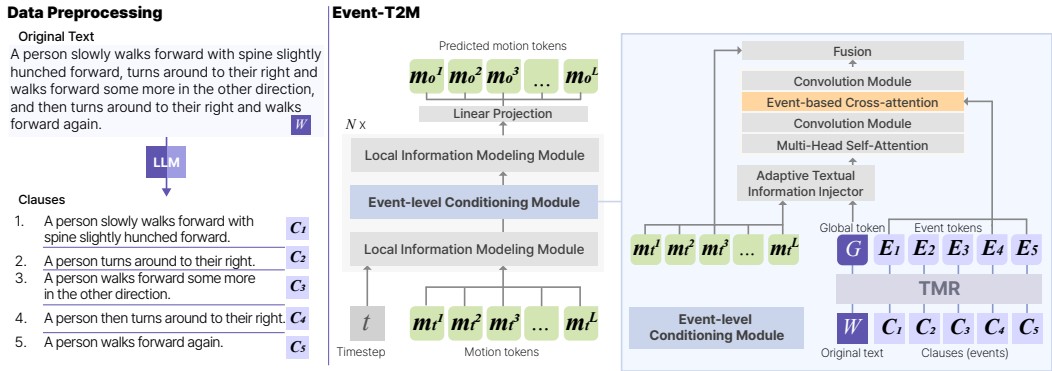

Figure 1: Main Architecture of Event-T2M. An input prompt is split into clauses by an LLM, encoded as event tokens with a TMR encoder, and fused with a global token. Tokens guide the diffusion process through an event-level module, enabling generation of sequentially complex motions.

tion: representative tokens are extracted per event, allowing feature matching that preserves temporal order and enhancing robustness to sequentially complex motions.

## 3 METHOD

We design Event-T2M, a diffusion-based text-to-motion generator tailored to handle complex sequential motions by explicitly modeling event structure. Our approach builds on three key ideas: (1) decomposing text into an *event* sequence using a Large Language Model (LLM), (2) embedding each *event* into an *event token* via a motion-specialized TMR encoder, and (3) injecting these *event tokens* through an event-based cross-attention (ECA) module inside Conformer (Gulati et al., 2020) blocks to capture both local and global sequencing.

### 3.1 TEXT TO EVENT TOKENS

We formalize an *event* as the smallest semantically self-contained action or state change described in a natural-language prompt, whose execution can be temporally isolated and mapped to a contiguous segment of the target motion. This definition is inspired by prior work on temporal action segmentation (Jin et al., 2023).

Formally, a text prompt $W$ is segmented into a sequence of clauses $\{C_k\}_{k=1}^{K}$ by an LLM, where $K$ denotes the number of clauses obtained under fixed linguistic rules. A clause $C_k$ is mapped to an event if it (1) expresses an action or state change by the same agent, (2) is semantically interpretable without requiring adjacent clauses, and (3) corresponds to a temporally coherent segment in motion space. This formulation yields a natural intermediate unit between words and full sentences: for example, "A person steps backward, jumps up, runs forward, then runs backward" is segmented into four events, each corresponding to one atomic action. We use Gemini 2.5 Flash (Comanici et al., 2025) to segment $W$. The used prompt is depicted in Appendix A.9.

To interface with motion models, we represent each event as an event token. Concretely, each clause (or event) $C_k$ is embedded using the TMR encoder, which we denote by $f_{\text{TMR}}$:

$$E_k = f_{\text{TMR}}(C_k), \qquad E_k \in \mathbb{R}^{D_y},$$

where $D_y$ is the embedding dimension of the TMR encoder. Stacking yields the *event tokens* used by cross-attention.

$$E = \begin{bmatrix} E_1^\top \\ \cdots \\ E_K^\top \end{bmatrix} \in \mathbb{R}^{K \times D_y}$$

To complement these event-level representations, we introduce a global text token $G = f_{\text{TMR}}(W)$ derived from the entire prompt $W$. This token serves as a holistic summary of the text, allowing

the model to fall back on global semantics when local event cues are ambiguous and to maintain coherence across long or compositional sequences.

## 3.2 ARCHITECTURE OVERVIEW

Overall architecture is shown in Figure 1. Given a textual prompt $W$ and its corresponding motion token sequence $M = \{m_i\}_{i=1}^{L}$, where each $m_i \in \mathbb{R}^{D_m}$ and $D_m$ is the pose feature dimension, we train a conditional denoiser $\varphi_\theta$ under a standard forward diffusion process with variance schedule $\{\beta_t\}_{t=1}^{T}$.

**Block overview.** We stack $N$ identical Event-T2M blocks. The input to the network is $M$. In the diffusion process, $M$ is represented as the clean motion $x_0$, and at each step $t$, we maintain a noisy motion $x_t$. Each block then updates it:

$$x_t \leftarrow x_t + \text{LIMM}(\text{concat}(x_t, t)), \tag{1}$$
$$x_t \leftarrow \text{ATII}(x_t, G), \tag{2}$$
$$x_t \leftarrow 0.5 * x_t + \text{FFN}(x_t). \tag{3}$$
$$x_t \leftarrow x_t + \text{ConformerSA}(x_t), \tag{4}$$
$$x_t \leftarrow x_t + \text{ECA}(x_t, E), \tag{5}$$
$$x_t \leftarrow x_t + \text{ConformerConv}(x_t), \tag{6}$$
$$x_t \leftarrow 0.5 * x_t + \text{FFN}(x_t), \tag{7}$$
$$x_t \leftarrow x_t + \text{LIMM}(x_t). \tag{8}$$

Text dropout implements classifier-free guidance (CFG) (Ho & Salimans, 2022) during training. Empirically, we observed that a 0.5 residual weight yields smoother optimization and improved stability under strong event-level supervision while keeping the feed-forward contribution balanced with the attention and event-conditioned branches, which aligns with the intuition from Macaron-style architectures (Lu et al., 2019) that split the feed-forward effect across two residual paths.

**Local Information Modeling Module (LIMM).** We implement the LIMM as a depthwise-pointwise 1D convolutional block, followed by GroupNorm and ReLU. This design enforces short-horizon smoothness with negligible parameter cost:

$$\text{LIMM}(x_t) = \text{ReLU}\big(\text{GN}\big(\text{PW}(\text{DW}(x_t))\big)\big), \tag{9}$$

where DW denotes depthwise convolution (kernel size 3), PW denotes pointwise convolution, and GN denotes GroupNorm. This reduces reliance on global attention for local kinematics while improving stability and contacts.

**Adaptive Textual Information Injector (ATII).** Unlike conventional cross-attention that mixes text and motion indiscriminately, ATII injects segment-aware semantics through channel-wise gating. Inspired by (Zeng et al., 2025), we first downsample the input motion sequence by a ratio of $S$ via a lightweight point-wise convolution layer, yielding $M' = \{m'_j\}_{j=1}^{L'}$. Then, the global text embedding $G$ is adaptively filtered by the local downsampled motion state $m'_j$:

$$\hat{g}_j = \text{Sigmoid}\big(W_c[m'_j \oplus G]\big) \odot G, \tag{10}$$

where $W_c$ is a fully connected projection, $\oplus$ denotes concatenation, $\odot$ is channel-wise product. The gated text feature $\hat{g}_i$ encodes segment-specific semantics, which are then fused with motion by another projection:

$$ATII(x_t, G)_j = W_f[m'_j \oplus \hat{g}_j]. \tag{11}$$

This adaptive injection mechanism provides stronger alignment between text and local motion, while avoiding excessive overhead compared to full cross-attention.

**Conformer for global and local sequencing.** ConformerSA($\cdot$) and ConformerConv($\cdot$) correspond to the self-attention and convolutional submodules of a Conformer-style architecture (Gulati et al., 2020). Specifically, ConformerSA($\cdot$) implements multi-head self-attention with relative positional

bias along time, allowing motion tokens to capture long-range temporal dependencies. In contrast, ConformerConv($\cdot$) applies a depthwise separable 1D convolution with Gated Linear Units (GLU) (Dauphin et al., 2017), modeling short-range and phase-local motion patterns. These submodules play complementary roles: self-attention integrates global context across the motion sequence, while convolution sharpens local dynamics such as step phases or contact transitions. We follow the standard Conformer design, where self-attention and convolution are placed between two feed-forward layers.

**Event-based Cross-attention (ECA).** To inject event-level semantics, we replace the standard self-attention sublayer in each Conformer block with a motion-to-text cross-attention mechanism. In this formulation, the motion tokens provide the *queries*, while the event tokens act as the *keys* and *values*.

Let $x_t^{\text{ctx}} \in \mathbb{R}^{L' \times D}$ be the current motion context, obtained from the ConformerSA sublayer. For $H$ heads of dimension $d_h$, we compute motion-to-text cross-attention by projecting motion tokens $x_t^{\text{ctx}}$ into queries and event tokens $E$ into keys and values:

$$Q_m = x_t^{\text{ctx}} W^Q \in \mathbb{R}^{(L') \times (Hd_h)}, \tag{12}$$

$$K_e = EW^K \in \mathbb{R}^{(K) \times (Hd_h)}, \tag{13}$$

$$V_e = EW^V \in \mathbb{R}^{(K) \times (Hd_h)}, \tag{14}$$

where splitting across heads gives $Q_m^{(h)}, K_e^{(h)}, V_e^{(h)} \in \mathbb{R}^{(\cdot) \times d_h}$. Multi-head cross-attention is then applied as

$$A^{(h)} = \text{softmax}\left( \frac{Q_m^{(h)} (K_e^{(h)})^\top}{\sqrt{d_h}} \right), \tag{15}$$

$$Z^{(h)} = A^{(h)} V_e^{(h)}, \tag{16}$$

with outputs concatenated as $Z = \text{Concat}_h Z^{(h)} W^O$. We then define the event-based cross-attention mapping as

$$ECA(x_t, E) = \gamma \cdot \text{Dropout}(Z), \tag{17}$$

where $\gamma$ is a learnable scaling factor initialized near zero for stable optimization.

### 3.3 DIFFUSION OBJECTIVE AND SAMPLING

We formulate motion generation as conditional denoising diffusion. At each $t$, a noisy motion sample is constructed as

$$x_t = \sqrt{\bar{\alpha}_t} x_0 + \sqrt{1 - \bar{\alpha}_t} \epsilon, \quad \epsilon \sim \mathcal{N}(0, I). \tag{18}$$

Then, the denoiser $\varphi_\theta$ is trained to recover $x_0$ from $x_t$ under event-level conditioning:

$$\mathcal{L}(\theta) = \mathbb{E}_{x_0, t, \epsilon} \left[ \| x_0 - \varphi_\theta(x_t, t, G, E) \|_2^2 \right]. \tag{19}$$

To enable CFG, text conditioning is randomly dropped with probability $\tau$, creating an unconditional path during training. At inference, we combine conditional and unconditional predictions through CFG, which sharpens motion-text alignment while preserving generative diversity. We adopt a 10-step Denoising Diffusion Probabilistic Models (DDPM) (Ho et al., 2020) for efficient generation.

## 4 EXPERIMENTS

We systematically evaluate Event-T2M to verify whether its event-level conditioning genuinely extends text-to-motion generation beyond single-action prompts. Our experiments combine standard quantitative benchmarks with newly constructed event-stratified test sets and complementary human studies. This design allows us to assess (1) competitiveness on existing benchmark settings, (2) robustness and compositional fidelity on multi-event prompts of increasing complexity, and (3) both the validity of our event-aware decomposition and the perceptual quality of generated motions from a user's perspective. Together, these evaluations offer a comprehensive view of Event-T2M's effectiveness and reveal how explicit event-level representations translate into measurable gains in realism, alignment, and human-perceived naturalness.

Table 1: Comparison on the HumanML3D, KIT-ML, and Motion-X test sets with existing state-of-the-art approaches. For each metric, "↑" denotes that larger values are better, while "↓" denotes that smaller values are better. The best score is marked in bold and the second-best is underlined.

| Datasets | Methods | R-Precision ↑ | | | FID ↓ | MM-Dist ↓ | MModality ↑ |
|---|---|---|---|---|---|---|---|
| | | Top-1 ↑ | Top-2 ↑ | Top-3 ↑ | | | |
| HumanML3D | T2M (Guo et al., 2022) | $0.455^{\pm.003}$ | $0.636^{\pm.003}$ | $0.736^{\pm.002}$ | $1.087^{\pm.021}$ | $3.347^{\pm.008}$ | $2.219^{\pm.074}$ |
| | MDM (Tevet et al., 2022) | $0.320^{\pm.005}$ | $0.498^{\pm.004}$ | $0.611^{\pm.007}$ | $0.544^{\pm.044}$ | $5.566^{\pm.027}$ | $\mathbf{2.799}^{\pm.072}$ |
| | MotionDiffuse (Zhang et al., 2024a) | $0.491^{\pm.001}$ | $0.681^{\pm.001}$ | $0.782^{\pm.001}$ | $0.630^{\pm.011}$ | $3.113^{\pm.001}$ | $1.553^{\pm.042}$ |
| | MLD (Chen et al., 2023) | $0.481^{\pm.003}$ | $0.673^{\pm.003}$ | $0.772^{\pm.002}$ | $0.473^{\pm.013}$ | $3.196^{\pm.010}$ | $2.413^{\pm.079}$ |
| | T2M-GPT (Zhang et al., 2023a) | $0.491^{\pm.003}$ | $0.680^{\pm.003}$ | $0.775^{\pm.002}$ | $0.116^{\pm.004}$ | $3.118^{\pm.011}$ | $1.856^{\pm.011}$ |
| | AttT2M (Zhong et al., 2023) | $0.499^{\pm.003}$ | $0.690^{\pm.002}$ | $0.786^{\pm.002}$ | $0.112^{\pm.006}$ | $3.038^{\pm.007}$ | $2.452^{\pm.051}$ |
| | FineMoGen (Zhang et al., 2023c) | $0.504^{\pm.003}$ | $0.690^{\pm.002}$ | $0.784^{\pm.002}$ | $0.151^{\pm.008}$ | $2.998^{\pm.008}$ | $2.696^{\pm.079}$ |
| | GraphMotion (Jin et al., 2023) | $0.504^{\pm.003}$ | $0.699^{\pm.002}$ | $0.785^{\pm.002}$ | $0.116^{\pm.007}$ | $3.070^{\pm.008}$ | $\underline{2.766}^{\pm.096}$ |
| | MMM (Pinyoanuntapong et al., 2024) | $0.515^{\pm.002}$ | $0.708^{\pm.002}$ | $0.804^{\pm.002}$ | $0.089^{\pm.005}$ | $2.926^{\pm.007}$ | $1.226^{\pm.035}$ |
| | MoMask (Guo et al., 2024) | $0.521^{\pm.002}$ | $0.713^{\pm.003}$ | $0.807^{\pm.002}$ | $0.045^{\pm.002}$ | $2.958^{\pm.008}$ | $1.241^{\pm.040}$ |
| | Light-T2M (Zeng et al., 2025) | $0.511^{\pm.003}$ | $0.699^{\pm.002}$ | $0.795^{\pm.002}$ | $\underline{0.040}^{\pm.002}$ | $3.002^{\pm.008}$ | $1.670^{\pm.061}$ |
| | MoGenTS (Yuan et al., 2024) | $\underline{0.529}^{\pm.003}$ | $\underline{0.719}^{\pm.002}$ | $\underline{0.812}^{\pm.002}$ | $\mathbf{0.033}^{\pm.001}$ | $2.867^{\pm.006}$ | - |
| | **Event-T2M (Ours)** | $\mathbf{0.562}^{\pm.002}$ | $\mathbf{0.754}^{\pm.003}$ | $\mathbf{0.842}^{\pm.002}$ | $0.056^{\pm.002}$ | $\mathbf{2.711}^{\pm.005}$ | $0.949^{\pm.026}$ |
| KIT-ML | T2M (Guo et al., 2022) | $0.361^{\pm.006}$ | $0.559^{\pm.007}$ | $0.681^{\pm.007}$ | $3.022^{\pm.107}$ | $3.488^{\pm.028}$ | $2.052^{\pm.107}$ |
| | MDM (Tevet et al., 2022) | - | - | $0.396^{\pm.004}$ | $0.497^{\pm.021}$ | $9.191^{\pm.022}$ | $1.907^{\pm.214}$ |
| | MotionDiffuse (Zhang et al., 2024a) | $0.417^{\pm.004}$ | $0.621^{\pm.004}$ | $0.739^{\pm.004}$ | $1.954^{\pm.062}$ | $2.958^{\pm.005}$ | $0.730^{\pm.013}$ |
| | MLD (Chen et al., 2023) | $0.390^{\pm.008}$ | $0.609^{\pm.008}$ | $0.734^{\pm.007}$ | $0.404^{\pm.027}$ | $3.204^{\pm.027}$ | $2.192^{\pm.071}$ |
| | T2M-GPT (Zhang et al., 2023a) | $0.402^{\pm.006}$ | $0.619^{\pm.005}$ | $0.737^{\pm.006}$ | $0.717^{\pm.041}$ | $3.053^{\pm.026}$ | $1.912^{\pm.036}$ |
| | AttT2M (Zhong et al., 2023) | $0.413^{\pm.006}$ | $0.632^{\pm.006}$ | $0.751^{\pm.006}$ | $0.870^{\pm.039}$ | $3.039^{\pm.021}$ | $2.281^{\pm.047}$ |
| | FineMoGen (Zhang et al., 2023c) | $0.432^{\pm.006}$ | $0.649^{\pm.005}$ | $0.772^{\pm.006}$ | $0.178^{\pm.007}$ | $2.869^{\pm.014}$ | $1.877^{\pm.093}$ |
| | GraphMotion (Jin et al., 2023) | $0.429^{\pm.007}$ | $0.648^{\pm.006}$ | $0.769^{\pm.006}$ | $0.313^{\pm.013}$ | $3.076^{\pm.022}$ | $\mathbf{3.627}^{\pm.113}$ |
| | MMM (Pinyoanuntapong et al., 2024) | $0.404^{\pm.005}$ | $0.621^{\pm.005}$ | $0.744^{\pm.004}$ | $0.316^{\pm.028}$ | $2.977^{\pm.019}$ | $1.232^{\pm.039}$ |
| | MoMask (Guo et al., 2024) | $0.433^{\pm.007}$ | $0.656^{\pm.005}$ | $0.781^{\pm.005}$ | $0.204^{\pm.011}$ | $2.779^{\pm.022}$ | $1.131^{\pm.043}$ |
| | Light-T2M (Zeng et al., 2025) | $\underline{0.444}^{\pm.006}$ | $\underline{0.670}^{\pm.007}$ | $\underline{0.794}^{\pm.005}$ | $0.161^{\pm.009}$ | $2.746^{\pm.016}$ | $1.005^{\pm.036}$ |
| | MoGenTS (Yuan et al., 2024) | $\mathbf{0.445}^{\pm.006}$ | $\mathbf{0.671}^{\pm.006}$ | $\mathbf{0.797}^{\pm.005}$ | $\mathbf{0.143}^{\pm.004}$ | $\mathbf{2.711}^{\pm.024}$ | - |
| | **Event-T2M (Ours)** | $0.439^{\pm.005}$ | $0.669^{\pm.006}$ | $0.788^{\pm.005}$ | $\underline{0.159}^{\pm.004}$ | $\underline{2.742}^{\pm.016}$ | $0.762^{\pm.026}$ |
| Motion-X | AttT2M (Zhong et al., 2023) | $0.461^{\pm.004}$ | $0.664^{\pm.004}$ | $0.768^{\pm.004}$ | $0.232^{\pm.016}$ | $3.455^{\pm.015}$ | $\mathbf{2.053}^{\pm.043}$ |
| | MoMask (Guo et al., 2024) | $0.460^{\pm.004}$ | $0.662^{\pm.004}$ | $0.768^{\pm.004}$ | $0.297^{\pm.016}$ | $3.510^{\pm.018}$ | $1.442^{\pm.041}$ |
| | Light-T2M (Zeng et al., 2025) | $\underline{0.473}^{\pm.006}$ | $\underline{0.669}^{\pm.004}$ | $\underline{0.773}^{\pm.003}$ | $0.131^{\pm.012}$ | $\underline{3.409}^{\pm.017}$ | $1.594^{\pm.068}$ |
| | MoGenTS (Yuan et al., 2024) | $0.458^{\pm.003}$ | $0.664^{\pm.005}$ | $0.768^{\pm.004}$ | $\mathbf{0.102}^{\pm.008}$ | $3.498^{\pm.018}$ | $0.763^{\pm.034}$ |
| | **Event-T2M (Ours)** | $\mathbf{0.519}^{\pm.005}$ | $\mathbf{0.729}^{\pm.004}$ | $\mathbf{0.823}^{\pm.005}$ | $\underline{0.109}^{\pm.005}$ | $\mathbf{2.979}^{\pm.016}$ | $0.921^{\pm.035}$ |

## 4.1 BENCHMARKS AND METRICS

**Standard Benchmarks.** We adopt the official *train*, *val*, and *test* splits of HumanML3D (Guo et al., 2022), KIT-ML (Plappert et al., 2016), and Motion-X (Lin et al., 2023). Following prior work, motions are represented in root space with root velocities and local joint features. HumanML3D provides long and diverse descriptions, KIT-ML offers shorter prompts, and Motion-X serves as a large-scale dataset, allowing us to assess complex, simpler, and diverse settings.

**Event-stratified Subset: HumanML3D-E.** To examine performance under compositional complexity, we apply our LLM-based event decomposition to HumanML3D *test* prompts and group them by event count: at least 2 events, at least 3 events, and at least 4 events (e.g., "walk left, turn, jump, kick" falls into ≥4 group). These subsets provide increasingly challenging settings to test whether gains on simple prompts transfer to longer, sequential instructions. Full construction details are in Appendix A.5.

**Evaluation Metrics.** We follow the standard HumanML3D evaluation pipeline: (1) sample $N$ candidate motions per text ($N{=}20$ by default), (2) embed text and motion using the released evaluators, and (3) compute widely used metrics. Specifically, we report *FID* (generation realism), *R-Precision* (text–motion alignment, Top-1/2/3), *MM-Dist* (absolute alignment), and *MModality* (intra-prompt diversity). Following recent recommendations (Guo et al., 2024; Zeng et al., 2025), we omit the Diversity metric due to instability. All numbers are averaged over the test set with 95% confidence intervals estimated from repeated sampling. A detailed definition of each metric is provided in Appendix A.2.

Table 2: Comparison on the HumanML3D, KIT-ML, and Motion-X test sets with MARDM approaches. For each metric, "↑" denotes that larger values are better, while "↓" denotes that smaller values are better.

| Datasets | Methods | R-Precision ↑ | | | FID ↓ | MM-Dist ↓ | MModality ↑ | CLIP-score ↑ |
|---|---|---|---|---|---|---|---|---|
| | | Top-1 ↑ | Top-2 ↑ | Top-3 ↑ | | | | |
| HumanML3D | MARDM-DDPM (Meng et al., 2024) | $0.492^{\pm.006}$ | $0.690^{\pm.005}$ | $0.790^{\pm.005}$ | $0.116^{\pm.004}$ | $3.349^{\pm.010}$ | $2.470^{\pm.053}$ | $0.637^{\pm.005}$ |
| | MARDM-SiT (Meng et al., 2024) | $0.500^{\pm.004}$ | $0.695^{\pm.003}$ | $0.795^{\pm.003}$ | $\mathbf{0.114}^{\pm.007}$ | $3.270^{\pm.009}$ | $\mathbf{2.231}^{\pm.071}$ | $0.642^{\pm.002}$ |
| | **Event-T2M (Ours)** | $\mathbf{0.549}^{\pm.002}$ | $\mathbf{0.744}^{\pm.001}$ | $\mathbf{0.836}^{\pm.001}$ | $\mathbf{0.114}^{\pm.003}$ | $\mathbf{2.948}^{\pm.008}$ | $1.008^{\pm.052}$ | $\mathbf{0.665}^{\pm.001}$ |
| KIT-ML | MARDM-DDPM (Meng et al., 2024) | $0.375^{\pm.006}$ | $0.597^{\pm.008}$ | $0.739^{\pm.006}$ | $0.340^{\pm.020}$ | $3.489^{\pm.018}$ | $\mathbf{1.479}^{\pm.078}$ | $0.681^{\pm.003}$ |
| | MARDM-SiT (Meng et al., 2024) | $\mathbf{0.387}^{\pm.006}$ | $\mathbf{0.610}^{\pm.006}$ | $\mathbf{0.749}^{\pm.006}$ | $\mathbf{0.242}^{\pm.014}$ | $\mathbf{3.374}^{\pm.019}$ | $1.312^{\pm.053}$ | $\mathbf{0.692}^{\pm.002}$ |
| | **Event-T2M (Ours)** | $0.379^{\pm.005}$ | $0.599^{\pm.005}$ | $0.732^{\pm.006}$ | $0.273^{\pm.013}$ | $3.573^{\pm.022}$ | $0.933^{\pm.046}$ | $0.690^{\pm.001}$ |
| Motion-X | MARDM-DDPM (Meng et al., 2024) | $0.392^{\pm.003}$ | $0.592^{\pm.004}$ | $0.711^{\pm.004}$ | $0.132^{\pm.008}$ | $3.844^{\pm.014}$ | $\mathbf{2.058}^{\pm.067}$ | $0.639^{\pm.001}$ |
| | MARDM-SiT (Meng et al., 2024) | $0.405^{\pm.003}$ | $0.606^{\pm.004}$ | $0.721^{\pm.003}$ | $0.134^{\pm.006}$ | $3.761^{\pm.014}$ | $1.973^{\pm.061}$ | $0.648^{\pm.001}$ |
| | **Event-T2M (Ours)** | $\mathbf{0.547}^{\pm.002}$ | $\mathbf{0.743}^{\pm.002}$ | $\mathbf{0.834}^{\pm.002}$ | $\mathbf{0.115}^{\pm.004}$ | $\mathbf{2.942}^{\pm.007}$ | $0.963^{\pm.044}$ | $\mathbf{0.666}^{\pm.001}$ |

Table 3: Comparative results on HumanML3D-E against state-of-the-art baselines. "Condition 2/3/4" denotes prompts with at least 2, 3, and 4 events, respectively.

| Condition | Methods | R-Precision ↑ | | | FID ↓ | MM-Dist ↓ | MModality ↑ |
|---|---|---|---|---|---|---|---|
| | | Top-1 ↑ | Top-2 ↑ | Top-3 ↑ | | | |
| 2 | AttT2M (Zhong et al., 2023) | $0.479^{\pm.003}$ | $0.665^{\pm.003}$ | $0.761^{\pm.003}$ | $0.171^{\pm.007}$ | $3.181^{\pm.010}$ | $\underline{1.899}^{\pm.115}$ |
| | GraphMotion (Jin et al., 2023) | $0.468^{\pm.003}$ | $0.646^{\pm.003}$ | $0.741^{\pm.002}$ | $0.252^{\pm.012}$ | $3.302^{\pm.012}$ | $\mathbf{2.415}^{\pm.079}$ |
| | MoMask (Guo et al., 2024) | $\underline{0.497}^{\pm.004}$ | $\underline{0.691}^{\pm.003}$ | $\underline{0.790}^{\pm.003}$ | $\underline{0.065}^{\pm.002}$ | $3.061^{\pm.009}$ | $1.282^{\pm.043}$ |
| | Light-T2M (Zeng et al., 2025) | $0.462^{\pm.003}$ | $0.647^{\pm.003}$ | $0.747^{\pm.004}$ | $0.077^{\pm.004}$ | $3.278^{\pm.010}$ | $1.692^{\pm.058}$ |
| | MoGenTS (Yuan et al., 2024) | $0.496^{\pm.003}$ | $0.690^{\pm.002}$ | $0.787^{\pm.002}$ | $\mathbf{0.049}^{\pm.003}$ | $\underline{3.039}^{\pm.010}$ | $0.868^{\pm.037}$ |
| | **Event-T2M (Ours)** | $\mathbf{0.536}^{\pm.002}$ | $\mathbf{0.732}^{\pm.002}$ | $\mathbf{0.824}^{\pm.002}$ | $0.079^{\pm.003}$ | $\mathbf{2.836}^{\pm.006}$ | $0.976^{\pm.043}$ |
| 3 | AttT2M (Zhong et al., 2023) | $0.431^{\pm.005}$ | $0.613^{\pm.005}$ | $0.715^{\pm.004}$ | $0.464^{\pm.031}$ | $3.329^{\pm.018}$ | $1.960^{\pm.105}$ |
| | GraphMotion (Jin et al., 2023) | $0.420^{\pm.006}$ | $0.599^{\pm.007}$ | $0.698^{\pm.006}$ | $0.458^{\pm.026}$ | $3.440^{\pm.023}$ | $\mathbf{2.427}^{\pm.065}$ |
| | MoMask (Guo et al., 2024) | $\underline{0.466}^{\pm.006}$ | $\underline{0.652}^{\pm.006}$ | $\underline{0.752}^{\pm.005}$ | $\underline{0.142}^{\pm.008}$ | $3.169^{\pm.015}$ | $1.320^{\pm.038}$ |
| | Light-T2M (Zeng et al., 2025) | $0.404^{\pm.005}$ | $0.594^{\pm.006}$ | $0.699^{\pm.004}$ | $0.193^{\pm.009}$ | $3.396^{\pm.015}$ | $\underline{1.740}^{\pm.055}$ |
| | MoGenTS (Yuan et al., 2024) | $0.452^{\pm.004}$ | $0.644^{\pm.005}$ | $0.751^{\pm.005}$ | $0.147^{\pm.009}$ | $\underline{3.122}^{\pm.018}$ | $0.894^{\pm.028}$ |
| | **Event-T2M (Ours)** | $\mathbf{0.487}^{\pm.005}$ | $\mathbf{0.687}^{\pm.004}$ | $\mathbf{0.790}^{\pm.004}$ | $\mathbf{0.137}^{\pm.003}$ | $\mathbf{2.928}^{\pm.010}$ | $1.010^{\pm.029}$ |
| 4 | AttT2M (Zhong et al., 2023) | $0.407^{\pm.013}$ | $0.581^{\pm.010}$ | $0.688^{\pm.010}$ | $1.077^{\pm.104}$ | $3.455^{\pm.041}$ | $2.049^{\pm.099}$ |
| | GraphMotion (Jin et al., 2023) | $0.399^{\pm.012}$ | $0.615^{\pm.012}$ | $0.723^{\pm.010}$ | $0.857^{\pm.056}$ | $3.521^{\pm.049}$ | $\mathbf{2.547}^{\pm.066}$ |
| | MoMask (Guo et al., 2024) | $\underline{0.441}^{\pm.013}$ | $\underline{0.633}^{\pm.014}$ | $\underline{0.734}^{\pm.013}$ | $0.418^{\pm.030}$ | $\underline{3.205}^{\pm.042}$ | $1.334^{\pm.046}$ |
| | Light-T2M (Zeng et al., 2025) | $0.365^{\pm.010}$ | $0.552^{\pm.006}$ | $0.662^{\pm.010}$ | $0.627^{\pm.027}$ | $3.586^{\pm.027}$ | $\underline{1.863}^{\pm.064}$ |
| | MoGenTS (Yuan et al., 2024) | $0.420^{\pm.012}$ | $0.613^{\pm.010}$ | $0.715^{\pm.013}$ | $0.423^{\pm.038}$ | $3.241^{\pm.039}$ | $0.879^{\pm.032}$ |
| | **Event-T2M (Ours)** | $\mathbf{0.466}^{\pm.008}$ | $\mathbf{0.660}^{\pm.008}$ | $\mathbf{0.767}^{\pm.007}$ | $\mathbf{0.265}^{\pm.007}$ | $\mathbf{3.063}^{\pm.015}$ | $1.039^{\pm.028}$ |

## 4.2 MAIN RESULTS

**Standard test sets (HumanML3D, KIT-ML, and Motion-X).** On the standard HumanML3D, KIT-ML, and Motion-X test splits, Event-T2M achieves performance on par with recent strong baselines (Table 1 and 2). This demonstrates that the proposed event-based cross-attention preserves competitiveness on the simple, single-event prompts that dominate existing benchmarks, ensuring that our improvements on complex prompts do not come at the expense of overall accuracy.

**Event-stratified sets (HumanML3D-E).** We emphasize that all models are trained on the standard HumanML3D training set and only evaluated on HumanML3D-E. On these event-stratified subsets, Event-T2M exhibits consistent and substantial improvements, particularly under the most demanding setting of ≥4 events (Table 3, Figure 2a). As event count increases, baseline methods that rely on a single global text embedding frequently underfit later actions or conflate multiple events, leading to degraded quality (*FID*) and weaker alignment (*R-Precision*). In contrast, Event-T2M's explicit event-level conditioning enables sequentially faithful synthesis, preserving both action order and smooth transitions.

**Efficiency analysis.** Figure 2b plots *FID* at ≥4 events against the number of trainable parameters. Event-T2M occupies a favorable point on this curve: it achieves substantially better fidelity under complex prompts while maintaining parameter counts comparable to, or smaller than, recent baselines.

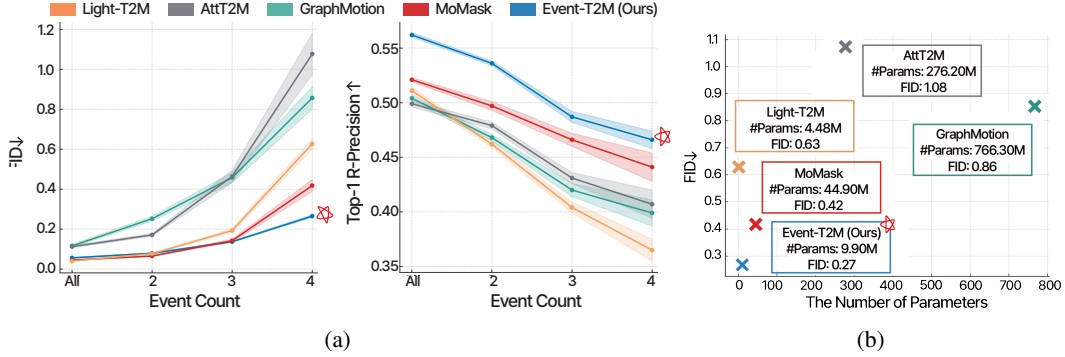

Figure 2: Overall comparison of Event-T2M: (a) As event counts increase ($\geq 1, \geq 2, \geq 3, \geq 4$), Event-T2M consistently achieves the lowest *FID* and the highest *R-Precision*, while baselines degrade sharply under compositional complexity. (b) Efficiency analysis at $\geq 4$ events shows that Event-T2M achieves high accuracy with low model size, demonstrating its compactness and scalability.

Table 4: Ablation study on text encoders and conditioning methods on HumanML3D-E.

| Text Encoder | Condition | Methods | R-Precision ↑ | | | FID ↓ | MM-Dist ↓ | MModality ↑ |
|---|---|---|---|---|---|---|---|---|
| | | | Top-1 ↑ | Top-2 ↑ | Top-3 ↑ | | | |
| TMR | 2 | Event-T2M (Token-level) | $0.521^{\pm.003}$ | $0.718^{\pm.002}$ | $0.815^{\pm.002}$ | $0.082^{\pm.003}$ | $2.915^{\pm.008}$ | $\mathbf{0.999}^{\pm.032}$ |
| | | **Event-T2M (Event-level)** | $\mathbf{0.536}^{\pm.002}$ | $\mathbf{0.732}^{\pm.002}$ | $\mathbf{0.824}^{\pm.002}$ | $\mathbf{0.079}^{\pm.003}$ | $\mathbf{2.836}^{\pm.006}$ | $0.976^{\pm.043}$ |
| | 3 | Event-T2M (Token-level) | $0.463^{\pm.005}$ | $0.664^{\pm.005}$ | $0.773^{\pm.003}$ | $0.162^{\pm.006}$ | $3.031^{\pm.009}$ | $\mathbf{1.035}^{\pm.045}$ |
| | | **Event-T2M (Event-level)** | $\mathbf{0.487}^{\pm.005}$ | $\mathbf{0.687}^{\pm.004}$ | $\mathbf{0.790}^{\pm.004}$ | $\mathbf{0.137}^{\pm.003}$ | $\mathbf{2.928}^{\pm.010}$ | $1.010^{\pm.029}$ |
| | 4 | Event-T2M (Token-level) | $0.440^{\pm.011}$ | $0.635^{\pm.010}$ | $0.740^{\pm.009}$ | $0.355^{\pm.011}$ | $3.168^{\pm.016}$ | $\mathbf{1.141}^{\pm.026}$ |
| | | **Event-T2M (Event-level)** | $\mathbf{0.466}^{\pm.008}$ | $\mathbf{0.660}^{\pm.008}$ | $\mathbf{0.767}^{\pm.007}$ | $\mathbf{0.265}^{\pm.007}$ | $\mathbf{3.063}^{\pm.015}$ | $1.039^{\pm.028}$ |
| CLIP | 2 | Event-T2M (Token-level) | $0.474^{\pm.003}$ | $0.664^{\pm.003}$ | $0.767^{\pm.003}$ | $0.153^{\pm.004}$ | $3.149^{\pm.010}$ | $\mathbf{1.875}^{\pm.057}$ |
| | | **Event-T2M (Event-level)** | $\mathbf{0.494}^{\pm.003}$ | $\mathbf{0.681}^{\pm.003}$ | $\mathbf{0.779}^{\pm.003}$ | $\mathbf{0.052}^{\pm.002}$ | $\mathbf{3.079}^{\pm.010}$ | $1.577^{\pm.060}$ |
| | 3 | Event-T2M (Token-level) | $0.423^{\pm.006}$ | $\mathbf{0.618}^{\pm.005}$ | $0.728^{\pm.004}$ | $0.206^{\pm.008}$ | $3.254^{\pm.011}$ | $\mathbf{1.905}^{\pm.056}$ |
| | | **Event-T2M (Event-level)** | $0.423^{\pm.005}$ | $\mathbf{0.618}^{\pm.005}$ | $\mathbf{0.729}^{\pm.005}$ | $\mathbf{0.141}^{\pm.004}$ | $\mathbf{3.245}^{\pm.015}$ | $1.627^{\pm.052}$ |
| | 4 | Event-T2M (Token-level) | $\mathbf{0.399}^{\pm.012}$ | $\mathbf{0.597}^{\pm.010}$ | $\mathbf{0.709}^{\pm.010}$ | $0.468^{\pm.021}$ | $3.339^{\pm.032}$ | $\mathbf{1.991}^{\pm.060}$ |
| | | **Event-T2M (Event-level)** | $0.374^{\pm.010}$ | $0.578^{\pm.007}$ | $0.690^{\pm.007}$ | $\mathbf{0.425}^{\pm.022}$ | $3.467^{\pm.022}$ | $1.674^{\pm.059}$ |

## 4.3 ABLATIONS AND ANALYSIS

**Effect of ECA.** Table 4 shows that adding the ECA consistently improves performance on HumanML3D-E. *R-Precision* rises across conditions, reflecting stronger text–motion alignment, while *FID* decreases, indicating more coherent and realistic motion. Unlike token-level attention, which disperses semantics across individual words and often fails to preserve ordered dependencies, ECA grounds generation directly in event tokens. This targeted conditioning prevents the model from collapsing sequential actions, enabling it to respect event order with greater fidelity.

**Effect of Text Encoder.** Replacing CLIP with TMR yields consistent improvements in *R-Precision*, particularly on prompts with $\geq 3$ events. While CLIP's large-scale image–text pretraining captures simple, single-action semantics, it lacks the motion-specific knowledge needed for sequential behaviors. TMR, trained directly on motion–text pairs, provides richer event-centric representations. As a result, Event-T2M with TMR not only maintains competitive performance on simple cases but achieves clear gains on multi-event prompts, validating our hypothesis that motion-aware encoders are crucial for scaling beyond single-event benchmarks.

## 4.4 USER STUDY

We conducted two user studies with distinct goals; Full details are in Appendix A.6 and A.7.

**Study 1: Validating event decomposition.** To test whether our event definition yields a convincing evaluation basis, we compared three alternatives: (1) human-annotated action splits, (2) verb-aware LLM segmentation, and (3) our event-aware LLM segmentation. Verb-aware segmentation simply splits prompts by action verbs (e.g., "run"), without considering temporal coherence

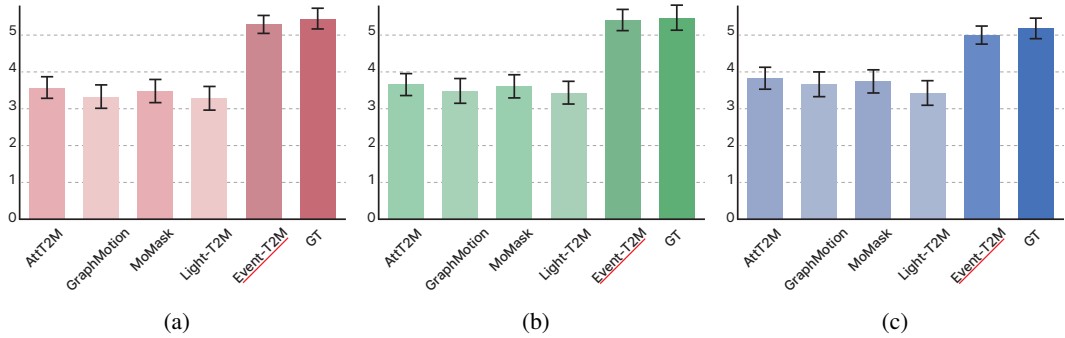

Figure 3: Results of the user study (7-point Likert scale). Error bars denote standard errors. (a) Fidelity, (b) Order alignment, and (c) Naturalness. Event-T2M achieves significant gains over all competing methods and performs on par with ground-truth (GT).

Figure 4: Qualitative comparison with a complex multi-event prompt. Event-T2M executes all events in order and with correct counts, while baselines often fail to generate them faithfully. See supplementary video for full motions.

or semantic self-containment. For each participant, 20 prompts were randomly sampled from HumanML3D-E with $\geq 3$ events, and human evaluators rated whether the resulting segmentation was natural and distinguishable.

The study analysis presents that event-aware segmentation was rated on par with human annotation and significantly better than naive verb-aware segmentation. This validates both the plausibility of our event definition and the reliability of HumanML3D-E as a benchmark.

**Study 2: Perceptual Validation of Event-T2M.** To examine the perceptual quality of motions generated by Event-T2M, we compared it with baselines (AttT2M, GraphMotion, Light-T2M, and MoMask) and with ground-truth. Human evaluators rated 20 samples from HumanML3D-E along three criteria: (1) how well the motion follows the text without omissions or additions (Fidelity), (2) how well the motion follows the order specified in the text (Order alignment), and (3) how natural the motion appears (Naturalness). Ratings were collected on a 7-point Likert scale.

As shown in Figure 3, Event-T2M consistently outperformed baselines and achieved scores indistinguishable from ground-truth, demonstrating its strength in faithfully generating multi-event motions.

## 4.5 QUALITATIVE RESULTS

Figure 4 illustrates model outputs for the challenging prompt "A man walks forward and kicks with one foot two times and walks backward then pauses and then kicks into the air two times.", which contains seven distinct events. Among the methods, Event-T2M is the only one that realizes all actions in the correct sequence while ensuring smooth transitions. In contrast, baselines often shorten the motion, blend distinct events, or substitute unrelated actions. As further confirmed in additional examples (see supplementary video and Appendix A.8), Event-T2M faithfully maintains both event semantics and temporal order.

## 5 FAILURE CASE ANALYSIS

To better understand the limitations of existing text-to-motion models in complex scenarios, we conduct a targeted failure case analysis on text prompts containing $\geq 4$ events. We specifically focus on long, compositional descriptions that require preserving a chain of distinct sub-actions. Across such prompts, we observe a consistent pattern: existing models (AttT2M, GraphMotion, Light-T2M, MARDM (Meng et al., 2024), MoGenTS (Yuan et al., 2024), and MoMask) frequently omit events, generate incorrect or spurious events, and occasionally produce motions that exceed their maximum motion length or exhibit clear physical artifacts. In particular, MARDM and MoMask often fail to reconstruct the full event sequence, either by dropping intermediate sub-motions or by inserting unintended transitions, suggesting that they struggle to faithfully realize multi-stage, event-rich instructions.

These tendencies are illustrated by the following example: *"Man is standing straight, feet not moving, hinges at the waist to reach both hands down to his feet, then puts his arms up, bent at the elbows, twists his torso to the left, and then to the right, and then facing forward leans over to the left, and then over to the right, stretching."* For this prompt, AttT2M fails to realize the "hinges at the waist" event, GraphMotion exhibits both event omission and physical errors, Light-T2M and MARDM omit events and sometimes exceed the maximum motion length, and MoGenTS and MoMask also miss parts of the described sequence. By contrast, Event-T2M generates all of the described events, albeit with a slightly permuted order, indicating that it can still cover the complete set of intended sub-actions even when the motion is long and structurally complex.

Overall, these observations suggest that Event-T2M is comparatively more robust at preserving the full set of events in complex, multi-event prompts than prior sentence-level or token-level approaches. We attribute this robustness to the event-level formulation, which encourages the model to align distinct motion segments with explicit semantic units, thereby reducing the likelihood of dropping, conflating, or corrupting events during generation.

## 6 CONCLUSION

We presented Event-T2M, a diffusion framework that leverages event-level decomposition and cross-attention to synthesize sequentially complex motions from natural language. Across HumanML3D, KIT-ML, and our event-stratified HumanML3D-E, Event-T2M demonstrates strong performance on standard test sets and consistent improvements under multi-event prompts. Human evaluations further confirm that our model generates motions that preserve both order and semantics while maintaining naturalness comparable to real data.

Our work establishes explicit event-level conditioning as a scalable recipe for robust text-to-motion generation, moving the field beyond single-event benchmarks. Nonetheless, challenges remain: current models do not consider long-horizon physical plausibility, natural human–object interactions, and seamless integration into downstream applications such as animation pipelines, embodied agents, and video production. Future directions include incorporating physics-aware objectives, enabling fine-grained event editing, and extending event-based conditioning to multimodal settings involving vision and audio.

## 7 REPRODUCIBILITY STATEMENT

To facilitate reproducibility, we provide detailed descriptions of our model architecture, training setup, and evaluation protocols in the main text and supplementary. In addition, the full implementation and code are included in the supplementary materials, enabling independent researchers to replicate our results.

ACKNOWLEDGEMENT

This work was supported by the National Research Foundation of Korea (NRF) grants funded by the Korean government (MSIT) (No. RS-2025-00518643, No. RS-2025-24802983), and by the ICT Creative Consilience Program through the Institute of Information & Communications Technology Planning & Evaluation (IITP) grant funded by the Korea government (MSIT) (IITP-2026-RS-2020-II201819).

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

# A    APPENDIX

## A.1    IMPLEMENTATION DETAILS

We set the maximum diffusion step to 1000, with linearly increasing variances $\beta_t$ ranging from $1 \times 10^{-4}$ to $2 \times 10^{-2}$. For fast inference, we employ UniPC (Zhao et al., 2023) using 10 time steps. The model architecture consists of $N = 4$ blocks with a hidden dimension of 256 and a downsampling factor of 8. The guidance scale is fixed at 4, while text dropout is applied with probability 0.2. Training is carried out with AdamW (Loshchilov & Hutter, 2017), using a learning rate of $1 \times 10^{-4}$, cosine annealing scheduling, and a batch size of 128 on two NVIDIA RTX 4090 GPUs. We train for 600 epochs on HumanML3D and 1,000 epochs on KIT-ML.

During training, checkpoints are saved at regular intervals, and the final model is selected based on the lowest *FID* score on the validation set. In particular, each Event-level Conditioning Module uses a local convolution width of 4, and a block expansion factor of 2. The Depth-wise Conv1D layers have a kernel size of 3 and a stride of 1.

## A.2    EVALUATION METRICS DETAILS

We evaluate our model using several widely adopted measures introduced in T2M (Guo et al., 2022). Below, we briefly describe each metric without relying on explicit formulae.

*FID* assesses how close the generated motions are to real human motions in terms of feature distribution. Specifically, it compares the mean and covariance of features extracted from generated samples with those from ground-truth. A smaller value indicates that the distribution of generated motions better matches the real data.

*R-Precision* measures the semantic consistency between text descriptions and generated motions. For each motion, we form a candidate pool of text descriptions that includes the correct caption and several distractors randomly sampled from the dataset. If the true description is found within the top-$k$ retrieved captions when ranking by similarity, the retrieval is counted as correct. We report results for $k = 1, 2$, and 3 to capture different levels of retrieval difficulty.

MM-Dist evaluates the alignment between a generated motion and its paired textual description. It computes the average distance between their respective feature embeddings. Lower values indicate stronger semantic correspondence between the motion and its caption.

*Multimodality* focuses on variation among motions generated from the same text description. For each caption, multiple motion samples are generated, and their feature differences are averaged. A model that achieves higher scores on this metric is better at producing a wide range of plausible motions from identical textual input.

## A.3    DETAILS ON HUMANML3D AND KIT-ML

The HumanML3D corpus (Guo et al., 2022) is built by integrating motion data from two large-scale sources: HumanAct12 (Guo et al., 2020) and AMASS (Mahmood et al., 2019). These source datasets cover a broad spectrum of movements, including everyday behaviors like walking or jumping, athletic activities such as swimming and karate, acrobatic skills like cartwheels, and performance-oriented motions such as dancing.

For consistency, the raw motion clips are standardized to 20 frames per second (FPS). All data are retargeted onto a unified skeleton and aligned so that the character initially faces the positive Z direction. To attach natural language descriptions, annotations were collected via Amazon Mechanical Turk (AMT). Workers were instructed to write descriptions, and each motion was labeled independently by almost three different annotators.

In total, HumanML3D provides 14,616 motion clips paired with 44,970 descriptions, using a vocabulary of 5,371 distinct tokens. The dataset amounts to about 28.6 hours of motion, with clips ranging from 2–10 seconds (average length 7.1s). Captions are on average 12 words long, with a median of 10. To further enrich the dataset, mirroring was applied: for instance, the sequence "A man kicks something or someone with his left leg" was mirrored and relabeled as "A man kicks something or someone with his right leg," ensuring balanced left and right motion coverage.

Table 5: Sampling Step ablation. R represents *R-Precision*.

| Condition | Step | FID ↓ | R Top-1 ↑ | R Top-3 ↑ |
|---|---|---|---|---|
| 2 | 5 | 0.103 | 0.512 | 0.805 |
|  | 7 | **0.069** | 0.529 | 0.820 |
|  | 10 | 0.079 | **0.536** | **0.824** |
|  | 20 | 0.096 | 0.530 | 0.820 |
| 3 | 5 | 0.164 | 0.471 | 0.775 |
|  | 7 | 0.138 | **0.490** | 0.788 |
|  | 10 | 0.137 | 0.487 | 0.790 |
|  | 20 | **0.134** | 0.484 | **0.792** |
| 4 | 5 | 0.280 | 0.449 | 0.745 |
|  | 7 | 0.292 | 0.460 | 0.757 |
|  | 10 | **0.265** | **0.466** | **0.767** |
|  | 20 | 0.271 | 0.461 | 0.754 |

Table 6: CFG Scale ablation.

| Condition | Scale | FID ↓ | R Top-1 ↑ | R Top-3 ↑ |
|---|---|---|---|---|
| 2 | 3 | 0.054 | 0.534 | 0.823 |
|  | 4 | **0.079** | **0.536** | **0.824** |
|  | 5 | 0.092 | 0.517 | 0.712 |
|  | 6 | 0.171 | 0.498 | 0.800 |
| 3 | 3 | **0.112** | 0.483 | 0.788 |
|  | 4 | 0.137 | **0.487** | **0.790** |
|  | 5 | 0.186 | 0.463 | 0.767 |
|  | 6 | 0.223 | 0.465 | 0.769 |
| 4 | 3 | 0.335 | **0.466** | 0.745 |
|  | 4 | **0.265** | **0.466** | **0.767** |
|  | 5 | 0.368 | 0.464 | 0.757 |
|  | 6 | 0.413 | 0.437 | 0.718 |

Table 7: Ablation study on the architecture design (Transformer vs. Conformer).

| Condition | Backbone | FID ↓ | R Top-1 ↑ | R Top-3 ↑ |
|---|---|---|---|---|
| 2 | Transformer | 0.080 | 0.453 | 0.736 |
|  | Conformer | **0.079** | **0.536** | **0.824** |
| 3 | Transformer | 0.187 | 0.402 | 0.700 |
|  | Conformer | **0.137** | **0.487** | **0.790** |
| 4 | Transformer | 0.533 | 0.373 | 0.670 |
|  | Conformer | **0.265** | **0.466** | **0.767** |

The KIT-ML dataset (Plappert et al., 2016) contains 3,911 motion sequences with 6,278 associated textual annotations. The text spans a vocabulary of 1,623 words, normalized to ignore capitalization and punctuation. Motions originate from the KIT (Plappert et al., 2016) and CMU (De la Torre et al., 2009) motion capture datasets, but are resampled at 12.5 FPS. Each sequence is paired with between one and four textual descriptions, averaging roughly 8 words per sentence.

## A.4 FURTHER RESULTS OF ABLATION STUDY

We conducted ablation studies to determine the sampling steps and CFG scaling for our model, and the results are summarized in Table 5 and 6, respectively. In addition, we examined whether to adopt a Transformer or a Conformer within the ECA module, and the results are reported in Table 7.

## A.5 DETAILS ON HUMANML3D-E

For reproducibility of our HumanML3D-E, we provide the prompts in Table 8 and 9. The LLM we used is Gemini 2.5 Flash, which not only demonstrates strong performance but also offers significantly faster speed compared to other LLMs. The HumanML3D test set contains 4,646 samples, with 2,622 in Condition 2, 927 in Condition 3, and 260 in Condition 4. The visualization of these statistics is shown in Figure 5a.

## A.6 DETAILS OF STUDY 1: VALIDATING EVENT DECOMPOSITION

In our user study, we involved 21 participants (11 male, 10 female; $\mu = 26.38$ years, $\sigma = 4.60$, range = 22-44). Each participant evaluated 20 samples per condition using a 7-point Likert scale. The average ratings were: event-aware prompt ($\mu = 6.08$, $\sigma = 1.03$), verb-aware prompt ($\mu = 5.07$, $\sigma = 1.62$), and human ($\mu = 6.09$, $\sigma = 1.00$). Because the rating data did not satisfy normality assumptions, we adopted non-parametric methods. A Friedman test revealed a significant effect of condition ($\chi^2(2) = 11.16$, $p < .01$). Pairwise Wilcoxon signed-rank tests with Holm adjustment showed that the event-aware prompt and verb-aware prompt differed significantly ($p < .01$), as

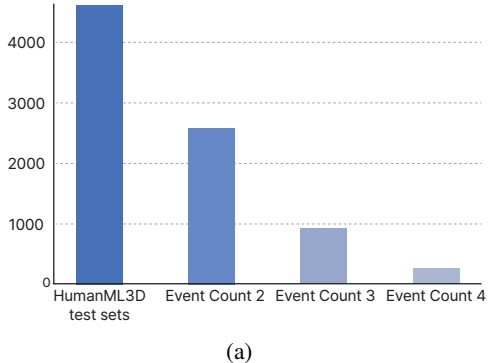
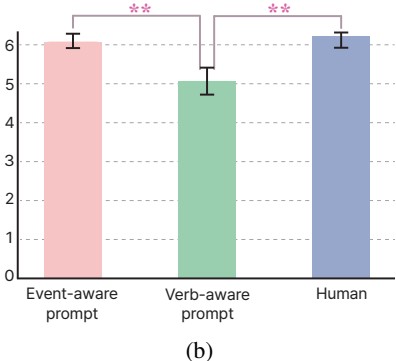

(a)                                                                          (b)

Figure 5: (a) Number of samples in the HumanML3D test set and HumanML3D-E. (b) User study of prompts. Error bars denote standard errors. Asterisks denote statistical significance ($**$: $p < 0.01$).

did the verb-aware prompt and human ($p < .01$). In contrast, no reliable difference was observed between the event-aware prompt and human ($p = 0.1546$). These findings suggest that the verb-aware prompt condition was consistently rated lower compared to both the event-aware prompt and human, while ratings for the event-aware prompt and human were statistically comparable. Figure 5b summarizes these outcomes.

## A.7 DETAILS OF STUDY 2: PERCEPTUAL VALIDATION OF EVENT-T2M

We conducted a user study with 20 participants (11 male, 9 female; $\mu = 27.8$ years, $\sigma = 6.25$, range = 22–46). Each participant evaluated motion outputs from six conditions—AttT2M, GraphMotion, MoMask, Light-T2M, Event-T2M, and ground-truth—on a 7-point Likert scale.

Across the three metrics—Fidelity, Order consistency, and Naturalness—clear differences were observed. Event-T2M and Ground-truth achieved the highest scores. Event-T2M showed $\mu = 5.29$, $\sigma = 1.23$ (Fidelity), $\mu = 5.41$, $\sigma = 1.25$ (Order consistency), and $\mu = 5.03$, $\sigma = 1.40$ (Naturalness). Ground-truth performed similarly with $\mu = 5.45$, $\sigma = 1.39$, $\mu = 5.47$, $\sigma = 1.38$, and $\mu = 5.20$, $\sigma = 1.63$. In contrast, the other models remained in the mid-3 range: AttT2M ($\mu = 3.57$, $\sigma = 1.43$; $\mu = 3.64$, $\sigma = 1.47$; $\mu = 3.84$, $\sigma = 1.42$), GraphMotion ($\mu = 3.33$, $\sigma = 1.56$; $\mu = 3.48$, $\sigma = 1.62$; $\mu = 3.67$, $\sigma = 1.59$), Light-T2M ($\mu = 3.28$, $\sigma = 1.60$; $\mu = 3.43$, $\sigma = 1.63$; $\mu = 3.44$, $\sigma = 1.49$), and MoMask ($\mu = 3.48$, $\sigma = 1.52$; $\mu = 3.61$, $\sigma = 1.58$; $\mu = 3.77$, $\sigma = 1.54$). These results indicate that Event-T2M and Ground-truth clearly outperformed the other methods across all evaluation dimensions.

Since normality assumptions were not met, we employed non-parametric tests. Friedman tests revealed significant effects of condition for all three criteria (Fidelity: $\chi^2(5) = 53.87$, $p < .01$; Order alignment: $\chi^2(5) = 51.82$, $p < .01$; Naturalness: $\chi^2(5) = 47.85$, $p < .01$). Pairwise Wilcoxon signed-rank tests with Holm correction further indicated that Event-T2M consistently outperformed all competing models ($p < .01$ across comparisons). In contrast, comparisons between Event-T2M and ground-truth did not yield reliable differences (Fidelity: $p = .0890$; Order alignment: $p = .2905$; Naturalness: $p = .2785$).

These results suggest that Event-T2M achieves ratings comparable to ground truth while significantly surpassing existing baselines across all evaluation aspects.

## A.8 FURTHER VISUALIZATION

Figure 6 compares generations for the prompt "A person steps backward, jumps up, runs forward, then runs backward." Event-T2M alone faithfully executes all four actions in the correct order, with smooth transitions. In contrast, baselines either truncate the sequence, merge events, or substitute incorrect motions.

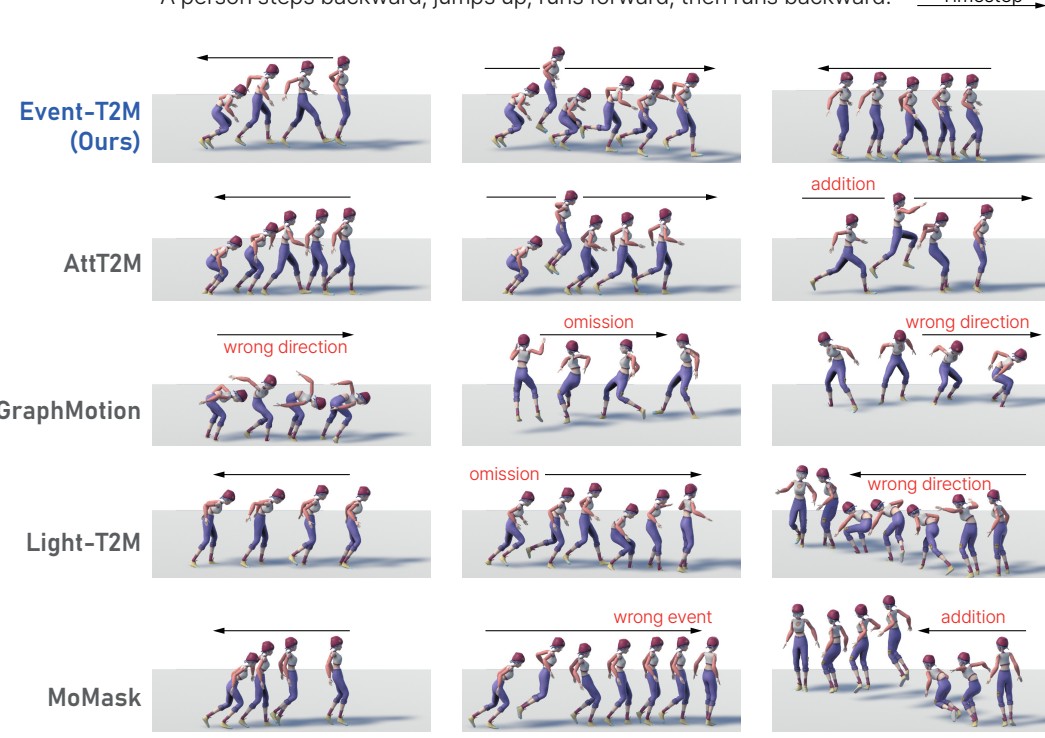

Figure 6: Further qualitative comparison.

## A.9 RESULTS OF DECOMPOSITION BY LLM PROMPTS

The quality of event decomposition is highly dependent on the instructions provided to the LLM. To investigate this, we designed two different prompting strategies, as shown in Table 8 and 9:

Quantitative results in Figure 7 show that the event-aware prompt produces more reliable and consistent decompositions. When using event-aware decomposition, Event-T2M achieves higher *R-Precision* and lower *FID*, particularly in the $\geq 3$ and $\geq 4$ event subsets. By contrast, the verb-aware prompt often over-segments or under-segments the text (e.g., splitting "walk forward while waving" into two independent actions, or merging "run, stop, and jump" into one), which introduces noise in the event tokens and weakens the conditioning signal. This results in degraded alignment between generated motions and text, as reflected in both retrieval-based and distributional metrics.

These findings confirm that a carefully designed event-aware prompt is crucial for leveraging LLMs in motion-text decomposition. Rather than naively extracting all actions, grounding the segmentation process in a principled definition of events yields more stable event tokens and stronger downstream performance.

## A.10 LLM USAGE STATEMENT

In preparing this paper, we used an LLM solely as a writing assistance tool for grammar correction and minor language polishing. The LLM was not involved in research ideation, methodology design, data analysis, or result interpretation. All scientific content, experiments, and conclusions were fully conceived and verified by the authors.

**"a person steps forward, sits down, taps feet together while rubbing hands together."**

| Event-aware | Verb-aware |
|---|---|
| - "a person steps forward."
- "a person sits down."
- "a person taps feet together while rubbing hands together." | - "a person steps forward."
- "a person sits down."
- "a person taps feet together."
- "a person rubs hands together." |

**"a standing person is swaying gently back and forth
as he holds his left hand to the left side of his head for a moment.
he drops his arm, briefly, then raises it to look at his hand."**

| Event-aware | Verb-aware |
|---|---|
| - "a standing person is swaying gently back and forth as he holds his left hand to the left side of his head for a moment."
- "a standing person drops his arm, briefly."
- "a standing person then raises it to look at his hand." | - "a standing person is swaying gently back and forth."
- "a standing person holds his left hand to the left side of his head for a moment."
- "a standing person drops his arm, briefly."
- "a standing person then raises it to look at his hand." |

**"a person kicks with their right leg twice, and then once with their left."**

| Event-aware | Verb-aware |
|---|---|
| - "a person kicks with their right leg."
- "a person kicks with their right leg."
- "a person kicks once with their left." | - "a person kicks with their right leg twice."
- "a person then kicks once with their left." |

**"a person is making rapid swinging motions with their right leg in the air,
while holding onto something with their right hand."**

| Event-aware | Verb-aware |
|---|---|
| - "a person is making rapid swinging motions with their right leg in the air, while holding onto something with their right hand." | - "a person is making rapid swinging motions with their right leg in the air."
- "a person is holding onto something with their right hand." |

**"a person dodges things thrown at them by blocking with their left hand and then ducking."**

| Event-aware | Verb-aware |
|---|---|
| - "a person dodges things thrown at them by blocking with their left hand."
- "a person then ducks." | - "a person dodges things thrown at them."
- "a person blocks with their left hand."
- "a person then ducks." |

Figure 7: Overall comparison of Event-T2M.

Table 8: Event-aware prompt: Incorporates our proposed definition of event to guide segmentation.

---

**Event-aware Prompt**

---

**Please segment a single input sentence into multiple sentences that each represent a distinct event, following the rules below:**

- A bundle of actions performed simultaneously at a specific point in time is defined as a single "event".
- Each segmented sentence must start with the subject used in the original sentence (e.g., "a person", "a man", etc.).
- Do not remove or simplify any adverbs, adjectives, or modifiers that appear in the original sentence — preserve them as much as possible.
- Parts separated by the # symbol must be included in each segmented sentence.
- Do not add any new sentences — only break down the given text as instructed.
- Do not include your thinking process or output reasoning. Only output the segmented sentences following the format.
- If the sentence cannot be further segmented into multiple events, leave it as is and output the original sentence without any changes.
- Even if there are grammatical errors in the sentence, proceed with the processing.
- If the input sentence contains multiple actions, the output must contain the same number of actions as the input sentence.

---

Good Example 1 - Input

a man lifts something on his left and places it down on his right.
#a/DET man/NOUN lift/VERB something/PRON on/ADP his/DET left/NOUN and/CCONJ place/VERB it/PRON down/ADP on/ADP his/DET right/NOUN#0.0#0.0

---

Good Example 1 - Output

a man lifts something on his left.#a/DET man/NOUN lift/VERB something/PRON on/ADP his/DET left/NOUN#0.0#0.0
a man places it down on his right.#a/DET man/NOUN place/VERB it/PRON down/ADP on/ADP his/DET right/NOUN#0.0#0.0

---

Good Example 2 - Input

a man kicks something with his left leg.#a/DET man/NOUN kick/VERB something/PRON with/ADP his/DET left/ADJ leg/NOUN#0.0#0.0

---

Good Example 2 - Output

a man kicks something with his left leg.#a/DET man/NOUN kick/VERB something/PRON with/ADP his/DET left/ADJ leg/NOUN#0.0#0.0

---

---

**Good Example 3 - Input**

A person waves their hand while stepping sideways, then jumps up and spins, and finally lands and bows. #A/DET person/NOUN wave/VERB their/DET hand/NOUN while/SCONJ step/VERB sideways/ADV then/ADV jump/VERB up/ADV and/CCONJ spin/VERB and/CCONJ finally/ADV land/VERB and/CCONJ bow/VERB#0.0#0.0

---

**Good Example 3 - Output**

A person waves their hand while stepping sideways.#A/DET person/NOUN wave/VERB their/DET hand/NOUN while/SCONJ step/VERB sideways/ADV#0.0#0.0
A person jumps up and spins.#A/DET person/NOUN jump/VERB up/ADV and/CCONJ spin/VERB#0.0#0.0
A person lands and bows.#A/DET person/NOUN land/VERB and/ CCONJ bow/VERB#0.0#0.0

---

**Bad Example 1 - Input**

someone is sprinting side to side#someone/PRON is/AUX sprint/VERB side/NOUN to/PART side/VERB#0.0#0.0

---

**Bad Example 1 - Output**

The input sentence cannot be segmented as requested because "side to side" is a single adverbial phrase modifying the verb "sprinting," and "side" is used as a noun and a prepositional phrase. The sentence only contains one action (sprinting).

---

**Bad Example 2 - Input**

A person claps their hands while sitting on the ground. #A/DET person/NOUN clap/VERB their/DET hands/NOUN while/SCONJ sit/VERB on/ADP the/DET ground/NOUN#0.0#0.0

---

**Bad Example 2 - Output**

A person claps their hands. #A/DET person/NOUN clap/VERB their/DET hands/NOUN#0.0#0.0 A person sits on the ground. #A/DET person/NOUN sit/VERB on/ADP the/DET ground/NOUN#0.0#0.0

---

**Bad Example 3 - Input**

A person bends their knees and raises both arms at the same time. #A/DET person/NOUN bend/VERB their/DET knees/NOUN and/CCONJ raise/VERB both/DET arms/NOUN at/ADP the/DET same/ADJ time/NOUN#0.0#0.0

---

**Bad Example 3 - Output**

A person bends their knees.#A/DET person/NOUN bend/VERB their/DET knees/NOUN#0.0#0.0
A person raises both arms.#A/DET person/NOUN raise/VERB both/DET arms/NOUN#0.0#0.0

---

Table 9: Verb-aware prompt: Segments sentences based only on individual actions without an event-level definition.

---

Verb-aware Prompt

---

**Please segment a single input sentence into multiple sentences that each represent a distinct event, following the rules below:**

- Each segmented sentence must start with the subject used in the original sentence (e.g., "a person", "a man", etc.).

- Do not remove or simplify any adverbs, adjectives, or modifiers that appear in the original sentence — preserve them as much as possible.

- Parts separated by the # symbol must be included in each segmented sentence.

- Do not add any new sentences — only break down the given text as instructed.

- Do not include your thinking process or output reasoning. Only output the segmented sentences following the format.

- Even if there are grammatical errors in the sentence, proceed with the processing.

- If the input sentence contains multiple actions, the output must contain same number of actions as the input sentence.

> **Good Example 1 - Input**
>
> a man lifts something on his left and places it down on his right.#a/DET man/NOUN lift/VERB something/PRON on/ADP his/DET left/NOUN and/CCONJ place/VERB it/PRON down/ADP on/ADP his/DET right/NOUN#0.0#0.0

> **Good Example 1 - Output**
>
> a man lifts something on his left.#a/DET man/NOUN lift/VERB something/PRON on/ADP his/DET left/NOUN#0.0#0.0 a man places it down on his right.#a/DET man/NOUN place/VERB it/PRON down/ADP on/ADP his/DET right/NOUN#0.0#0.0

> **Good Example 2 - Input**
>
> a man kicks something with his left leg.#a/DET man/NOUN kick/VERB something/PRON with/ADP his/DET left/ADJ leg/NOUN#0.0#0.0

> **Good Example 2 - Output**
>
> a man kicks something with his left leg.#a/DET man/NOUN kick/VERB something/PRON with/ADP his/DET left/ADJ leg/NOUN#0.0#0.0

> **Good Example 3 - Input**
>
> A man picks up a heavy box and carries it across the room.#A/DET man/NOUN pick/VERB up/ADP a/DET heavy/ADJ box/NOUN and/CCONJ carry/VERB it/PRON across/ADP the/DET room/NOUN#0.0#0.0

> **Good Example 3 - Output**
>
> A man picks up a heavy box.#A/DET man/NOUN pick/VERB up/ADP a/DET heavy/ADJ box/NOUN#0.0#0.0 A man carries it across the room.#A/DET man/NOUN carry/VERB it/PRON across/ADP the/DET room/NOUN#0.0#0.0

> **Bad Example 1 - Input**
>
> someone is sprinting side to side#someone/PRON is/AUX sprint/VERB side/NOUN to/PART side/VERB#0.0#0.0

> **Bad Example 1 - Output**
>
> The input sentence cannot be segmented as requested because "side to side" is a single adverbial phrase modifying the verb "sprinting," and "side" is used as a noun and a prepositional phrase. The sentence only contains one action (sprinting).

> **Bad Example 2 - Input**
>
> A person jumps high and lands softly.#A/DET person/NOUN jump/VERB high/ADV and/CCONJ land/VERB softly/ADV#0.0#0.0

> **Bad Example 2 - Output**
>
> A person jumps high and lands softly.#A/DET person/NOUN jump/VERB high/ADV and/CCONJ land/VERB softly/ADV#0.0#0.0

> **Bad Example 3 - Input**
>
> a man runs quickly and turns left.#a/DET man/NOUN run/VERB quickly/ADV and/CCONJ turn/VERB left/ADV#0.0#0.0

> **Bad Example 3 - Output**
>
> a man runs quickly.#a/DET man/NOUN run/VERB quickly/ADV#0.0#0.0

## A.11 VALIDATING THE ACCURACY OF LLM-GENERATED EVENT LABELS.

To directly measure the reliability of the LLM-based event decomposition, we perform a human evaluation on 300 randomly sampled HumanML3D-E prompts. Three trained annotators independently assess whether each LLM-generated segment (1) is grammatically well-formed and (2) matches our event definition as a minimal, semantically self-contained action. A segmentation is accepted only when both criteria are satisfied. Under this protocol, the LLM achieves a 93.3% correctness rate, indicating that the event splits are sufficiently accurate for benchmark construction.

Table 10: Comparative results on KIT-ML-E and Motion-X-E against baselines. "Condition 2/3/4" denotes prompts with at least 2, 3, and 4 events, respectively.

| Datasets | Condition | Methods | R-Precision ↑ | | | FID ↓ | MM-Dist ↓ | MModality ↑ |
|---|---|---|---|---|---|---|---|---|
| | | | Top-1 ↑ | Top-2 ↑ | Top-3 ↑ | | | |
| KIT-ML-E | 2 | AttT2M (Zhong et al., 2023) | $0.320^{\pm.011}$ | $0.514^{\pm.010}$ | $0.640^{\pm.012}$ | $0.636^{\pm.067}$ | $3.568^{\pm.046}$ | $\mathbf{2.097}^{\pm.072}$ |
| | | MoMask (Guo et al., 2024) | $\underline{0.393}^{\pm.009}$ | $\underline{0.586}^{\pm.010}$ | $0.708^{\pm.010}$ | $\underline{0.380}^{\pm.026}$ | $3.054^{\pm.037}$ | $1.268^{\pm.038}$ |
| | | Light-T2M (Zeng et al., 2025) | $\mathbf{0.417}^{\pm.012}$ | $\mathbf{0.614}^{\pm.011}$ | $\mathbf{0.734}^{\pm.011}$ | $0.392^{\pm.018}$ | $\mathbf{2.907}^{\pm.028}$ | $\underline{1.337}^{\pm.050}$ |
| | | MoGenTS (Yuan et al., 2024) | $0.353^{\pm.008}$ | $0.581^{\pm.009}$ | $\underline{0.721}^{\pm.008}$ | $0.424^{\pm.025}$ | $3.100^{\pm.035}$ | $0.720^{\pm.034}$ |
| | | **Event-T2M (Ours)** | $0.368^{\pm.010}$ | $0.575^{\pm.008}$ | $0.707^{\pm.009}$ | $\mathbf{0.378}^{\pm.016}$ | $\underline{3.020}^{\pm.026}$ | $0.794^{\pm.045}$ |
| | 3 | AttT2M (Zhong et al., 2023) | $0.176^{\pm.010}$ | $0.299^{\pm.008}$ | $0.405^{\pm.010}$ | $1.795^{\pm.174}$ | $3.524^{\pm.054}$ | $\mathbf{2.053}^{\pm.052}$ |
| | | MoMask (Guo et al., 2024) | $\underline{0.242}^{\pm.012}$ | $0.374^{\pm.016}$ | $0.476^{\pm.020}$ | $0.991^{\pm.117}$ | $3.126^{\pm.076}$ | $1.201^{\pm.050}$ |
| | | Light-T2M (Zeng et al., 2025) | $\mathbf{0.287}^{\pm.019}$ | $\mathbf{0.448}^{\pm.019}$ | $\mathbf{0.554}^{\pm.019}$ | $0.700^{\pm.044}$ | $\underline{2.785}^{\pm.075}$ | $\underline{1.265}^{\pm.039}$ |
| | | MoGenTS (Yuan et al., 2024) | $0.197^{\pm.014}$ | $0.346^{\pm.013}$ | $0.470^{\pm.011}$ | $0.855^{\pm.060}$ | $3.058^{\pm.054}$ | $0.712^{\pm.026}$ |
| | | **Event-T2M (Ours)** | $0.241^{\pm.013}$ | $\underline{0.406}^{\pm.012}$ | $\underline{0.520}^{\pm.010}$ | $\mathbf{0.678}^{\pm.020}$ | $\mathbf{2.672}^{\pm.040}$ | $0.769^{\pm.023}$ |
| | 4 | AttT2M (Zhong et al., 2023) | $0.316^{\pm.032}$ | $0.528^{\pm.032}$ | $0.688^{\pm.034}$ | $9.190^{\pm1.051}$ | $3.956^{\pm.191}$ | $\mathbf{2.527}^{\pm.131}$ |
| | | MoMask (Guo et al., 2024) | $\mathbf{0.434}^{\pm.027}$ | $\mathbf{0.653}^{\pm.036}$ | $0.713^{\pm.032}$ | $4.565^{\pm.507}$ | $3.801^{\pm.214}$ | $1.292^{\pm.061}$ |
| | | Light-T2M (Zeng et al., 2025) | $\underline{0.416}^{\pm.025}$ | $\mathbf{0.653}^{\pm.051}$ | $\mathbf{0.734}^{\pm.046}$ | $\underline{3.639}^{\pm.507}$ | $3.459^{\pm.103}$ | $\underline{1.568}^{\pm.072}$ |
| | | MoGenTS (Yuan et al., 2024) | $0.338^{\pm.016}$ | $0.597^{\pm.029}$ | $0.697^{\pm.032}$ | $3.894^{\pm.278}$ | $\mathbf{3.357}^{\pm.051}$ | $0.866^{\pm.047}$ |
| | | **Event-T2M (Ours)** | $0.350^{\pm.018}$ | $\underline{0.641}^{\pm.030}$ | $\underline{0.716}^{\pm.025}$ | $\mathbf{3.429}^{\pm.276}$ | $3.661^{\pm.072}$ | $0.990^{\pm.030}$ |
| Motion-X-E | 2 | AttT2M (Zhong et al., 2023) | $0.425^{\pm.005}$ | $0.628^{\pm.006}$ | $0.741^{\pm.005}$ | $0.350^{\pm.021}$ | $3.728^{\pm.027}$ | $\mathbf{2.289}^{\pm.062}$ |
| | | MoMask (Guo et al., 2024) | $0.429^{\pm.006}$ | $0.633^{\pm.004}$ | $0.746^{\pm.003}$ | $0.362^{\pm.017}$ | $3.698^{\pm.014}$ | $1.492^{\pm.044}$ |
| | | Light-T2M (Zeng et al., 2025) | $\underline{0.441}^{\pm.003}$ | $0.643^{\pm.003}$ | $0.749^{\pm.003}$ | $0.174^{\pm.010}$ | $\underline{3.674}^{\pm.010}$ | $\underline{1.657}^{\pm.053}$ |
| | | MoGenTS (Yuan et al., 2024) | $0.432^{\pm.005}$ | $\underline{0.645}^{\pm.003}$ | $\underline{0.757}^{\pm.004}$ | $\underline{0.116}^{\pm.008}$ | $3.693^{\pm.017}$ | $0.833^{\pm.030}$ |
| | | **Event-T2M (Ours)** | $\mathbf{0.524}^{\pm.008}$ | $\mathbf{0.728}^{\pm.006}$ | $\mathbf{0.825}^{\pm.004}$ | $\mathbf{0.111}^{\pm.003}$ | $\mathbf{2.984}^{\pm.011}$ | $0.902^{\pm.048}$ |
| | 3 | AttT2M (Zhong et al., 2023) | $0.347^{\pm.006}$ | $0.541^{\pm.006}$ | $0.658^{\pm.005}$ | $0.750^{\pm.042}$ | $4.071^{\pm.023}$ | $\mathbf{2.512}^{\pm.076}$ |
| | | MoMask (Guo et al., 2024) | $0.359^{\pm.008}$ | $\underline{0.559}^{\pm.006}$ | $\underline{0.681}^{\pm.004}$ | $0.695^{\pm.027}$ | $\underline{3.867}^{\pm.020}$ | $1.620^{\pm.052}$ |
| | | Light-T2M (Zeng et al., 2025) | $\underline{0.369}^{\pm.006}$ | $0.558^{\pm.008}$ | $0.672^{\pm.006}$ | $0.364^{\pm.021}$ | $3.991^{\pm.019}$ | $\underline{1.736}^{\pm.057}$ |
| | | MoGenTS (Yuan et al., 2024) | $0.366^{\pm.006}$ | $0.557^{\pm.006}$ | $0.672^{\pm.006}$ | $\underline{0.169}^{\pm.012}$ | $3.896^{\pm.015}$ | $0.843^{\pm.025}$ |
| | | **Event-T2M (Ours)** | $\mathbf{0.530}^{\pm.007}$ | $\mathbf{0.729}^{\pm.006}$ | $\mathbf{0.825}^{\pm.005}$ | $\mathbf{0.140}^{\pm.012}$ | $\mathbf{2.981}^{\pm.025}$ | $0.949^{\pm.058}$ |
| | 4 | AttT2M (Zhong et al., 2023) | $0.311^{\pm.010}$ | $0.493^{\pm.014}$ | $0.610^{\pm.016}$ | $1.456^{\pm.131}$ | $4.224^{\pm.076}$ | $\mathbf{2.782}^{\pm.078}$ |
| | | MoMask (Guo et al., 2024) | $0.300^{\pm.011}$ | $0.504^{\pm.016}$ | $0.625^{\pm.017}$ | $1.224^{\pm.095}$ | $\underline{4.066}^{\pm.054}$ | $1.770^{\pm.046}$ |
| | | Light-T2M (Zeng et al., 2025) | $0.310^{\pm.012}$ | $0.495^{\pm.011}$ | $0.616^{\pm.013}$ | $1.082^{\pm.064}$ | $4.224^{\pm.056}$ | $\underline{1.915}^{\pm.068}$ |
| | | MoGenTS (Yuan et al., 2024) | $\underline{0.323}^{\pm.012}$ | $\underline{0.516}^{\pm.012}$ | $\underline{0.629}^{\pm.009}$ | $\underline{0.744}^{\pm.058}$ | $4.118^{\pm.039}$ | $0.953^{\pm.034}$ |
| | | **Event-T2M (Ours)** | $\mathbf{0.362}^{\pm.008}$ | $\mathbf{0.567}^{\pm.009}$ | $\mathbf{0.697}^{\pm.006}$ | $\mathbf{0.431}^{\pm.005}$ | $\mathbf{3.959}^{\pm.017}$ | $0.953^{\pm.049}$ |

## A.12 EVALUATION ON MORE DATASETS

We constructed event-level benchmarks for KIT-ML and Motion-X and evaluated all baseline models under identical conditions (Table 10 and 11). Because KIT-ML consists largely of simpler motions, the number of samples containing $\geq 4$ events is extremely limited; thus, we used a batch size of 16 for evaluation under that specific condition. Across both benchmarks, Event-T2M demonstrated consistent and competitive performance, indicating that its effectiveness is not restricted to any particular dataset or benchmark setting.

## A.13 COMPARISON WITH RETRIEVAL-BASED MOTION SYNTHESIS

We additionally compare Event-T2M with a representative retrieval-augmented baseline, ReMoDiffuse (Zhang et al., 2023b), which improves motion quality by retrieving motion conditioned on the input text and refining them with a diffusion model. Quantitative results on HumanML3D-E are reported in Table 12. Across all conditions, Event-T2M achieves better text–motion alignment and lower FID than ReMoDiffuse, and the margin becomes larger as the number of events in the prompt increases (Conditions 2, 3, and 4).

We attribute this trend to a fundamental limitation of retrieval-based pipelines under compositional complexity. ReMoDiffuse relies on matching the prompt to existing motions, which is effective when the description corresponds to a single, relatively simple action, but does not provide an explicit mechanism to align multiple ordered semantic units to distinct temporal segments. As a result, for long multi-event prompts, the retrieved motions often cover only part of the description or fail to respect the full temporal structure.

In contrast, Event-T2M explicitly introduces event boundaries in the text domain and conditions the denoising process on event tokens that are intended to control temporally extended motion segments. This event-level conditioning allows the model to synthesize novel combinations of sub-actions rather than relying solely on recombining existing trajectories. Empirically, this design leads to

Table 11: Comparative results on HumanML3D-E, KIT-ML-E and Motion-X-E against MARDM baselines. "Condition 2/3/4" denotes prompts with at least 2, 3, and 4 events, respectively.

| Datasets | Condition | Methods | R-Precision ↑ | | | FID ↓ | MM-Dist ↓ | MModality ↑ | CLIP-score ↑ |
|---|---|---|---|---|---|---|---|---|---|
| | | | Top-1 ↑ | Top-2 ↑ | Top-3 ↑ | | | | |
| HumanML3D-E | 2 | MARDM-DDPM (Meng et al., 2024) | $0.464^{\pm.005}$ | $0.658^{\pm.004}$ | $0.762^{\pm.003}$ | $0.157^{\pm.007}$ | $3.465^{\pm.015}$ | $\mathbf{2.331}^{\pm.079}$ | $0.621^{\pm.001}$ |
| | | MARDM-SiT (Meng et al., 2024) | $0.479^{\pm.004}$ | $0.671^{\pm.004}$ | $0.771^{\pm.003}$ | $0.171^{\pm.009}$ | $3.404^{\pm.011}$ | $2.296^{\pm.093}$ | $0.632^{\pm.001}$ |
| | | **Event-T2M (Ours)** | $\mathbf{0.535}^{\pm.003}$ | $\mathbf{0.683}^{\pm.002}$ | $\mathbf{0.782}^{\pm.003}$ | $\mathbf{0.116}^{\pm.002}$ | $\mathbf{3.013}^{\pm.005}$ | $1.034^{\pm.039}$ | $\mathbf{0.663}^{\pm.001}$ |
| | 3 | MARDM-DDPM (Meng et al., 2024) | $0.433^{\pm.005}$ | $0.621^{\pm.004}$ | $0.731^{\pm.004}$ | $0.301^{\pm.015}$ | $3.590^{\pm.019}$ | $2.461^{\pm.069}$ | $0.616^{\pm.002}$ |
| | | MARDM-SiT (Meng et al., 2024) | $0.440^{\pm.006}$ | $0.632^{\pm.003}$ | $0.733^{\pm.004}$ | $0.327^{\pm.018}$ | $3.544^{\pm.017}$ | $\mathbf{2.466}^{\pm.079}$ | $0.625^{\pm.002}$ |
| | | **Event-T2M (Ours)** | $\mathbf{0.508}^{\pm.004}$ | $\mathbf{0.708}^{\pm.003}$ | $\mathbf{0.806}^{\pm.004}$ | $\mathbf{0.131}^{\pm.004}$ | $\mathbf{3.045}^{\pm.008}$ | $1.082^{\pm.028}$ | $\mathbf{0.663}^{\pm.001}$ |
| | 4 | MARDM-DDPM (Meng et al., 2024) | $0.397^{\pm.013}$ | $0.585^{\pm.011}$ | $0.698^{\pm.011}$ | $0.643^{\pm0.063}$ | $3.697^{\pm.052}$ | $\mathbf{2.507}^{\pm.068}$ | $0.613^{\pm.004}$ |
| | | MARDM-SiT (Meng et al., 2024) | $0.420^{\pm.010}$ | $0.608^{\pm.011}$ | $0.707^{\pm.013}$ | $0.719^{\pm.056}$ | $3.676^{\pm.050}$ | $2.506^{\pm.072}$ | $0.621^{\pm.004}$ |
| | | **Event-T2M (Ours)** | $\mathbf{0.463}^{\pm.006}$ | $\mathbf{0.663}^{\pm.007}$ | $\mathbf{0.781}^{\pm.003}$ | $\mathbf{0.259}^{\pm.012}$ | $\mathbf{3.281}^{\pm.015}$ | $1.137^{\pm.085}$ | $\mathbf{0.659}^{\pm.003}$ |
| KIT-ML-E | 2 | MARDM-DDPM (Meng et al., 2024) | $\mathbf{0.390}^{\pm.012}$ | $\mathbf{0.578}^{\pm.012}$ | $0.685^{\pm.010}$ | $0.670^{\pm.049}$ | $3.729^{\pm.049}$ | $\mathbf{2.494}^{\pm.068}$ | $0.611^{\pm.004}$ |
| | | MARDM-SiT (Meng et al., 2024) | $0.332^{\pm.010}$ | $0.546^{\pm.009}$ | $\mathbf{0.688}^{\pm.010}$ | $\mathbf{0.442}^{\pm.034}$ | $3.555^{\pm.033}$ | $1.595^{\pm.075}$ | $\mathbf{0.674}^{\pm.002}$ |
| | | **Event-T2M (Ours)** | $0.346^{\pm.010}$ | $0.568^{\pm.009}$ | $0.621^{\pm.036}$ | $0.621^{\pm.036}$ | $\mathbf{3.434}^{\pm.036}$ | $0.963^{\pm.038}$ | $0.671^{\pm.001}$ |
| | 3 | MARDM-DDPM (Meng et al., 2024) | $0.202^{\pm.013}$ | $0.345^{\pm.015}$ | $0.443^{\pm.015}$ | $3.547^{\pm.722}$ | $3.987^{\pm.135}$ | $\mathbf{2.297}^{\pm.096}$ | $0.598^{\pm.009}$ |
| | | MARDM-SiT (Meng et al., 2024) | $0.205^{\pm.012}$ | $0.352^{\pm.014}$ | $0.454^{\pm.013}$ | $1.529^{\pm.294}$ | $3.561^{\pm.069}$ | $1.693^{\pm.063}$ | $0.650^{\pm.007}$ |
| | | **Event-T2M (Ours)** | $\mathbf{0.241}^{\pm.013}$ | $\mathbf{0.405}^{\pm.013}$ | $\mathbf{0.530}^{\pm.010}$ | $\mathbf{1.457}^{\pm.120}$ | $\mathbf{3.075}^{\pm.045}$ | $1.122^{\pm.035}$ | $\mathbf{0.681}^{\pm.002}$ |
| | 4 | MARDM-DDPM (Meng et al., 2024) | $0.341^{\pm.025}$ | $0.575^{\pm.024}$ | $0.669^{\pm.025}$ | $10.378^{\pm0.891}$ | $4.880^{\pm.193}$ | $2.038^{\pm.103}$ | $0.567^{\pm.018}$ |
| | | MARDM-SiT (Meng et al., 2024) | $\mathbf{0.359}^{\pm.023}$ | $\mathbf{0.584}^{\pm.030}$ | $\mathbf{0.700}^{\pm.025}$ | $7.942^{\pm.806}$ | $4.819^{\pm.202}$ | $\mathbf{2.285}^{\pm.146}$ | $0.515^{\pm.017}$ |
| | | **Event-T2M (Ours)** | $0.346^{\pm.023}$ | $0.571^{\pm.023}$ | $0.687^{\pm.021}$ | $\mathbf{3.102}^{\pm.580}$ | $\mathbf{3.733}^{\pm.151}$ | $1.051^{\pm.038}$ | $\mathbf{0.637}^{\pm.006}$ |
| Motion-X-E | 2 | MARDM-DDPM (Meng et al., 2024) | $0.383^{\pm.005}$ | $0.583^{\pm.004}$ | $0.703^{\pm.004}$ | $0.184^{\pm.009}$ | $3.926^{\pm.014}$ | $\mathbf{2.189}^{\pm.052}$ | $0.623^{\pm.001}$ |
| | | MARDM-SiT (Meng et al., 2024) | $0.397^{\pm.004}$ | $0.598^{\pm.003}$ | $0.715^{\pm.004}$ | $0.171^{\pm.009}$ | $3.844^{\pm.013}$ | $2.130^{\pm.052}$ | $0.633^{\pm.001}$ |
| | | **Event-T2M (Ours)** | $\mathbf{0.538}^{\pm.002}$ | $\mathbf{0.735}^{\pm.002}$ | $\mathbf{0.826}^{\pm.001}$ | $\mathbf{0.112}^{\pm.003}$ | $\mathbf{3.009}^{\pm.006}$ | $1.047^{\pm.054}$ | $\mathbf{0.664}^{\pm.001}$ |
| | 3 | MARDM-DDPM (Meng et al., 2024) | $0.349^{\pm.006}$ | $0.540^{\pm.008}$ | $0.660^{\pm.007}$ | $0.302^{\pm.015}$ | $3.824^{\pm.025}$ | $\mathbf{2.176}^{\pm.061}$ | $0.603^{\pm.003}$ |
| | | MARDM-SiT (Meng et al., 2024) | $0.356^{\pm.005}$ | $0.551^{\pm.007}$ | $0.674^{\pm.006}$ | $0.266^{\pm.017}$ | $3.727^{\pm.017}$ | $2.157^{\pm.051}$ | $0.617^{\pm.002}$ |
| | | **Event-T2M (Ours)** | $\mathbf{0.507}^{\pm.006}$ | $\mathbf{0.705}^{\pm.004}$ | $\mathbf{0.803}^{\pm.003}$ | $\mathbf{0.124}^{\pm.006}$ | $\mathbf{3.052}^{\pm.008}$ | $1.061^{\pm.047}$ | $\mathbf{0.663}^{\pm.001}$ |
| | 4 | MARDM-DDPM (Meng et al., 2024) | $0.322^{\pm.015}$ | $0.534^{\pm.011}$ | $0.680^{\pm.014}$ | $0.925^{\pm.079}$ | $3.672^{\pm.060}$ | $\mathbf{2.154}^{\pm.066}$ | $0.556^{\pm.004}$ |
| | | MARDM-SiT (Meng et al., 2024) | $0.360^{\pm.014}$ | $0.555^{\pm.015}$ | $0.687^{\pm.011}$ | $0.972^{\pm.088}$ | $3.640^{\pm.045}$ | $2.150^{\pm.060}$ | $0.571^{\pm.005}$ |
| | | **Event-T2M (Ours)** | $\mathbf{0.452}^{\pm.007}$ | $\mathbf{0.660}^{\pm.006}$ | $\mathbf{0.776}^{\pm.007}$ | $\mathbf{0.250}^{\pm.009}$ | $\mathbf{3.292}^{\pm.017}$ | $1.106^{\pm.029}$ | $\mathbf{0.659}^{\pm.001}$ |

Table 12: Comparative results on HumanML3D-E against retrieval-based baselines. "Condition 2/3/4" denotes prompts with at least 2, 3, and 4 events, respectively.

| Condition | Methods | R-Precision ↑ | | | FID ↓ | MM-Dist ↓ | MModality ↑ |
|---|---|---|---|---|---|---|---|
| | | Top-1 ↑ | Top-2 ↑ | Top-3 ↑ | | | |
| 2 | ReMoDiffuse (Zhang et al., 2023b) | $0.475^{\pm.003}$ | $0.657^{\pm.004}$ | $0.755^{\pm.003}$ | $0.151^{\pm.009}$ | $3.127^{\pm.016}$ | $\mathbf{2.937}^{\pm.067}$ |
| | **Event-T2M (Ours)** | $\mathbf{0.536}^{\pm.002}$ | $\mathbf{0.732}^{\pm.002}$ | $\mathbf{0.824}^{\pm.002}$ | $\mathbf{0.079}^{\pm.003}$ | $\mathbf{2.836}^{\pm.006}$ | $0.976^{\pm.043}$ |
| 3 | ReMoDiffuse (Zhang et al., 2023b) | $0.444^{\pm.005}$ | $0.630^{\pm.004}$ | $0.732^{\pm.004}$ | $0.292^{\pm.016}$ | $3.174^{\pm.024}$ | $\mathbf{2.851}^{\pm.087}$ |
| | **Event-T2M (Ours)** | $\mathbf{0.487}^{\pm.005}$ | $\mathbf{0.687}^{\pm.004}$ | $\mathbf{0.790}^{\pm.004}$ | $\mathbf{0.137}^{\pm.003}$ | $\mathbf{2.928}^{\pm.010}$ | $1.010^{\pm.029}$ |
| 4 | ReMoDiffuse (Zhang et al., 2023b) | $0.392^{\pm.011}$ | $0.572^{\pm.009}$ | $0.674^{\pm.009}$ | $0.583^{\pm.039}$ | $3.244^{\pm.046}$ | $\mathbf{3.106}^{\pm.116}$ |
| | **Event-T2M (Ours)** | $\mathbf{0.466}^{\pm.008}$ | $\mathbf{0.660}^{\pm.008}$ | $\mathbf{0.767}^{\pm.007}$ | $\mathbf{0.265}^{\pm.007}$ | $\mathbf{3.063}^{\pm.015}$ | $1.039^{\pm.028}$ |

better preservation of event ordering and fewer omissions on HumanML3D-E, particularly in the high-complexity regime where retrieval-based approaches struggle.

## A.14 VERB-AWARE VS. EVENT-AWARE CONDITIONING

To directly assess whether event-aware conditioning offers advantages over verb-aware or hybrid (verb + event) formulations, we conducted additional experiments using the verb-aware prompts introduced in Table 9 of the supplementary material. Based on these prompts, we re-preprocessed HumanML3D-E and retrained (i) a verb-aware variant that conditions solely on verb-level units, (ii) a hybrid variant that combines verb-level units with global text features, and (iii) our event-aware model built from the event-aware prompts in Table 8. Quantitative results across different event-count subsets are reported in Table 13.

As summarized in Table 13, the event-aware model consistently achieves higher R-Precision and exhibits more stable FID than both the verb-aware and hybrid variants across all complexity levels. Verb-aware conditioning treats individual verbs as the primary semantic units, which captures instantaneous actions but does not encode how they unfold as temporally coherent segments. In contrast, event-aware conditioning operates on clauses that bundle the action together with its arguments, affected body parts, and temporal scope, providing a more suitable alignment target for motion segments.

Table 13: Comparative results on HumanML3D-E against verb-aware and event-aware conditioning. "Condition 2/3/4" denotes prompts with at least 2, 3, and 4 events, respectively.

| Condition | Methods | R-Precision ↑ | | | FID ↓ | MM-Dist ↓ | MModality ↑ |
| | | Top-1 ↑ | Top-2 ↑ | Top-3 ↑ | | | |
|---|---|---|---|---|---|---|---|
| baseline | Verb-aware conditioning | $0.548^{\pm.002}$ | $0.738^{\pm.002}$ | $0.830^{\pm.001}$ | $0.085^{\pm.004}$ | $2.772^{\pm.008}$ | $\mathbf{1.164}^{\pm.042}$ |
| | **Event-aware conditioning** | $\mathbf{0.562}^{\pm.002}$ | $\mathbf{0.754}^{\pm.003}$ | $\mathbf{0.842}^{\pm.002}$ | $\mathbf{0.056}^{\pm.002}$ | $\mathbf{2.711}^{\pm.005}$ | $0.949^{\pm.026}$ |
| 2 | Verb-aware conditioning | $0.518^{\pm.002}$ | $0.712^{\pm.003}$ | $0.810^{\pm.002}$ | $0.128^{\pm.003}$ | $2.913^{\pm.006}$ | $\mathbf{1.309}^{\pm.039}$ |
| | **Event-aware conditioning** | $\mathbf{0.536}^{\pm.002}$ | $\mathbf{0.732}^{\pm.002}$ | $\mathbf{0.824}^{\pm.002}$ | $\mathbf{0.079}^{\pm.003}$ | $\mathbf{2.836}^{\pm.006}$ | $0.976^{\pm.043}$ |
| 3 | Verb-aware conditioning | $0.479^{\pm.005}$ | $0.681^{\pm.006}$ | $0.780^{\pm.005}$ | $0.193^{\pm.006}$ | $2.990^{\pm.011}$ | $\mathbf{1.292}^{\pm.044}$ |
| | **Event-aware conditioning** | $\mathbf{0.487}^{\pm.005}$ | $\mathbf{0.687}^{\pm.004}$ | $\mathbf{0.790}^{\pm.004}$ | $\mathbf{0.137}^{\pm.003}$ | $\mathbf{2.928}^{\pm.010}$ | $1.010^{\pm.029}$ |
| 4 | Verb-aware conditioning | $\mathbf{0.466}^{\pm.009}$ | $0.654^{\pm.010}$ | $0.748^{\pm.008}$ | $0.347^{\pm.011}$ | $3.102^{\pm.026}$ | $\mathbf{1.364}^{\pm.044}$ |
| | **Event-aware conditioning** | $\mathbf{0.466}^{\pm.008}$ | $\mathbf{0.660}^{\pm.008}$ | $\mathbf{0.767}^{\pm.007}$ | $\mathbf{0.265}^{\pm.007}$ | $\mathbf{3.063}^{\pm.015}$ | $1.039^{\pm.028}$ |

Table 14: Comparative results on HumanML3D-E ($\geq 4$ events) with human-annotated, LLM-free test set.

| Annotator | Methods | R-Precision ↑ | | | FID ↓ | MM-Dist ↓ | MModality ↑ |
| | | Top-1 ↑ | Top-2 ↑ | Top-3 ↑ | | | |
|---|---|---|---|---|---|---|---|
| Human 1 | AttT2M (Zhong et al., 2023) | $0.410^{\pm.012}$ | $0.584^{\pm.015}$ | $0.687^{\pm.010}$ | $1.054^{\pm.004}$ | $3.464^{\pm.043}$ | $1.273^{\pm.506}$ |
| | MoMask (Guo et al., 2024) | $0.443^{\pm.013}$ | $0.631^{\pm.013}$ | $0.733^{\pm.011}$ | $0.413^{\pm.030}$ | $3.205^{\pm.041}$ | $\mathbf{1.337}^{\pm.045}$ |
| | **Event-T2M (Ours)** | $\mathbf{0.459}^{\pm.009}$ | $\mathbf{0.650}^{\pm.009}$ | $\mathbf{0.762}^{\pm.008}$ | $\mathbf{0.297}^{\pm.008}$ | $\mathbf{3.036}^{\pm.022}$ | $1.040^{\pm.029}$ |
| Human 2 | AttT2M (Zhong et al., 2023) | $0.408^{\pm.019}$ | $0.595^{\pm.024}$ | $0.696^{\pm.020}$ | $1.052^{\pm.169}$ | $3.495^{\pm.070}$ | $\mathbf{1.656}^{\pm.918}$ |
| | MoMask (Guo et al., 2024) | $0.435^{\pm.012}$ | $0.625^{\pm.013}$ | $0.729^{\pm.012}$ | $0.420^{\pm.025}$ | $3.238^{\pm.046}$ | $1.349^{\pm.042}$ |
| | **Event-T2M (Ours)** | $\mathbf{0.457}^{\pm.010}$ | $\mathbf{0.667}^{\pm.010}$ | $\mathbf{0.766}^{\pm.009}$ | $\mathbf{0.281}^{\pm.007}$ | $\mathbf{3.044}^{\pm.021}$ | $1.061^{\pm.031}$ |
| Human 3 | AttT2M (Zhong et al., 2023) | $0.393^{\pm.012}$ | $0.674^{\pm.013}$ | $0.679^{\pm.014}$ | $1.078^{\pm.087}$ | $3.513^{\pm.057}$ | $\mathbf{1.431}^{\pm.354}$ |
| | MoMask (Guo et al., 2024) | $0.441^{\pm.011}$ | $0.639^{\pm.011}$ | $0.746^{\pm.011}$ | $0.437^{\pm.029}$ | $3.187^{\pm.036}$ | $1.389^{\pm.036}$ |
| | **Event-T2M (Ours)** | $\mathbf{0.494}^{\pm.007}$ | $\mathbf{0.685}^{\pm.007}$ | $\mathbf{0.781}^{\pm.008}$ | $\mathbf{0.276}^{\pm.008}$ | $\mathbf{3.005}^{\pm.017}$ | $1.048^{\pm.040}$ |
| LLM | AttT2M (Zhong et al., 2023) | $0.407^{\pm.013}$ | $0.581^{\pm.010}$ | $0.688^{\pm.010}$ | $1.077^{\pm.104}$ | $3.455^{\pm.041}$ | $\mathbf{2.049}^{\pm.099}$ |
| | MoMask (Guo et al., 2024) | $0.441^{\pm.013}$ | $0.633^{\pm.014}$ | $0.734^{\pm.013}$ | $0.418^{\pm.030}$ | $3.205^{\pm.042}$ | $1.334^{\pm.046}$ |
| | **Event-T2M (Ours)** | $\mathbf{0.466}^{\pm.008}$ | $\mathbf{0.660}^{\pm.008}$ | $\mathbf{0.767}^{\pm.007}$ | $\mathbf{0.265}^{\pm.007}$ | $\mathbf{3.063}^{\pm.015}$ | $1.039^{\pm.028}$ |

These differences become most pronounced in the 4-event setting, where long-range sequential dependencies are strongest. In this regime, the verb-aware model frequently merges or reorders sub-actions, and the hybrid model only partially alleviates these issues by mixing verb-level information with global context, but still lacks explicit temporal boundaries. The event-aware model, on the other hand, benefits from conditioning on temporally extended event tokens that map more directly to contiguous motion segments, which leads to better preservation of event ordering and fewer omissions on complex multi-event prompts. Overall, these findings suggest that the gains of event-aware conditioning come not from additional supervision, but from providing the diffusion model with semantically and temporally grounded units that better match the structure of human motion.

A.15 EVALUATION ON A HUMAN-ANNOTATED, LLM-FREE EVENT TEST SET

To test Event-T2M independently of the LLM-based event decomposition, we constructed an additional complex-motion test subset with fully human-segmented events. Three trained annotators manually segmented all prompts with 4 events according to our event definition. For these long and structured descriptions, annotators produced highly consistent segmentations, and the resulting subsets substantially overlapped with the prompts selected by the LLM. We then evaluated Event-T2M and all baselines on each of the three human-segmented splits. As reported in Table 14, the performance trends closely match those observed on the original LLM-based HumanML3D-E split: Event-T2M maintains a clear advantage over all baselines across annotators in terms of both text–motion alignment and FID.

To directly address the concern that the baselines' poorer performance might be an artifact of the LLM pipeline, we further analyzed AttT2M and MoMask on the same three human-annotated subsets. For each method, we compare results on (i) the original LLM-based split and (ii) the three

Table 15: Comparative results on HumanML3D-E against baselines with TMR encoder setting. "Condition 2/3/4" denotes prompts with at least 2, 3, and 4 events, respectively.

| Condition | Methods | R-Precision ↑ | | | FID ↓ | MM-Dist ↓ | MModality ↑ |
|---|---|---|---|---|---|---|---|
| | | Top-1 ↑ | Top-2 ↑ | Top-3 ↑ | | | |
| baseline | AttT2M (Zhong et al., 2023) | $0.518^{\pm.006}$ | $0.707^{\pm.005}$ | $0.799^{\pm.006}$ | $0.146^{\pm.015}$ | $2.957^{\pm.029}$ | $\mathbf{1.594}^{\pm.171}$ |
| | MoMask (Guo et al., 2024) | $0.487^{\pm.002}$ | $0.684^{\pm.002}$ | $0.783^{\pm.003}$ | $0.284^{\pm.007}$ | $3.116^{\pm.008}$ | $1.158^{\pm.040}$ |
| | MoMask (Guo et al., 2024) (k/v) | $0.494^{\pm.003}$ | $0.683^{\pm.002}$ | $0.781^{\pm.002}$ | $0.240^{\pm.008}$ | $3.113^{\pm.008}$ | $\underline{1.214}^{\pm.045}$ |
| | Light-T2M (Zeng et al., 2025) | $\underline{0.554}^{\pm.003}$ | $\underline{0.746}^{\pm.002}$ | $\underline{0.836}^{\pm.002}$ | $\mathbf{0.053}^{\pm.002}$ | $2.750^{\pm.008}$ | $0.970^{\pm.049}$ |
| | **Event-T2M (Ours)** | $\mathbf{0.562}^{\pm.002}$ | $\mathbf{0.754}^{\pm.003}$ | $\mathbf{0.842}^{\pm.002}$ | $\underline{0.056}^{\pm.002}$ | $2.711^{\pm.005}$ | $0.949^{\pm.026}$ |
| 2 | AttT2M (Zhong et al., 2023) | $0.500^{\pm.014}$ | $0.681^{\pm.004}$ | $0.777^{\pm.005}$ | $0.199^{\pm.008}$ | $3.089^{\pm.015}$ | $\mathbf{1.698}^{\pm.082}$ |
| | MoMask (Guo et al., 2024) | $0.470^{\pm.002}$ | $0.665^{\pm.002}$ | $0.769^{\pm.003}$ | $0.385^{\pm.011}$ | $3.194^{\pm.011}$ | $1.174^{\pm.041}$ |
| | MoMask (Guo et al., 2024) (k/v) | $0.479^{\pm.004}$ | $0.668^{\pm.003}$ | $0.766^{\pm.002}$ | $0.291^{\pm.010}$ | $3.196^{\pm.010}$ | $\underline{1.217}^{\pm.048}$ |
| | Light-T2M (Zeng et al., 2025) | $\underline{0.527}^{\pm.003}$ | $\underline{0.722}^{\pm.003}$ | $\underline{0.815}^{\pm.002}$ | $\underline{0.087}^{\pm.002}$ | $2.885^{\pm.006}$ | $0.984^{\pm.032}$ |
| | **Event-T2M (Ours)** | $\mathbf{0.536}^{\pm.002}$ | $\mathbf{0.732}^{\pm.002}$ | $\mathbf{0.824}^{\pm.002}$ | $\mathbf{0.079}^{\pm.003}$ | $2.836^{\pm.006}$ | $0.976^{\pm.043}$ |
| 3 | AttT2M (Zhong et al., 2023) | $0.445^{\pm.017}$ | $0.631^{\pm.011}$ | $0.732^{\pm.011}$ | $0.474^{\pm.053}$ | $3.251^{\pm.024}$ | $\mathbf{1.798}^{\pm.117}$ |
| | MoMask (Guo et al., 2024) | $0.430^{\pm.006}$ | $0.622^{\pm.007}$ | $0.733^{\pm.006}$ | $0.544^{\pm.017}$ | $3.289^{\pm.020}$ | $1.190^{\pm.038}$ |
| | MoMask (Guo et al., 2024) (k/v) | $0.445^{\pm.004}$ | $0.629^{\pm.006}$ | $0.731^{\pm.004}$ | $0.364^{\pm.017}$ | $3.297^{\pm.020}$ | $\underline{1.228}^{\pm.045}$ |
| | Light-T2M (Zeng et al., 2025) | $\underline{0.487}^{\pm.006}$ | $\underline{0.680}^{\pm.006}$ | $\underline{0.780}^{\pm.004}$ | $\underline{0.139}^{\pm.004}$ | $2.987^{\pm.010}$ | $1.005^{\pm.033}$ |
| | **Event-T2M (Ours)** | $\mathbf{0.487}^{\pm.005}$ | $\mathbf{0.687}^{\pm.004}$ | $\mathbf{0.790}^{\pm.004}$ | $\mathbf{0.137}^{\pm.003}$ | $2.928^{\pm.010}$ | $1.010^{\pm.029}$ |
| 4 | AttT2M (Zhong et al., 2023) | $0.424^{\pm.017}$ | $0.598^{\pm.025}$ | $0.701^{\pm.018}$ | $0.789^{\pm.046}$ | $3.335^{\pm.107}$ | $\mathbf{1.647}^{\pm.409}$ |
| | MoMask (Guo et al., 2024) | $0.417^{\pm.012}$ | $0.601^{\pm.008}$ | $0.708^{\pm.010}$ | $0.821^{\pm.056}$ | $3.392^{\pm.037}$ | $1.230^{\pm.040}$ |
| | MoMask (Guo et al., 2024) (k/v) | $0.413^{\pm.011}$ | $0.590^{\pm.011}$ | $0.703^{\pm.011}$ | $0.682^{\pm.054}$ | $3.479^{\pm.042}$ | $\underline{1.251}^{\pm.051}$ |
| | Light-T2M (Zeng et al., 2025) | $\underline{0.436}^{\pm.009}$ | $\underline{0.609}^{\pm.008}$ | $\underline{0.711}^{\pm.007}$ | $\underline{0.361}^{\pm.009}$ | $\underline{3.319}^{\pm.017}$ | $1.029^{\pm.028}$ |
| | **Event-T2M (Ours)** | $\mathbf{0.466}^{\pm.008}$ | $\mathbf{0.660}^{\pm.008}$ | $\mathbf{0.767}^{\pm.007}$ | $\mathbf{0.265}^{\pm.007}$ | $3.063^{\pm.015}$ | $1.039^{\pm.028}$ |

human-segmented splits (Table 14). Across all metrics, the differences between the LLM-based and human-annotated results are very small. For AttT2M, Top-k R-Precision and FID on the human-segmented splits remain within a narrow margin of the LLM-based scores, without any consistent upward shift. MoMask exhibits the same pattern: Top-k retrieval, FID, MM-Dist, and multimodality on the human-annotated subsets fluctuate only slightly around the LLM-based values, sometimes marginally higher and sometimes marginally lower.

These observations do not support the hypothesis that the baselines' weaker performance on HumanML3D-E is caused by a mismatch with the LLM-based segmentation pipeline. Instead, they indicate that AttT2M and MoMask behave similarly on both LLM-segmented and human-segmented complex prompts, while Event-T2M retains a consistent advantage in all cases (Table 14).

## A.16 EVALUATION IN MODEL CONFIGURATIONS EMPLOYING THE TMR ENCODER

To ensure a fair comparison, we conducted additional experiments by replacing the CLIP text encoder with TMR in baseline models. Especially, we conducted an experiment on MoMask that uses TMR's word-level tokens as key/value in cross-attention (indicated as MoMask (k/v)).

Table 15 shows that replacing CLIP with TMR does not act as a uniformly strong boost across all baselines. For AttT2M, using TMR tends to slightly improve text–motion alignment metrics, but at the same time leads to less favorable behavior in terms of distributional quality (FID) and motion diversity (MModality). MoMask is an even more extreme case: with CLIP it is very strong on distributional metrics, but when we drop in TMR with the same architecture, both alignment and FID move in a clearly worse direction. In contrast, Light-T2M benefits more consistently from TMR: alignment improves across the board and, especially on subsets with a larger number of events, FID also becomes better. In other words, we observe a heterogeneous pattern where gains and losses coexist and depend strongly on the underlying architecture.

The MoMask (k/v) experiment was designed to test whether injecting TMR word-level tokens more directly as key/value would change this picture. We find that in some conditions this variant outperforms the plain TMR version of MoMask in terms of alignment or FID, but compared to CLIP-based MoMask the overall distributional quality is still worse, and on the most event-complex subsets the performance gap is not fully closed. Thus, neither switching the encoder to TMR nor directly feeding TMR tokens into K/V makes MoMask adopt TMR as a clearly and consistently superior configuration over CLIP on HumanML3D-E.

Table 16: Ablations for LIMM and ATII. "Condition 2/3/4" denotes prompts with at least 2, 3, and 4 events, respectively.

| Condition | Methods | R-Precision ↑ | | | FID ↓ | MM-Dist ↓ | MModality ↑ |
| --- | --- | --- | --- | --- | --- | --- | --- |
| | | Top-1 ↑ | Top-2 ↑ | Top-3 ↑ | | | |
| baseline | w/o LIMM, ATII | $0.514^{\pm.003}$ | $0.703^{\pm.002}$ | $0.797^{\pm.002}$ | $0.302^{\pm.006}$ | $3.104^{\pm.009}$ | $\mathbf{1.386}^{\pm.068}$ |
| | w/o LIMM | $0.535^{\pm.002}$ | $0.730^{\pm.003}$ | $0.824^{\pm.002}$ | $0.255^{\pm.005}$ | $2.841^{\pm.008}$ | $1.035^{\pm.032}$ |
| | w/o ATII | $0.519^{\pm.002}$ | $0.709^{\pm.001}$ | $0.802^{\pm.001}$ | $\mathbf{0.052}^{\pm.002}$ | $2.945^{\pm.005}$ | $1.255^{\pm.033}$ |
| | **Event-T2M (Ours)** | $\mathbf{0.562}^{\pm.002}$ | $\mathbf{0.754}^{\pm.003}$ | $\mathbf{0.842}^{\pm.002}$ | $0.056^{\pm.002}$ | $\mathbf{2.711}^{\pm.005}$ | $0.949^{\pm.026}$ |
| 2 | w/o LIMM, ATII | $0.498^{\pm.002}$ | $0.693^{\pm.002}$ | $0.792^{\pm.003}$ | $0.347^{\pm.007}$ | $3.263^{\pm.008}$ | $\mathbf{1.555}^{\pm.057}$ |
| | w/o LIMM | $0.519^{\pm.003}$ | $0.716^{\pm.002}$ | $0.814^{\pm.002}$ | $0.227^{\pm.005}$ | $2.908^{\pm.005}$ | $1.056^{\pm.037}$ |
| | w/o ATII | $0.490^{\pm.003}$ | $0.676^{\pm.002}$ | $0.772^{\pm.002}$ | $\mathbf{0.076}^{\pm.003}$ | $3.098^{\pm.008}$ | $1.385^{\pm.043}$ |
| | **Event-T2M (Ours)** | $\mathbf{0.536}^{\pm.002}$ | $\mathbf{0.732}^{\pm.002}$ | $\mathbf{0.824}^{\pm.002}$ | $0.079^{\pm.003}$ | $\mathbf{2.836}^{\pm.006}$ | $0.976^{\pm.043}$ |
| 3 | w/o LIMM, ATII | $0.445^{\pm.004}$ | $0.644^{\pm.005}$ | $0.754^{\pm.004}$ | $0.536^{\pm.012}$ | $3.453^{\pm.011}$ | $\mathbf{1.639}^{\pm.052}$ |
| | w/o LIMM | $0.470^{\pm.005}$ | $0.669^{\pm.003}$ | $0.773^{\pm.003}$ | $0.349^{\pm.009}$ | $3.021^{\pm.009}$ | $1.120^{\pm.031}$ |
| | w/o ATII | $0.426^{\pm.004}$ | $0.616^{\pm.004}$ | $0.719^{\pm.004}$ | $0.207^{\pm.009}$ | $3.284^{\pm.011}$ | $1.521^{\pm.041}$ |
| | **Event-T2M (Ours)** | $\mathbf{0.487}^{\pm.005}$ | $\mathbf{0.687}^{\pm.004}$ | $\mathbf{0.790}^{\pm.004}$ | $\mathbf{0.137}^{\pm.003}$ | $\mathbf{2.928}^{\pm.010}$ | $1.010^{\pm.029}$ |
| 4 | w/o LIMM, ATII | $0.366^{\pm.008}$ | $0.547^{\pm.009}$ | $0.664^{\pm.009}$ | $0.813^{\pm.042}$ | $3.629^{\pm.023}$ | $\mathbf{1.853}^{\pm.046}$ |
| | w/o LIMM | $0.436^{\pm.007}$ | $0.652^{\pm.006}$ | $0.753^{\pm.006}$ | $0.520^{\pm.015}$ | $3.226^{\pm.014}$ | $1.118^{\pm.027}$ |
| | w/o ATII | $0.402^{\pm.004}$ | $0.599^{\pm.008}$ | $0.707^{\pm.010}$ | $0.368^{\pm.022}$ | $3.406^{\pm.016}$ | $1.606^{\pm.044}$ |
| | **Event-T2M (Ours)** | $\mathbf{0.466}^{\pm.008}$ | $\mathbf{0.660}^{\pm.008}$ | $\mathbf{0.767}^{\pm.007}$ | $\mathbf{0.265}^{\pm.007}$ | $\mathbf{3.063}^{\pm.015}$ | $1.039^{\pm.028}$ |

| Methods | MARDM-SiT | MARDM-DDPM | MoGenTS | MoMask | Light-T2M | AttT2M | **Event-T2M (+ LLM execution time)** |
| --- | --- | --- | --- | --- | --- | --- | --- |
| AIT (s) | 13.177 | 3.038 | 0.973 | 0.103 | 0.142 | 3.853 | 0.167 (+1.430) |

Table 17: Average inference time (AIT) comparison.

## A.17 EXTENDED ABLATIONS AND ROLE OF EACH MODULE

To more thoroughly analyze the contribution of the newly introduced components beyond our main architecture, we conducted additional ablations on LIMM and ATII. Starting from the full Event-T2M model, we removed each module individually and retrained under identical settings. The results are summarized in Table 16. In both cases, we observe consistent degradation in motion quality and text–motion alignment: FID becomes worse and R-Precision drops across all event-count subsets on HumanML3D-E, with the largest declines appearing on prompts with higher event complexity. This indicates that neither LIMM nor ATII is a superficial add-on; both play a meaningful role in stabilizing and aligning event-conditioned generation.

Conceptually, our goal is not to introduce entirely new low-level architectural primitives, but to design an event-centric pipeline whose components are tailored to the structure induced by event-level conditioning. LIMM is used to regularize local temporal dynamics and smooth transitions within and between event segments, preventing abrupt changes when strong event-level signals are injected. ATII adaptively injects global textual information in a way that depends on the current motion context, helping the model decide when to rely more on event-local cues and when to fall back on global semantics. The ablation results in Table 16 show that removing either LIMM or ATII consistently harms both distributional quality and alignment, supporting our claim that these modules are integral parts of the event-aware design rather than generic, easily replaceable components.

## A.18 COMPUTATIONAL COST AND LLM OVERHEAD

We evaluated the computational cost of Event-T2M on an NVIDIA A5000 GPU using an Average Inference Time (AIT) analysis that explicitly includes the latency introduced by the LLM-based event decomposition stage. To estimate the model-side inference cost, we randomly sampled 100 test captions and measured the time required to generate motions, obtaining an AIT of 0.1667 seconds per sample. We then separately measured the LLM latency by applying our event decomposition procedure to another set of 100 randomly selected test captions, which resulted in an average execution time of 1.4299 seconds per caption.

Combining these two components yields a total average cost of 1.5966 seconds per caption–motion pair. This analysis shows that, while Event-T2M does introduce an additional LLM preprocessing step, the resulting overhead is moderate relative to the overall inference pipeline and can be clearly quantified. The detailed comparison with existing methods is summarized in Table 17.

