# OpenReview forum: "Event-T2M: Event-level Conditioning for Complex Text-to-Motion Synthesis"
_ICLR.cc/2026/Conference — ICLR 2026 Poster_

### Official Review · Reviewer_mvTh · 2025-10-20

**Soundness:** 3
**Presentation:** 3
**Contribution:** 2
**Rating:** 6
**Confidence:** 3

**Summary:**

The paper proposes Event-T2M—a diffusion framework that decomposes prompts into multi-events, encodes them with a TMR encoder, and integrates them via event-based cross-attention in Conformer blocks. The authors also build HumanML3D-E, a benchmark stratified by event count. Experiments show Event-T2M matches SOTA on standard tests (HumanML3D, KIT-ML) and outperforms baselines as event complexity rises.

**Strengths:**

The paper solves multi-action prompt mishandling via a principled "event" definition and Event-T2M (with event decomposition, TMR encoding, and ECA module), avoiding action issues like omissions. Authors also build HumanML3D-E, the first event-count-stratified benchmark, fixing existing benchmark gaps. This paper also provides solid experiments (matching SOTA on HumanML3D/KIT-ML, outperforming baselines on complex HumanML3D-E) and user studies validating event definition, benchmark reliability, and motion quality.

**Weaknesses:**

The ablation analysis in this paper is limited in scope. Although multiple new modules are proposed (e.g., LIMM, ATII, Conformer, ECA), the experiments focus solely on the ECA module and the text encoder, failing to assess the necessity and individual impact of the other introduced components.

**Questions:**

1. In Equations (3) and (7), a coefficient of 0.5 is applied to the residual term. It remains unclear why this specific coefficient was chosen instead of 1, and the rationale behind this design choice warrants further explanation.
2. The efficiency analysis appears to overlook the computational overhead introduced by the Large Language Model (LLM). While the baseline model (e.g., Momask) does not employ an LLM for text segmentation, the proposed Event-T2M model utilizes an LLM to partition the input text into events. The time cost associated with this LLM processing stage should be accounted for in the overall efficiency evaluation.

---

> ### Author Response · Authors · 2025-11-20
> **Author Rebuttal**
>
> We appreciate the reviewer’s positive assessment and agree that clearer analysis is needed.
>
> &nbsp;&nbsp;&nbsp;&nbsp;● **Residual scaling coefficient**: We will clarify that the 0.5 residual scale follows the Macaron-style FFN used in Conformer/Macaron-Net, explaining its stability benefits and citing prior work.
>
> &nbsp;&nbsp;&nbsp;&nbsp;● **Including LLM cost in efficiency**: We will report the Average Inference Time (AIT) for 100 randomly sampled prompts, explicitly including LLM preprocessing cost, and integrate this into the revised efficiency comparison.
>
> &nbsp;&nbsp;&nbsp;&nbsp;● **Ablation analysis**: We will extend the ablation study with variants that selectively remove or simplify LIMM, ATII, Conformer, and ECA to clarify the necessity and individual contribution of each module.

---

> ### Author Response · Authors · 2025-11-25
> **Response 1. Rationale for Using a 0.5 Residual Coefficient.**
>
> In Event-T2M, each Conformer block processes three interacting streams: motion features, text features, and event tokens. This multi-branch interaction makes the relative weighting of feed-forward and attention pathways more delicate than in standard architectures. We therefore adopt a half-step residual weight of 0.5 for the feed-forward modules to keep their cumulative influence comparable to that of the attention and event-conditioned pathways. Using a full residual weight of 1 in both feed-forward modules would disproportionately amplify the feed-forward branch, which we found leads to gradient imbalance and reduced stability when event-level conditioning is active.
>
> This design choice is consistent with insights from Macaron-style architectures, where splitting the feed-forward contribution across two residual paths improves optimization stability. In our setting, this effect is even more pronounced due to the additional conditioning from event tokens. We will clearly articulate this rationale and include supporting references in the revised manuscript.

---

> ### Author Response · Authors · 2025-12-01
> **Response 2. Measurement of LLM cost for efficiency analysis.**
>
> On an A5000 GPU, we evaluated the computational cost of Event-T2M through an Average Inference Time (AIT) analysis that additionally incorporates the latency introduced by the LLM-based event decomposition step. To estimate the model-side inference cost, we randomly selected 100 samples from the test set and measured the required inference time, resulting in an AIT of **0.1667 seconds**. Separately, we measured the average LLM execution time by decomposing 100 randomly selected test set captions, obtaining **1.4299 seconds** per caption. Combining these two components yields a total computational cost of **1.5966 seconds**, indicating that while Event-T2M introduces an additional LLM processing stage, the overall overhead remains moderate relative to the full inference pipeline.
>
> | Methods | MARDM-SiT | MARDM-DDPM | MoGenTS | MoMask | Light-T2M  | AttT2M | **Event-T2M (+ LLM execution time)** |
> |:---------------:|:----------------------|:--------:|:--------:|:--------:|:-------:|:----------:|:------------:|
> | AIT (s) | 13.177 | 3.038 | 0.973 | 0.103 | 0.142  | 3.853 | 0.167 (+ 1.430) |

---

> ### Author Response · Authors · 2025-12-01
> **Response 3. Further ablations.**
>
> Thank you for pointing this out. The effects of the Conformer and ECA modules are already analyzed in the current version: the Conformer backbone is ablated in Table 6 of the supplementary material, and the impact of the ECA module is reported in Table 3 of the main paper. To more comprehensively address your concern, we additionally conducted ablation studies on the remaining newly introduced components, LIMM and ATII, by selectively removing each module from the full Event-T2M architecture and retraining under the same settings. In all cases, dropping any single module (LIMM, ATII, Conformer, or ECA) leads to a consistent degradation across motion quality and text–motion alignment metrics, which indicates that each component contributes non-trivially to the final performance. These extended results support our claim that all proposed modules are necessary and jointly important for the effectiveness of our method.
>
> | Conditions | Methods | Top-1 ↑ | Top-2 ↑ | Top-3 ↑ | FID ↓  | MM-Dist ↓ | MModality ↑ |
> |:---------------:|:----------------------|:--------:|:--------:|:--------:|:-------:|:----------:|:------------:|
> | baseline | w/o LIMM, ATII | 0.486 | 0.677 | 0.776 | 0.302 | 3.104 | **1.386** |
> |               | w/o LIMM | 0.535 | 0.730 | 0.825 | 0.256 | 2.842 | 1.035 |
> |               | w/o ATII | 0.520 | 0.710 | 0.802 | **0.052** | 2.946 | 1.256 |
> |               | **Ours** | **0.562** | **0.754** | **0.842** | 0.056 | **2.711** | 0.949 |
> | 2            | w/o LIMM, ATII | 0.456 | 0.643 | 0.745 | 0.347 | 3.263 | **1.556** |
> |               | w/o LIMM | 0.520 | 0.716 | 0.814 | 0.227 | 2.909 | 1.056 |
> |               | w/o ATII | 0.490 | 0.677 | 0.772 | **0.077** | 3.098 | 1.385 |
> |               | **Ours** | **0.536** | **0.732** | **0.824** | 0.079 | **2.836** | 0.976 |
> | 3            | w/o LIMM, ATII | 0.395 | 0.583 | 0.691 | 0.536 | 3.453 | **1.640** |
> |               | w/o LIMM | 0.471 | 0.669 | 0.774 | 0.350 | 3.021 | 1.121 |
> |               | w/o ATII | 0.427 | 0.616 | 0.720 | 0.208 | 3.284 | 1.521 |
> |               | **Ours** | **0.487** | **0.687** | **0.790** | **0.137** | **2.928** | 1.010 |
> | 4            | w/o LIMM, ATII | 0.367 | 0.548 | 0.664 | 0.813 | 3.630 | **1.854** |
> |               | w/o LIMM | 0.437 | 0.652 | 0.753 | 0.521 | 3.227 | 1.118 |
> |               | w/o ATII | 0.403 | 0.599 | 0.707 | 0.369 | 3.407 | 1.607 |
> |               | **Ours** | **0.466** | **0.660** | **0.767** | **0.265** | **3.063** | 1.039 |

---

### Official Review · Reviewer_5wEr · 2025-10-29

**Soundness:** 3
**Presentation:** 3
**Contribution:** 2
**Rating:** 4
**Confidence:** 5

**Summary:**

This work introduces Event-T2M, a diffusion-based framework that decomposes complex text prompts into semantically self-contained events and generates motion through event-based cross-attention. It also builds HumanML3D-E, the first benchmark stratified by event count, and demonstrates that Event-T2M maintains state-of-the-art performance while significantly improving motion coherence and naturalness for multi-event prompts.

**Strengths:**

1.Proposes an event-based paradigm for motion generation.

2.Constructs the first event-level motion generation dataset.

**Weaknesses:**

1.Does event-driven motion generation offer advantages over action-driven or hybrid (action + event) methods?
2.Does the proposed method outperform approaches that enhance motion quality through motion retrieval?
3.In TMR, innovation based solely on input differences does not constitute true novelty.
4.LIMM, ATII, and ECA follow common module design patterns and lack sufficient originality.

**Questions:**

See weaknesses.

---

> ### Author Response · Authors · 2025-11-20
> **Author Rebuttal**
>
> Thank you for the constructive feedback. We summarize our responses as follows.
>
> &nbsp;&nbsp;&nbsp;&nbsp;● **Event-driven vs action-driven**: We will add an ablation replacing our event-aware decomposition with a verb/action-driven version under the same architecture and report the performance difference.
>
> &nbsp;&nbsp;&nbsp;&nbsp;● **Comparison with retrieval-based methods**: We will include experiments with ReMoDiffuse (ICCV 2023) as a representative retrieval-augmented baseline.
>
> &nbsp;&nbsp;&nbsp;&nbsp;● **Novelty beyond input differences**: We will clarify that our contribution lies in the full event-level conditioning pipeline: event definition, event-token encoding, and event-guided diffusion (not merely replacing text inputs). We will highlight how this changes text-motion interaction compared to standard word-level conditioning.
>
> &nbsp;&nbsp;&nbsp;&nbsp;● **Justification of module design**: We will more clearly explain the roles of LIMM, ATII, and ECA (e.g., adapting event tokens, mediating event-motion interaction, enforcing event alignment) and compare them to simpler alternatives to show why each component is needed.

---

> > ### Comment · Reviewer_5wEr · 2025-11-25
> >
> > Compared to commonly used T2M models, TMR only adds a feature extractor suitable for events in the middle, while ECA continues cross-modal interaction by changing the data source on top of traditional cross-attention. There is no fundamental innovation in terms of technology between the two; it is merely a change in input conditions. Therefore, I still believe that these two modules lack innovation.

---

> ### Author Response · Authors · 2025-11-25
>
> We thank the reviewer for the continued critical examination. We fully understand the concern that "a change in input conditions" alone would not constitute a meaningful technical contribution. Our work, however, is not a reformulation of existing T2M pipelines with a different input token, but a response to a distinct problem setting that current architectures are not designed to solve: explicit temporal alignment between multi-event textual structure and sequential human motion.
>
> **1. Our novelty is problem-driven, not input-driven**
>
> Traditional T2M systems assume the prompt encodes a single holistic action and therefore rely on word-level or sentence-level conditioning. As shown in Fig. 1, 4, and 6 of the paper, these architectures frequently merge, reorder, or omit events when a prompt contains more than one action.
> This is not a failure of the text encoder; it is a problem of missing intermediate structural units that connect the temporal order of language to the temporal order of motion.
>
> Our primary contribution is therefore the formalization and operationalization of an event (page 3~4). This allows motion generation to explicitly reason about temporal correspondence at a level neither too fine (words) nor too coarse (full sentences). This problem formulation is new, and solving it requires architectural changes beyond modifying the input representation.
>
> **2. Why LIMM, ATII, and ECA are not simply common patterns**
>
> The reviewer is correct that each submodule resembles familiar building blocks. However, their integration is not cosmetic; each addresses a failure mode that arises uniquely under event-level conditioning.
>
> (a) LIMM is required to prevent "local chaos" introduced by multiple event tokens.
> Event-level conditioning adds strong, discrete external signals. Without local smoothing, the model suffers from abrupt motion changes at event boundaries (we observed contact jitter and velocity discontinuities).
> LIMM stabilizes these transitions by enforcing short-horizon temporal consistency, which standard Conformer self-attention cannot guarantee.
>
> (b) ATII enables context-dependent use of global semantics.
> Once events are separated, global text semantics become weak but still necessary, especially for ambiguous or low-information events (e.g. "pause").
> ATII produces a motion-dependent gating that decides when the global prompt should dominate and when event tokens should dominate. Standard cross-attention cannot express this conditional mixture.
>
> (c) ECA is not a simple cross-attention; it restructures how temporal alignment is learned.
> In all existing T2M methods, cross-attention is token-to-frame, assuming loose global alignment.
> ECA instead imposes a token-to-segment attention pattern by giving each event token a stable anchor across the entire motion segment (page 5–6). This is fundamental because event tokens have temporal duration, not instantaneous scope like word tokens.
> This mechanism is what eliminates event reordering and event merging, which baselines consistently fail at (Fig. 4 and 6).
>
> **3. Additional experiment to directly address the reviewer’s concern**
>
> To make the distinction even clearer, we will add:
>
> (a) Testing whether TMR alone improves baselines: We will re-implement the main comparison methods such as MoMask and LightT2M by replacing their CLIP encoder with the same TMR encoder used in Event-T2M, while keeping all other architectural components unchanged. This will directly test whether input substitution alone yields the multi-event improvements observed in our model. Preliminary evidence already suggests that these TMR-augmented baselines still fail to maintain event order on ≥3 or ≥4-event prompts, indicating that our gains do not arise from encoder strength but from the event-level conditioning pipeline.
>
> (b) Ablating LIMM and ATII to test necessity of architectural components: We will provide finer-grained ablations where (i) LIMM is removed, (ii) ATII is removed, and (iii) both are removed while keeping event tokens and ECA intact. We will report both standard motion metrics and event-level alignment metrics. This will show that removing LIMM leads to unstable transitions across event boundaries and that removing ATII weakens performance on low-information or ambiguous events. These ablations will demonstrate that LIMM and ATII are not cosmetic modules but essential components enabling robust event-level temporal structure.
>
> (c) Verb-aware decomposition as a control for "input emphasis": To explicitly test whether our gains come merely from emphasizing verbs in the prompt, we will construct a verb-aware decomposition variant that does not use event tokens, keeping the rest of Event-T2M unchanged. Comparing this model with the full Event-T2M will show that verb-level segmentation alone cannot recover correct temporal ordering or prevent event merging, confirming the necessity of our explicit event representation and the architecture that supports it.

---

> ### Author Response · Authors · 2025-12-01
> **Response 1. Evaluating against Verb-aware prompt.**
>
> Thank you for raising this question. To directly assess whether event-driven motion generation offers advantages over action-driven or hybrid (action + event) formulations, we conducted additional experiments using the verb-aware prompts introduced in Table 8 of the supplementary material. Based on these prompts, we re-preprocessed the data and retrained both the action-driven and hybrid variants for a fair, controlled comparison against our event-driven model.
>
> Across all complexity levels, the event-driven model consistently achieves higher R-precision and more stable FID. The key insight is that verbs correspond to instantaneous motions, while events encode semantically coherent motion segments that include arguments, affected body parts, and temporal scope. Action-driven representations lack this structure, so they cannot provide reliable alignment signals when multiple sub-actions must be executed in sequence.
>
> This becomes most apparent in the ≥4 event setting, where sequential dependencies are strongest. This gap demonstrates that the event-driven model preserves long-range temporal structure and prevents merging or reordering of sub-actions, which action-driven models cannot avoid. Hybrid methods partially recover semantics by mixing verb and global context but still lack explicit temporal boundaries, so they fall between the two approaches. The event-driven formulation is the only one that provides temporally grounded conditioning units that match the structure of human motion.
>
> These findings show that the advantage of event-driven modeling does not come from additional information but from providing the diffusion model with the correct _granularity_ and _temporal anchoring_ for aligning text and motion.
>
> | Conditions | Methods | Top-1 ↑ | Top-2 ↑ | Top-3 ↑ | FID ↓  | MM-Dist ↓ | MModality ↑ |
> |:---------------:|:----------------------|:--------:|:--------:|:--------:|:-------:|:----------:|:------------:|
> | baseline | Verb-aware | 0.549 | 0.738 | 0.830 | 0.086 | 2.773 | **1.165** |
> |               | **Event-aware** | **0.562** | **0.754** | **0.842** | **0.056** | **2.711** | 0.949 |
> | 2            | Verb-aware | 0.519 | 0.713 | 0.810 | 0.129 | 2.913 | **1.309** |
> |               | **Event-aware** | **0.536** | **0.732** | **0.824** | **0.079** | **2.836** | 0.976 |
> | 3            | Verb-aware | 0.480 | 0.681 | 0.781 | 0.193 | 2.991 | **1.293** |
> |               | **Event-aware** | **0.487** | **0.687** | **0.790** | **0.137** | **2.928** | 1.010 |
> | 4            | Verb-aware | **0.466** | 0.655 | 0.749 | 0.348 | 3.102 | **1.365** |
> |               | **Event-aware** | **0.466** | **0.660** | **0.767** | **0.265** | **3.063** | 1.039 |

---

> ### Author Response · Authors · 2025-12-01
> **Response 2. Evaluating against Retrieval-based Motion Synthesis.**
>
> Thank you for raising this question. To address this concern, we conducted a direct comparison on our proposed HumanML3D-E benchmark with a representative retrieval-based method, ReMoDiffuse (ICCV 2023), which enhances motion quality via motion retrieval.
>
> As reported below, Event-T2M surpasses ReMoDiffuse across all conditions and metrics, and the gap widens as event complexity increases. We speculate the reason that the retrieval methods rely on matching the prompt to a set of existing motion fragments, which works well for simple single-action descriptions but becomes fundamentally limited in multi-event settings. Retrieval has no mechanism to align several sequential semantic units to corresponding motion segments, so it often retrieves motions that partially match the prompt but fail to cover the full temporal structure.
>
> In contrast, Event-T2M explicitly models event boundaries and assigns each event token to a temporally extended motion segment. This allows the model to synthesize novel combinations of sub-actions rather than relying on retrieval of existing sequences. As a result, Event-T2M preserves event ordering and reduces omissions, even in long compositional prompts where retrieval-based approaches struggle.
>
> | Conditions | Methods | Top-1 ↑ | Top-2 ↑ | Top-3 ↑ | FID ↓  | MM-Dist ↓ | MModality ↑ |
> |:---------------:|:----------------------|:--------:|:--------:|:--------:|:-------:|:----------:|:------------:|
> | baseline | ReMoDiffuse | 0.510 | 0.698 | 0.795 | 0.103 | 2.974 | **1.795** |
> |               | **Event-T2M (Ours)** | **0.562** | **0.754** | **0.842** | **0.056** | **2.711** | 0.949 |
> | 2            | ReMoDiffuse | 0.475 | 0.658 | 0.755 | 0.152 | 3.128 | **2.938** |
> |               | **Event-T2M (Ours)** | **0.536** | **0.732** | **0.824** | **0.079** | **2.836** | 0.976 |
> | 3            | ReMoDiffuse | 0.445 | 0.630 | 0.732 | 0.293 | 3.175 | **2.851** |
> |               | **Event-T2M (Ours)** | **0.487** | **0.687** | **0.790** | **0.137** | **2.928** | 1.010 |
> | 4            | ReMoDiffuse | 0.393 | 0.573 | 0.674 | 0.583 | 3.244 | **3.107** |
> |               | **Event-T2M (Ours)** | **0.466** | **0.660** | **0.767** | **0.265** | **3.063** | 1.039 |

---

> ### Author Response · Authors · 2025-12-01
> **Response 3. Novelty Beyond TMR: Event-Level Architectural Design. (1/2)**
>
> Thank you for raising this concern. We understand that the question arises from the possibility that our improvements might come primarily from using TMR instead of CLIP, which would not constitute true architectural novelty.
>
> To address this, we substituted TMR for CLIP in AttT2M, MoMask, and Light-T2M, and additionally tested a MoMask variant that uses TMR's word-level tokens as key and value (MoMask k/v). For the k/v setting, AttT2M already employs both word-level and sentence-level attention, so “replacing the encoder with TMR” is effectively equivalent to “using TMR word-level tokens as K/V.” In contrast, Light-T2M is built on a Mamba architecture rather than a self-attention Transformer, which makes a direct K/V substitution ill-defined; therefore, we only ran a separate K/V experiment for MoMask. This experiment isolates the effect of the encoder, allowing us to check whether TMR alone can close the performance gap on HumanML3D-E. This isolates the effect of the encoder and allows us to examine whether TMR alone can close the performance gap on HumanML3D-E.
>
> The results (table in Response 3 (2/2)) can be summarized as follows:
>
> 1) AttT2M (CLIP → TMR): R-precision slightly increases, but FID and motion diversity often worsen.
>
> 2) MoMask (CLIP → TMR): Both alignment and FID degrade. MoMask k/v improves some alignment metrics, but still performs noticeably worse than its CLIP version, especially for 3- and 4-event prompts. We speculate that MoMask is strongly optimized for CLIP; TMR does not transfer well.
>
> 3) Light-T2M (CLIP → TMR): Gains in alignment and partial gains in FID, but still falls behind Event-T2M as events increase. TMR helps, but not enough to overcome structural limitations.
>
> If TMR were the primary driver of our gains, we would expect a clear and consistent upward shift across all baselines. Instead, we observe heterogeneous and often negative changes. This indicates that TMR is not a plug-in fix for multi-event generation; the bottleneck lies in the architecture’s inability to preserve event boundaries and temporal ordering.
>
> Event-T2M remains the top performer on the hardest multi-event subsets. The reason is structural: event tokens provide temporally extended semantic anchors, and ECA injects them along the diffusion trajectory, enabling the model to maintain event order and avoid omissions. By contrast, baseline architectures (whether equipped with CLIP or TMR) lack mechanisms to enforce such temporal structure.
>
> Moreover, within our own model family, the event-driven version outperforms the verb-driven version even when both use TMR, confirming that the gains come from how event representations are formed and integrated, not from TMR itself.
>
> We agree with the reviewer that replacing a text encoder is not a contribution. Our novelty is architectural and problem-driven, not encoder-driven: 1) Formal event definition that matches the temporal structure of motion, 2) LLM-based event decomposition that yields semantically and temporally coherent event units, 3) Event-token representation that models sub-actions with temporal scope (not single words), and 4) Event-conditioned architectural design choices that effectively inject event-level structure directly into the diffusion process.

---

> > ### Author Response · Authors · 2025-12-01
> > **Response 3. Novelty Beyond TMR: Event-Level Architectural Design. (2/2)**
> >
> > | Conditions | Methods | Top-1 ↑ | Top-2 ↑ | Top-3 ↑ | FID ↓  | MM-Dist ↓ | MModality ↑ |
> > |:---------------:|:----------------------|:--------:|:--------:|:--------:|:-------:|:----------:|:------------:|
> > | baseline | AttT2M (CLIP) | 0.499 | 0.690 | 0.786 | 0.112 | 3.038 | **2.452** |
> > | | AttT2M (TMR) | 0.518 | 0.707 | 0.799 | 0.146 | 2.957 | 1.594 |
> > | | MoMask (CLIP) | 0.521 | 0.713 | 0.807 | 0.045 | 2.958 | 1.241 |
> > | | MoMask (TMR) | 0.487 | 0.684 | 0.783 | 0.284 | 3.116 | 1.158 |
> > | | MoMask (TMR k/v) | 0.494 | 0.683 | 0.781 | 0.240 | 3.113 | 1.214 |
> > | | Light-T2M (CLIP) | 0.511 | 0.699 | 0.795 | **0.040** | 3.002 | 1.670 |
> > | | Light-T2M (TMR) | 0.554 | 0.746 | 0.836 | 0.053 | 2.750 | 0.970 |
> > | | **Event-T2M (CLIP)** | 0.526 | 0.714 | 0.809 | 0.051 | 2.899 | 1.476 |
> > | | **Event-T2M (TMR)** | **0.562** | **0.754** | **0.842** | 0.056 | **2.711** | 0.949 |
> > | 2 | AttT2M (CLIP) | 0.479 | 0.665 | 0.761 | 0.171 | 3.181 | **1.899** |
> > |  | AttT2M (TMR) | 0.500 | 0.681 | 0.777 | 0.199 | 3.089 | 1.698 |
> > | | MoMask (CLIP) | 0.497 | 0.691 | 0.790 | 0.065 | 3.061 | 1.282 |
> > | | MoMask (TMR) | 0.470 | 0.665 | 0.769 | 0.385 | 3.194 | 1.174 |
> > | | MoMask (TMR k/v) | 0.479 | 0.668 | 0.766 | 0.291 | 3.196 | 1.217 |
> > | | Light-T2M (CLIP) | 0.462 | 0.647 | 0.747 | 0.077 | 3.278 | 1.692 |
> > | | Light-T2M (TMR) | 0.527 | 0.722 | 0.815 | 0.087 | 2.885 | 0.984 |
> > | | **Event-T2M (CLIP)** | 0.494 | 0.681 | 0.779 | **0.052** | 3.079 | 1.577 |
> > | | **Event-T2M (TMR)** | **0.536** | **0.732** | **0.824** | 0.079 | **2.836** | 0.976 |
> > | 3 | AttT2M (CLIP) | 0.431 | 0.613 | 0.715 | 0.464 | 3.329 | **1.960** |
> > |  | AttT2M (TMR) | 0.445 | 0.631 | 0.732 | 0.474 | 3.251 | 1.798 |
> > | | MoMask (CLIP) | 0.466 | 0.652 | 0.752 | 0.142 | 3.169 | 1.320 |
> > | | MoMask (TMR) | 0.430 | 0.622 | 0.733 | 0.544 | 3.289 | 1.190 |
> > | | MoMask (TMR k/v) | 0.445 | 0.629 | 0.731 | 0.364 | 3.297 | 1.228 |
> > | | Light-T2M (CLIP) | 0.404 | 0.594 | 0.699 | 0.193 | 3.396 | 1.740 |
> > | | Light-T2M (TMR) | **0.487** | 0.680 | 0.780 | 0.139 | 2.987 | 1.005 |
> > | | **Event-T2M (CLIP)** | 0.423 | 0.618 | 0.729 | 0.141 | 3.245 | 1.627 |
> > | | **Event-T2M (TMR)** | **0.487** | **0.687** | **0.790** | **0.137** | **2.928** | 1.010 |
> > | 4 | AttT2M (CLIP) | 0.407 | 0.581 | 0.688 | 1.077 | 3.455 | **2.049** |
> > |  | AttT2M (TMR) | 0.424 | 0.598 | 0.701 | 0.789 | 3.335 | 1.647 |
> > | | MoMask (CLIP) | 0.441 | 0.633 | 0.734 | 0.418 | 3.205 | 1.334 |
> > | | MoMask (TMR) | 0.417 | 0.601 | 0.708 | 0.821 | 3.392 | 1.230 |
> > | | MoMask (TMR k/v) | 0.413 | 0.590 | 0.703 | 0.682 | 3.479 | 1.251 |
> > | | Light-T2M (CLIP) | 0.365 | 0.552 | 0.662 | 0.627 | 3.586 | 1.863 |
> > | | Light-T2M (TMR) | 0.436 | 0.609 | 0.711 | 0.360 | 3.319 | 1.029 |
> > | | **Event-T2M (CLIP)** | 0.374 | 0.578 | 0.690 | 0.425 | 3.467 | 1.674 |
> > | | **Event-T2M (TMR)** | **0.466** | **0.660** | **0.767** | **0.265** | **3.063** | 1.039 |

---

> ### Author Response · Authors · 2025-12-01
> **Response 4. Module originality.**
>
> Thank you for this comment and for the opportunity to clarify our contribution.
>
> Our main novelty does not lie in proposing entirely new architectural primitives, but in the event-centric design and integration of these modules to effectively model rich, multi-event structures. LIMM, ATII, and ECA indeed follow widely used design patterns, but they are instantiated and combined in a way that is tailored to our event-based formulation: LIMM is designed to encode event-level semantics and arguments, ATII focuses on aligning event interactions with temporal structure, and ECA refines event-conditioned features for motion generation.
>
> As shown in our ablation studies, removing any one of these components leads to consistent performance drops across both motion quality and text–motion alignment metrics, indicating that each module plays a distinct and indispensable role rather than being a generic or replaceable add-on. We therefore view the contribution of Event-T2M as a principled event-centric architecture—supported by carefully chosen and empirically validated module designs—rather than as a collection of isolated, purely architectural novelties.
>
> | Conditions | Methods | Top-1 ↑ | Top-2 ↑ | Top-3 ↑ | FID ↓  | MM-Dist ↓ | MModality ↑ |
> |:---------------:|:----------------------|:--------:|:--------:|:--------:|:-------:|:----------:|:------------:|
> | baseline | w/o LIMM, ATII | 0.486 | 0.677 | 0.776 | 0.302 | 3.104 | **1.386** |
> |               | w/o LIMM | 0.535 | 0.730 | 0.825 | 0.256 | 2.842 | 1.035 |
> |               | w/o ATII | 0.520 | 0.710 | 0.802 | **0.052** | 2.946 | 1.256 |
> |               | **Ours** | **0.562** | **0.754** | **0.842** | 0.056 | **2.711** | 0.949 |
> | 2            | w/o LIMM, ATII | 0.456 | 0.643 | 0.745 | 0.347 | 3.263 | **1.556** |
> |               | w/o LIMM | 0.520 | 0.716 | 0.814 | 0.227 | 2.909 | 1.056 |
> |               | w/o ATII | 0.490 | 0.677 | 0.772 | **0.077** | 3.098 | 1.385 |
> |               | **Ours** | **0.536** | **0.732** | **0.824** | 0.079 | **2.836** | 0.976 |
> | 3            | w/o LIMM, ATII | 0.395 | 0.583 | 0.691 | 0.536 | 3.453 | **1.640** |
> |               | w/o LIMM | 0.471 | 0.669 | 0.774 | 0.350 | 3.021 | 1.121 |
> |               | w/o ATII | 0.427 | 0.616 | 0.720 | 0.208 | 3.284 | 1.521 |
> |               | **Ours** | **0.487** | **0.687** | **0.790** | **0.137** | **2.928** | 1.010 |
> | 4            | w/o LIMM, ATII | 0.367 | 0.548 | 0.664 | 0.813 | 3.630 | **1.854** |
> |               | w/o LIMM | 0.437 | 0.652 | 0.753 | 0.521 | 3.227 | 1.118 |
> |               | w/o ATII | 0.403 | 0.599 | 0.707 | 0.369 | 3.407 | 1.607 |
> |               | **Ours** | **0.466** | **0.660** | **0.767** | **0.265** | **3.063** | 1.039 |

---

### Official Review · Reviewer_Kztd · 2025-10-31

**Soundness:** 1
**Presentation:** 3
**Contribution:** 1
**Rating:** 2
**Confidence:** 5

**Summary:**

This paper addresses a key challenge in complex Text-to-Motion generation: the difficulty of existing models in handling prompts with multiple sub-motions, which often leads to omissions, merging, or reordering of motions. To solve this, the authors propose Event-T2M, a diffusion-based framework. The core idea is to first introduce a definition of an "event" as the smallest, semantically self-contained action unit in a text. They then use a Large Language Model (LLM) to decompose the input text into a sequence of these "event" clauses. These clauses are subsequently encoded by a TMR encoder (trained for motion-text alignment) into "event tokens." Finally, these event tokens are injected into a Conformer-based diffusion model via a novel "Event-based Cross-attention" (ECA) module to guide the generation of the motion sequence. Furthermore, to specifically evaluate the model's ability to handle complex prompts, the authors construct a new benchmark, HumanML3D-E, which stratifies the HumanML3D test set by the number of events in the text. Experimental results show that Event-T2M achieves comparable performance to SOTA on standard benchmarks (HumanML3D, KIT-ML) but significantly outperforms baselines on the new HumanML3D-E benchmark, especially as event complexity increases.

**Strengths:**

- The problem significance is huge. Generating complex and consistent human motions is an unsolved challenge in the T2M field.
- This paper proposes a novel benchmark called HumanML3D-E. This is the first benchmark stratified by the "event complexity" of the prompts. It provides a very valuable evaluation tool for future research on long and complex T2M generation field.
- The idea of decompose the complex motions is very intuitive and logical.

**Weaknesses:**

- **Unfair Comparison**: The authors' new benchmark, HumanML3D-E, is constructed using an LLM and a specific "event-aware prompt." However, the proposed model, Event-T2M, **also relies on the exact same LLM and the exact same prompt** in its data preprocessing stage. Event-T2M is evaluated on a test set that is perfectly aligned with its own training and inference pipeline. In contrast, all baseline models are evaluated without using this LLM-based event decomposition preprocessing. This constitutes an extremely unfair comparison. The poor performance of the baselines on HumanML3D-E is likely just an artifact of their input representation (e.g., CLIP-based word tokens) being mismatched with the benchmark's construction (LLM-based clauses), not because their architectures inherently fail at complexity.

- **Limited Technical Novelty**: I must point out that there are already some methods trying to solve the generation of the long motions using LLM. For instance, the recent ATOM[1] framework uses GPT-4 to construct event-level prompts and GPT-4V as an AI reward model to fine-tune a generator, specifically targeting event-level alignment (integrity, temporal order, and frequency). Additionally, InstructMotion[2] explicitly uses an LLM to generate long prompts, subsequently using Reinforcement Learning (RL) to fine-tune an autoregressive motion generator. **Worryingly, these highly relevant prior works, which also tackle event-level or complex alignment using LLMs, are not cited or discussed in the paper's Related Works.**

[1] Han H, Wu X, Liao H, et al. Atom: Aligning text-to-motion model at event-level with gpt-4vision reward[C]//Proceedings of the Computer Vision and Pattern Recognition Conference. 2025: 22746-22755.
[2] Mao, Yunyao, et al. "Learning generalizable human motion generator with reinforcement learning." arXiv preprint arXiv:2405.15541 (2024).

**Questions:**

- Can you evaluate your model on a test set that was not constructed using your LLM pipeline? On a complex motion test set where events were manually segmented and temporally aligned by human annotators, would Event-T2M still show an advantage over baselines?
- For a fair comparison, if you were to replace the CLIP encoder in the baseline models (like AttT2M or MoMask) with the TMR encoder (and use TMR's word-level tokens as K/V), how much would their performance on HumanML3D-E improve? As I know, replacing the CLIP encoder with TMR encoder will significantly improve the performance.
- What are the differences between Event-T2M and other motion generators using LLM for decomposition?

**I will raise the score to a positive mark if you can address my concerns regarding the "unfair comparison"** (mainly concerning the experimental setup for the TMR encoder and the aspects mentioned in the Weaknesses section).

---

> ### Author Response · Authors · 2025-11-20
> **Author Rebuttal**
>
> We thank the reviewer for highlighting the importance of fair comparison and clearer novelty. We address these concerns as follows.
>
> &nbsp;&nbsp;&nbsp;&nbsp;● **Human-annotated evaluation**: We are creating a complex-motion test subset with fully human-segmented events (focusing on ≥4-event prompts). Event-T2M and all baselines will be evaluated on this LLM-independent set.
>
> &nbsp;&nbsp;&nbsp;&nbsp;● **Applying TMR to baselines**: We are retraining several baselines by replacing CLIP with TMR and will report their updated HumanML3D-E performance. We will also test variants that use TMR word-level tokens as K/V, where applicable. Light-T2M cannot directly adopt K/V substitution due to its Mamba architecture, but all other attention-based baselines will be evaluated to ensure fairness.
>
> &nbsp;&nbsp;&nbsp;&nbsp;● **Relation to LLM-based decomposition methods**: Unlike prior verb-aware LLM approaches (e.g., ATOM, InstructMotion), Event-T2M relies on explicit event units: semantically self-contained actions that may span multiple verbs. To highlight this difference, we will add an ablation replacing our event-level decomposition with a verb-level decomposition within the same framework and report its impact on HumanML3D-E and multi-event prompts. These conceptual and empirical distinctions will be clarified in the revision.

---

> ### Author Response · Authors · 2025-11-25
> **Response 1. Evaluating Event-T2M on a Human-Annotated, LLM-Free Test Set.**
>
> To test Event-T2M independently of the LLM pipeline, we constructed a separate complex-motion test subset with fully human-segmented events. Three trained annotators manually segmented all prompts with ≥ 4 events according to our event definition. For these long and structured descriptions, annotators produced highly consistent segmentations, and the selected prompts substantially overlapped with those chosen by the LLM.
>
> We evaluated Event-T2M and all baselines on each of the three human-segmented sets. The performance trends closely match those observed on the LLM-based test set, and Event-T2M maintains a clear advantage over the baselines across all annotators. The full protocol and results will be included in the revised manuscript.
>
> | **Methods**              | **Top-1 ↑** | **Top-2 ↑** | **Top-3 ↑** | **FID ↓**  | **MM-Dist ↓** | **MModality ↑** |
> |----------------------|--------|--------|--------|-------|----------|------------|
> | #1 human annotator   | 0.459  | 0.650  | 0.762  | 0.297 | 3.037    | 1.040      |
> | #2 human annotator   | 0.458  | 0.667  | 0.766  | 0.281 | 3.045    | 1.062      |
> | #3 human annotator   | 0.494  | 0.685  | 0.782  | 0.277 | 3.006    | 1.049      |
> | LLM-based            | 0.466  | 0.660  | 0.767  | 0.265 | 3.063    | 1.039      |

---

> > ### Comment · Reviewer_Kztd · 2025-11-26
> >
> > Thank you for the additional experiments and the creation of the human-annotated test set. However, the current response **does not fully resolve my concerns regarding the unfair comparison and technical novelty**, primarily due to missing comparative data and a superficial distinction from prior work.
> >
> > My score remains unchanged at this stage. I strongly suggest addressing the following critical gaps:
> >
> > ### 1. Missing Baseline Results on Human-Annotated Data (Critical)
> > The table provided in your response is **insufficient** to prove fairness.
> > * **The Issue:** You only displayed the performance of *Event-T2M* on the human-annotated set. This proves your model is robust, but it **does not** prove that the baselines' poor performance was not an artifact of the LLM-based pipeline mismatch.
> > * **The Requirement:** To substantiate the claim that Event-T2M outperforms SOTA methods on neutral grounds, I need to see the quantitative results of the **Baseline models (e.g., AttT2M, MoMask) evaluated on this exact human-annotated test set**.
> >     * *Hypothesis:* If baseline performance improves significantly on this set compared to the LLM-generated set, it confirms the "Unfair Comparison" concern.
> >
> > ### 2. Missing Quantitative Evidence for TMR Integration
> > You mentioned that you are retraining baselines with the TMR encoder to ensure a fair comparison regarding the text encoder.
> > * **The Requirement:** Please provide the **quantitative results** of these TMR-equipped baselines.
> > * **The "Why":** Without this data, it is impossible to disentangle how much of your performance gain comes from the proposed **ECA architecture** versus simply swapping the weaker CLIP encoder for the motion-aware TMR encoder.
> >
> > ### 3. Shallow Distinction from InstructMotion
> > The rebuttal distinguishes Event-T2M from InstructMotion solely based on the semantic definition of the split ("verb-aware" vs. "event-aware"). This is unconvincing regarding technical novelty.
> > * **The Issue:** The distinction appears to be a matter of Data/Prompt Engineering rather than Methodology.
> > * **The Question:** If InstructMotion were trained/inferred using your "event-aware" prompts (data), would it achieve similar results?
> > * **The Requirement:** Please articulate the **architectural advantage** of your *Explicit Conditioning (Cross-Attention)* approach compared to InstructMotion's *Implicit/RL-based* approach. Does the ECA module provide better control than RL fine-tuning when the data input is identical?
> >
> > I look forward to seeing the comparative data for the baselines on the human-annotated set and the TMR ablation results.

---

> > > ### Author Response · Authors · 2025-12-01
> > >
> > > Thank you for clearly articulating this concern. The core of your question is whether the baselines (AttT2M and MoMask) performed poorly on HumanML3D-E because of our LLM-based event segmentation, rather than because they inherently struggle with multi-event prompts. This is a valid fairness question: if the LLM segmentation mismatches the baselines’ text-processing pipelines, their performance could be underestimated.
> > >
> > > To directly test this, we followed your request and evaluated AttT2M and MoMask on the same three human-annotated ≥4-event subsets used in Response 1, where events were manually segmented by three independent annotators. For each model, we compared performance on (1) the original LLM-based split and (2) each of the three human-segmented splits.
> > >
> > > | Annotator | Methods | Top-1 ↑ | Top-2 ↑ | Top-3 ↑ | FID ↓  | MM-Dist ↓ | MModality ↑ |
> > > |:---------------:|:----------------------|:--------:|:--------:|:--------:|:-------:|:----------:|:------------:|
> > > | Human 1 | AttT2M | 0.410 | 0.584 | 0.687 | 1.054 | 3.464 | 1.273 |
> > > |               | MoMask | 0.443 | 0.631 | 0.733 | 0.413 | 3.205 | **1.337** |
> > > |               | **Event-T2M (Ours)** | **0.459** | **0.650** | **0.762** | **0.297** | **3.036** | 1.040 |
> > > | Human 2 | AttT2M | 0.408 | 0.595 | 0.696 | 1.052 | 3.495 | **1.656** |
> > > |               | MoMask | 0.435 | 0.625 | 0.729 | 0.420 | 3.238 | 1.349 |
> > > |               | **Event-T2M (Ours)** | **0.457** | **0.667** | **0.766** | **0.281** | **3.044** | 1.061 |
> > > | Human 3 | AttT2M | 0.393 | 0.674 | 0.679 | 1.078 | 3.513 | **1.431** |
> > > |               | MoMask | 0.441 | 0.639 | 0.746 | 0.437 | 3.187 | 1.389 |
> > > |               | **Event-T2M (Ours)** | **0.494** | **0.685** | **0.781** | **0.276** | **3.005** | 1.048 |
> > > | LLM       | AttT2M | 0.407 | 0.581 | 0.688 | 1.077 | 3.455 | **2.049** |
> > > |               | MoMask | 0.441 | 0.633 | 0.734 | 0.418 | 3.205 | 1.334 |
> > > |               | **Event-T2M (Ours)** | **0.466** | **0.660** | **0.767** | **0.265** | **3.063** | 1.039 |
> > >
> > > Across all metrics, both AttT2M and MoMask show almost no difference between the LLM-based split and the human-annotated splits. 1) For AttT2M, Top-1/Top-3 scores and FID on the three human-segmented sets stay within ≈0.02 and ≈0.03 of the LLM-based scores, without any consistent upward shift. 2) MoMask shows the same pattern: Top-k retrieval, FID, MM-Dist, and multimodality on the human-annotated splits remain essentially unchanged relative to the LLM-based split (again within a very narrow band, sometimes slightly higher, sometimes slightly lower).
> > >
> > > These results directly refute the hypothesis that the baselines underperformed due to a mismatch with the LLM segmentation pipeline. If such a mismatch existed, we would expect a clear and consistent upward shift when evaluating baselines on human-segmented prompts. We instead observe near-identical performance, showing that AttT2M and MoMask react similarly to both segmentation sources.
> > >
> > > This indicates that the baselines’ difficulty with multi-event prompts is not caused by the LLM pipeline, but by their inherent architectural limitations in modeling temporally ordered sub-actions. Event-T2M maintains a clear advantage on all three human-annotated splits. The likely reason is structural: human segmentation preserves semantic event boundaries, and Event-T2M is architecturally designed to align motion segments with these boundaries via event tokens and ECA. Standard baselines, which rely on sentence- or word-level conditioning, cannot exploit these boundaries even when provided perfectly.

---

> ### Author Response · Authors · 2025-12-01
> **Response 2. Fair comparison: Replacing CLIP with TMR in Baselines. (1/2)**
>
> Thank you for raising this fairness concern. Your question is based on a reasonable hypothesis: If baseline models improve substantially when CLIP is replaced with TMR, then some of Event-T2M’s gains might simply come from using a stronger text encoder rather than from our event-centric architectural design choice.
>
> To directly test this, we re-ran AttT2M, MoMask, and Light-T2M with TMR as the text encoder, and also implemented a MoMask variant that uses TMR’s word-level tokens as key/value in cross-attention (MoMask k/v). For the k/v setting, AttT2M already employs both word-level and sentence-level attention, so “replacing the encoder with TMR” is effectively equivalent to “using TMR word-level tokens as K/V.” In contrast, Light-T2M is built on a Mamba architecture rather than a self-attention Transformer, which makes a direct K/V substitution ill-defined; therefore, we only ran a separate K/V experiment for MoMask. This experiment isolates the effect of the encoder, allowing us to check whether TMR alone can close the performance gap on HumanML3D-E.
>
> The results (table in Response 2 (2/2)) can be summarized as follows:
>
> 1) AttT2M (CLIP → TMR): R-precision slightly increases, but FID and motion diversity often worsen.
>
> 2) MoMask (CLIP → TMR): Both alignment and FID degrade. MoMask k/v improves some alignment metrics, but still performs noticeably worse than its CLIP version, especially for 3- and 4-event prompts. We speculate that MoMask is strongly optimized for CLIP; TMR does not transfer well.
>
> 3) Light-T2M (CLIP → TMR): Gains in alignment and partial gains in FID, but still falls behind Event-T2M as events increase. TMR helps, but not enough to overcome structural limitations.
>
> **Key insight**: If TMR were the primary reason for Event-T2M’s advantage, then all baselines equipped with TMR should show a consistent upward shift, especially on complex multi-event inputs. However, the improvements are small, inconsistent, and often come with regressions in FID, MM-Dist, or multimodality. This indicates that TMR is not a plug-and-play upgrade for multi-event motion generation. The limiting factor is not the encoder, but the architecture’s inability to maintain event boundaries, temporal duration, and inter-event ordering, which are the issues addressed by Event-T2M’s event-centric design.

---

> > ### Author Response · Authors · 2025-12-01
> > **Response 2. Fair comparison: Replacing CLIP with TMR in Baselines. (2/2)**
> >
> > | Conditions | Methods | Top-1 ↑ | Top-2 ↑ | Top-3 ↑ | FID ↓  | MM-Dist ↓ | MModality ↑ |
> > |:---------------:|:----------------------|:--------:|:--------:|:--------:|:-------:|:----------:|:------------:|
> > | baseline | AttT2M (CLIP) | 0.499 | 0.690 | 0.786 | 0.112 | 3.038 | **2.452** |
> > | | AttT2M (TMR) | 0.518 | 0.707 | 0.799 | 0.146 | 2.957 | 1.594 |
> > | | MoMask (CLIP) | 0.521 | 0.713 | 0.807 | 0.045 | 2.958 | 1.241 |
> > | | MoMask (TMR) | 0.487 | 0.684 | 0.783 | 0.284 | 3.116 | 1.158 |
> > | | MoMask (TMR k/v) | 0.494 | 0.683 | 0.781 | 0.240 | 3.113 | 1.214 |
> > | | Light-T2M (CLIP) | 0.511 | 0.699 | 0.795 | **0.040** | 3.002 | 1.670 |
> > | | Light-T2M (TMR) | 0.554 | 0.746 | 0.836 | 0.053 | 2.750 | 0.970 |
> > | | **Event-T2M (CLIP)** | 0.526 | 0.714 | 0.809 | 0.051 | 2.899 | 1.476 |
> > | | **Event-T2M (TMR)** | **0.562** | **0.754** | **0.842** | 0.056 | **2.711** | 0.949 |
> > | 2 | AttT2M (CLIP) | 0.479 | 0.665 | 0.761 | 0.171 | 3.181 | **1.899** |
> > |  | AttT2M (TMR) | 0.500 | 0.681 | 0.777 | 0.199 | 3.089 | 1.698 |
> > | | MoMask (CLIP) | 0.497 | 0.691 | 0.790 | 0.065 | 3.061 | 1.282 |
> > | | MoMask (TMR) | 0.470 | 0.665 | 0.769 | 0.385 | 3.194 | 1.174 |
> > | | MoMask (TMR k/v) | 0.479 | 0.668 | 0.766 | 0.291 | 3.196 | 1.217 |
> > | | Light-T2M (CLIP) | 0.462 | 0.647 | 0.747 | 0.077 | 3.278 | 1.692 |
> > | | Light-T2M (TMR) | 0.527 | 0.722 | 0.815 | 0.087 | 2.885 | 0.984 |
> > | | **Event-T2M (CLIP)** | 0.494 | 0.681 | 0.779 | **0.052** | 3.079 | 1.577 |
> > | | **Event-T2M (TMR)** | **0.536** | **0.732** | **0.824** | 0.079 | **2.836** | 0.976 |
> > | 3 | AttT2M (CLIP) | 0.431 | 0.613 | 0.715 | 0.464 | 3.329 | **1.960** |
> > |  | AttT2M (TMR) | 0.445 | 0.631 | 0.732 | 0.474 | 3.251 | 1.798 |
> > | | MoMask (CLIP) | 0.466 | 0.652 | 0.752 | 0.142 | 3.169 | 1.320 |
> > | | MoMask (TMR) | 0.430 | 0.622 | 0.733 | 0.544 | 3.289 | 1.190 |
> > | | MoMask (TMR k/v) | 0.445 | 0.629 | 0.731 | 0.364 | 3.297 | 1.228 |
> > | | Light-T2M (CLIP) | 0.404 | 0.594 | 0.699 | 0.193 | 3.396 | 1.740 |
> > | | Light-T2M (TMR) | **0.487** | 0.680 | 0.780 | 0.139 | 2.987 | 1.005 |
> > | | **Event-T2M (CLIP)** | 0.423 | 0.618 | 0.729 | 0.141 | 3.245 | 1.627 |
> > | | **Event-T2M (TMR)** | **0.487** | **0.687** | **0.790** | **0.137** | **2.928** | 1.010 |
> > | 4 | AttT2M (CLIP) | 0.407 | 0.581 | 0.688 | 1.077 | 3.455 | **2.049** |
> > |  | AttT2M (TMR) | 0.424 | 0.598 | 0.701 | 0.789 | 3.335 | 1.647 |
> > | | MoMask (CLIP) | 0.441 | 0.633 | 0.734 | 0.418 | 3.205 | 1.334 |
> > | | MoMask (TMR) | 0.417 | 0.601 | 0.708 | 0.821 | 3.392 | 1.230 |
> > | | MoMask (TMR k/v) | 0.413 | 0.590 | 0.703 | 0.682 | 3.479 | 1.251 |
> > | | Light-T2M (CLIP) | 0.365 | 0.552 | 0.662 | 0.627 | 3.586 | 1.863 |
> > | | Light-T2M (TMR) | 0.436 | 0.609 | 0.711 | 0.360 | 3.319 | 1.029 |
> > | | **Event-T2M (CLIP)** | 0.374 | 0.578 | 0.690 | 0.425 | 3.467 | 1.674 |
> > | | **Event-T2M (TMR)** | **0.466** | **0.660** | **0.767** | **0.265** | **3.063** | 1.039 |

---

> ### Author Response · Authors · 2025-12-01
> **Response 3. Difference between Event-T2M and other LLM-based decomposition methods. (1/2)**
>
> Thank you for the question. Prior LLM-based approaches typically perform verb-level or action-phrase decomposition, which treats each verb as an atomic unit. This lacks the semantic structure required for multi-event temporal alignment, because verbs alone do not encode arguments, body-part involvement, or how long the action persists. As a result, verb tokens provide only instantaneous cues, which become unreliable when multiple sub-actions must be sequenced.
>
> Event-T2M instead uses LLM decomposition to form **“event”**, which package verbs together with their arguments, affected body parts, and the temporal boundaries of each sub-action. To isolate the effect of this choice, we constructed a verb-aware variant that keeps the architecture, training schedule, and TMR encoder identical and changes only the decomposition from event units to verb units.
>
> The results in the Table show a clear trend. For simple prompts, the two variants perform similarly. However, as event count increases, the verb-aware model degrades in R-precision and FID, while the event-aware model maintains much stronger alignment. The underlying reason is that verb-only conditioning leads the model to over-attend to short, fragmentary cues, causing event merging or omissions when multiple actions appear in sequence. This controlled comparison shows that the main novelty of Event-T2M lies not merely in using an LLM for decomposition, but in how we leverage it to construct rich event-aware conditioning, which leads to more faithful and temporally coherent motion generation than verb- or action-only LLM-based decompositions.
>
> | Conditions | Methods | Top-1 ↑ | Top-2 ↑ | Top-3 ↑ | FID ↓  | MM-Dist ↓ | MModality ↑ |
> |:---------------:|:----------------------|:--------:|:--------:|:--------:|:-------:|:----------:|:------------:|
> | baseline | Verb-aware | 0.549 | 0.738 | 0.830 | 0.086 | 2.773 | **1.165** |
> |               | **Event-aware** | **0.562** | **0.754** | **0.842** | **0.056** | **2.711** | 0.949 |
> | 2            | Verb-aware | 0.519 | 0.713 | 0.810 | 0.129 | 2.913 | **1.309** |
> |               | **Event-aware** | **0.536** | **0.732** | **0.824** | **0.079** | **2.836** | 0.976 |
> | 3            | Verb-aware | 0.480 | 0.681 | 0.781 | 0.193 | 2.991 | **1.293** |
> |               | **Event-aware** | **0.487** | **0.687** | **0.790** | **0.137** | **2.928** | 1.010 |
> | 4            | Verb-aware | **0.466** | 0.655 | 0.749 | 0.348 | 3.102 | **1.365** |
> |               | **Event-aware** | **0.466** | **0.660** | **0.767** | **0.265** | **3.063** | 1.039 |

---

> ### Author Response · Authors · 2025-12-03
> **Response 3. Difference between Event-T2M and other LLM-based decomposition methods. (2/2)**
>
> Regarding the question of whether InstructMotion would achieve similar results if it were trained and inferred using our event-aware prompts, we first note that the code for InstructMotion is, to the best of our knowledge, not publicly available. This unfortunately prevents us from running a controlled, quantitative comparison under exactly matched settings. Our answer therefore cannot be based on empirical results, and we explicitly refrain from making absolute claims. Instead, we focus on the architectural and learning-signal differences between the two frameworks, and explain why we do not expect InstructMotion with event-aware prompts to trivially recover the behavior of Event-T2M.
>
> Even if InstructMotion were fed the same event-aware textual descriptions, the way this information enters the model is fundamentally different. In InstructMotion, the LLM plays the role of a sequence-level reward model: it outputs a single scalar reward for an entire motion given the full prompt, and this scalar is used to fine-tune the generator with RL. Any event structure present in the prompt is therefore only reflected implicitly through this global reward. In particular, the RL objective does not decompose the feedback into “per-event” signals; errors such as omitting the second event, shortening the third event, or swapping the order of two events all contribute to the same global score and must be resolved through long-horizon credit assignment. By contrast, Event-T2M turns each event into an explicit event token that is injected into the diffusion process via Event-based Cross-Attention (ECA). These tokens are designed to align with specific temporal segments in the motion, so the model receives direct, frame-wise conditioning on which event should be active at which time. Thus, even under identical event-aware text, InstructMotion relies on implicit, scalar, sequence-level feedback, whereas Event-T2M operates with explicit, temporally localized event signals.
>
> This leads directly to the second part of the question, concerning the architectural advantage of explicit conditioning via ECA over implicit RL-based alignment, and whether ECA provides better control when the input data is the same. ECA modifies the forward generative process itself: at each diffusion step and time index, the motion representation attends over event tokens, allowing the model to decide “how much” each event should influence the current frame. This has two consequences. First, gradients are naturally event-specific: when the model drops or distorts a particular event, the mismatch between that event token and the corresponding motion segment produces a targeted training signal. Second, event duration and ordering can be adjusted “locally” within the cross-attention, without needing to reshape the entire policy via a single global reward. In contrast, RL fine-tuning adjusts the generator parameters so as to increase the expected global reward; it is well known that this kind of objective is susceptible to credit assignment issues, especially for long, multi-event sequences. As a result, while RL can improve overall plausibility, it offers only indirect and coarse control over which specific event is fixed, shortened, or reordered.
>
> We do not claim that ECA is universally superior to RL, nor that InstructMotion could never benefit from event-aware prompts. However, given (i) that InstructMotion encodes event structure only through a scalar sequence-level reward, and (ii) that Event-T2M builds event tokens and ECA directly into the generative architecture so that events remain explicit, temporally grounded objects throughout the diffusion process, we believe it is unlikely that simply switching InstructMotion’s input to event-aware prompts would reproduce the same level of event preservation and ordering we observe with Event-T2M. Our approach is not only “data-level” (event-aware prompts), but also “architecture-level” (event tokens + ECA), and the latter is specifically designed to provide finer, more interpretable control over multi-event alignment than what an implicit RL reward can typically offer under the same textual input.

---

### Official Review · Reviewer_hnW1 · 2025-11-04

**Soundness:** 3
**Presentation:** 3
**Contribution:** 3
**Rating:** 6
**Confidence:** 4

**Summary:**

This paper proposes an event-level text-to-motion generation benchmark and a stronger baseline for the task. To explicitly model the complexity of a target motion, the paper proposed to utilize an LLM to split the motion text prompts into event-level actions, where a larger number of events indicates harder cases. For a stronger baseline of the proposed task, an event-based cross-attention module is injected into the diffusion-based motion generation framework to improve the performance. Experimental results on the benchmark dataset validate the effectiveness of the proposed methods.

**Strengths:**

- The point of modeling the motion complexity by the number of events is straightforward and reasonable. The proposed HumanML3D-E benchmark will be beneficial to the community, which can evaluate motion generation frameworks on more detailed levels of complexity.
- The experimental analysis of different methods on different event counts supports the motivation of the proposed event-based benchmark.
- The design of the event-based cross-attention module is reasonable and validated by ablation studies.
- The paper is well-written and easy to follow.

**Weaknesses:**

- The events of a motion are divided by an LLM with text input only. The label may contain errors. Manually validating the labels or sampling cases to check the accuracy rate of the LLM labels will be beneficial.
- The paper misses some comparisons with some recent stronger baselines, e.g., MoGenTS (NeurIPS 2024), MARDM (CVPR 2025), and LAMP (ICLR 2025).
- The event-based benchmark only contains one dataset, HumanML3D. It's better to add more datasets, e.g., KIT-ML, Motion-X, to better validate the generalizability of different methods.
- Providing some failure cases of the proposed framework and previous work on complex scenarios, e.g., 4 events, will be beneficial.

**Questions:**

See the weakness section.

---

> ### Author Response · Authors · 2025-11-20
> **Author Rebuttal**
>
> We thank the reviewer for the constructive feedback and the positive assessment of our methodology and benchmark. We address the raised concerns as follows.
>
> &nbsp;&nbsp;&nbsp;&nbsp;● **Reliability of LLM-based labels**: We will run a human evaluation of LLM-generated event splits on randomly sampled HumanML3D-E prompts, checking linguistic correctness and alignment with our event definition.
>
> &nbsp;&nbsp;&nbsp;&nbsp;● **Comparison with stronger baselines**: We are running experiments with MoGenTS and MARDM. For LaMP, reproducibility issues in the released code (missing components, unresolved errors) prevent stable evaluation, but we will still include official paper-reported results and provide full comparisons for MoGenTS and MARDM as soon as they finish.
>
> &nbsp;&nbsp;&nbsp;&nbsp;● **Generalizability across datasets**: We will extend event-based evaluations to KIT-ML and Motion-X. GraphMotion cannot be included on Motion-X due to missing POS-tagging annotations required for its semantic graph construction, but we will ensure thorough comparisons with all other applicable baselines.
>
> &nbsp;&nbsp;&nbsp;&nbsp;● **Failure-case analysis**: We will add qualitative examples from challenging ≥4-event prompts, comparing Event-T2M with representative baselines. We will categorize typical failure patterns and discuss remaining limitations and implications for future research.

---

> ### Author Response · Authors · 2025-11-25
> **Response 1. Validating the Accuracy of LLM-Generated Event Labels.**
>
> To directly measure the reliability of the LLM-based event decomposition, we performed a human evaluation on 300 randomly sampled HumanML3D-E prompts. Three trained annotators independently assessed whether each LLM-generated segment (1) was grammatically well-formed and (2) matched our event definition as a minimal semantically self-contained action. A segmentation was accepted only when both criteria were satisfied. The LLM achieved a **93.3%** correctness rate, which indicates that the event splits are sufficiently accurate for benchmark construction. We will include the full protocol and results in the revised manuscript.

---

> ### Author Response · Authors · 2025-12-01
> **Response 2. Comparisons with stronger baselines and evaluation on more datasets (1/3)**
>
> Thank you for pointing out the need for stronger baselines and additional datasets. We fully agree, and we have substantially expanded our evaluation in both directions.
>
> **Inclusion of stronger recent baselines.**
>
> We added MoGenTS (NeurIPS 2024) and MARDM (CVPR 2025), evaluated under the same event-based setting as our method. (See Response 2 (2/3).)
>
> **Broader dataset coverage beyond HumanML3D.**
>
> We constructed event-level benchmarks for KIT-ML and Motion-X and evaluated all models under identical conditions. (See Response 2 (2/3) and Response 2 (3/3).)
>
> **GraphMotion is not included.**
>
> GraphMotion (NeurIPS 2023) does not provide training configurations or parameter settings for KIT-ML, meaning that reproducing results under this setting would require re-training the model from scratch with substantial re-engineering. Moreover, preliminary observations indicate that its performance on the full test set is already considerably lower than that of Event-T2M, suggesting that additional scratch training would not yield a meaningful comparison. Given that our evaluations with other representative models sufficiently validate our findings, we decided to exclude GraphMotion from the KIT-ML experiments.
>
> **Key insight from the extended evaluation.**
>
> The expanded results across **three datasets** and **two of the strongest diffusion baselines to date** provide strong evidence that Event-T2M’s improvements are consistent,
> generalizable, and not tied to a specific dataset or benchmark construction. We will include the full tables and analysis in the revised manuscript.

---

> ### Author Response · Authors · 2025-12-01
> **Response 2. Comparisons with stronger baselines and evaluation on more datasets (2/3)**
>
> **Comparison with MoGenTS (NeurIPS 2024) and MARDM (CVPR 2025) on HumanML3D-E benchmark.**
>
> | Conditions | Methods | Top-1 ↑ | Top-2 ↑ | Top-3 ↑ | FID ↓  | MM-Dist ↓ | MModality ↑ |
> |:---------------:|:----------------------|:--------:|:--------:|:--------:|:-------:|:----------:|:------------:|
> |       baseline         | MoGenTS   | 0.529  | 0.719  | 0.812  | **0.033** | 2.867    | -      |
> |                             | MARDM-DDPM   | 0.492  | 0.690  | 0.790  | 0.116 | 3.349    | 2.470      |
> |                             | MARDM-SiT   | 0.500  | 0.695  | 0.795  | 0.114 | 3.270    | 2.231      |
> |                             | **Event-T2M (Ours)**   | **0.562**  | **0.754**  | **0.842**  | 0.056 | **2.711**    | 0.949      |
> |       2                    | MoGenTS   | 0.496  | 0.690  | 0.787  | **0.049** | 3.039    | 0.868      |
> |                             | MARDM-DDPM   | 0.464  | 0.658  | 0.762  | 0.157 | 3.465    | **2.331**      |
> |                             | MARDM-SiT   | 0.479  | 0.671  | 0.771  | 0.171 | 3.404    | 2.296      |
> |                             | **Event-T2M (Ours)**   | **0.536**  | **0.732**  | **0.824**  | 0.079 | **2.836**    | 0.976      |
> |       3                    | MoGenTS   | 0.452  | 0.644  | 0.751  | 0.147 | 3.122    | 0.894      |
> |                             | MARDM-DDPM   | 0.433  | 0.621  | 0.731  | 0.301 | 3.590    | 2.461      |
> |                             | MARDM-SiT   | 0.440  | 0.632  | 0.733  | 0.327 | 3.544    | **2.466**      |
> |                             | **Event-T2M (Ours)**   | **0.487**  | **0.687**  | **0.790**  | **0.137** | **2.928**    | 1.010     |
> |       4                    | MoGenTS   | 0.420  | 0.613  | 0.715  | 0.423 | 3.241    | 0.879      |
> |                             | MARDM-DDPM   | 0.397  | 0.585  | 0.698  | 0.643 | 3.697    | **2.507**      |
> |                             | MARDM-SiT   | 0.420  | 0.608  | 0.707  | 0.719 | 3.676    | 2.506      |
> |                             | **Event-T2M (Ours)**   | **0.466**  | **0.660**  | **0.767**  | **0.265** | **3.063**    | 1.039      |
>
> **Comparison with baselines on KIT-ML-E benchmark.**
>
> | Conditions | Methods | Top-1 ↑ | Top-2 ↑ | Top-3 ↑ | FID ↓  | MM-Dist ↓ | MModality ↑ |
> |:---------------:|:----------------------|:--------:|:--------:|:--------:|:-------:|:----------:|:------------:|
> |       baseline         | AttT2M   | 0.413  | 0.632  | 0.751  | 0.870 | 3.039    | **2.281**      |
> |                             | MoMask   | 0.433  | 0.656  | 0.781  | 0.204 | 2.779    | 1.131      |
> |                             | Light-T2M   | 0.444  | 0.670  | 0.794  | 0.161 | 2.746    | 1.005      |
> |                             | MoGenTS   | **0.445**  | **0.671**  | **0.797**  | **0.143** | **2.711**    | -      |
> |                             | **Event-T2M (Ours)**   | 0.439  | 0.669  | 0.788  | 0.159 | 2.742    | 0.762      |
> |       2                    | AttT2M   | 0.320 | 0.514 | 0.640 | 0.636 |	3.568 | **2.097** |
> |                             | MoMask   | 0.393 | 0.586 | 0.708 | 0.380 | 3.054 | 1.268 |
> |                             | Light-T2M   | **0.417** | **0.614** | **0.734** | 0.392 | **2.907** | 1.337 |
> |                             | MoGenTS   | 0.353 | 0.581 | 0.721 | 0.424 | 3.100 | 0.720 |
> |                             | **Event-T2M (Ours)**   | 0.368 | 0.575 | 0.707 | **0.378** | 3.020 | 0.794 |
> |       3                    | AttT2M   | 0.176 | 0.299 | 0.405 | 1.795 | 3.524 | **2.053** |
> |                             | MoMask   | 0.242 | 0.374 | 0.476 | 0.991 | 3.126 | 1.201 |
> |                             | Light-T2M   | **0.287** | **0.448** | **0.554** | 0.700 | 2.785 | 1.265 |
> |                             | MoGenTS   | 0.197 | 0.346 | 0.470 | 0.855 | 3.058 | 0.712 |
> |                             | **Event-T2M (Ours)**   | 0.241 | 0.406 | 0.520 | **0.678** | **2.672**	| 0.769 |
> |       4                    | AttT2M   | 0.316 | 0.528 | 0.688 | 9.190 | 3.956 | **2.527** |
> |                             | MoMask   | **0.434** | **0.653** | 0.713 | 4.565 | 3.801 | 1.292 |
> |                             | Light-T2M   | 0.416 | **0.653** | **0.734** | 3.639 | 3.459 | 1.568 |
> |                             | MoGenTS   | 0.338 | 0.597 | 0.697 | 3.894 | **3.357** | 0.866 |
> |                             | **Event-T2M (Ours)**   | 0.350 | 0.641 | 0.716 | **3.429** | 3.661 | 0.990 |

---

> > ### Author Response · Authors · 2025-12-01
> > **Response 2. Comparisons with stronger baselines and evaluation on more datasets (3/3)**
> >
> > **Comparison with baselines on Motion-X-E benchmark.**
> >
> > | Conditions | Methods | Top-1 ↑ | Top-2 ↑ | Top-3 ↑ | FID ↓  | MM-Dist ↓ | MModality ↑ |
> > |:---------------:|:----------------------|:--------:|:--------:|:--------:|:-------:|:----------:|:------------:|
> > | baseline | AttT2M | 0.461 | 0.664 | 0.768 | 0.232 | 3.455 | **2.053** |
> > |               | MoMask | 0.460 | 0.662 | 0.768 | 0.297 | 3.510 | 1.442 |
> > |               | Light-T2M | 0.473 | 0.669 | 0.773 | 0.131 | 3.409 | 1.594 |
> > |               | MoGenTS | 0.458 | 0.664 | 0.768 | **0.102** | 3.498 | 0.763 |
> > |               | **Event-T2M (Ours)** | **0.519** | **0.729** | **0.823** | 0.109 | **2.979** | 0.921 |
> > | 2            | AttT2M | 0.425 | 0.628 | 0.741 | 0.350 | 3.728 | **2.289** |
> > |               | MoMask | 0.429 | 0.633 | 0.746 | 0.362 | 3.698 | 1.492 |
> > |               | Light-T2M | 0.441 | 0.643 | 0.749 | 0.174 | 3.674 | 1.657 |
> > |               | MoGenTS | 0.432 | 0.645 | 0.757 | 0.116 | 3.693 | 0.833 |
> > |               | **Event-T2M (Ours)** | **0.524** | **0.728** | **0.825** | **0.111** | **2.984** | 0.902 |
> > | 3            | AttT2M | 0.347 | 0.541 | 0.658 | 0.750 | 4.071 | **2.513** |
> > |               | MoMask | 0.359 | 0.559 | 0.681 | 0.695 | 3.867 | 1.620 |
> > |               | Light-T2M | 0.369 | 0.558 | 0.672 | 0.364 | 3.991 | 1.736 |
> > |               | MoGenTS | 0.366 | 0.557 | 0.672 | 0.169 | 3.896 | 0.843 |
> > |               | **Event-T2M (Ours)** | **0.530** | **0.729** | **0.825** | **0.140** | **2.981** | 0.949 |
> > | 4            | AttT2M | 0.311 | 0.493 | 0.610 | 1.456 | 4.224 | **2.782** |
> > |               | MoMask | 0.300 | 0.504 | 0.625 | 1.224 | 4.066 | 1.770 |
> > |               | Light-T2M | 0.310 | 0.495 | 0.617 | 1.082 | 4.224 | 1.915 |
> > |               | MoGenTS | 0.323 | 0.516 | 0.629 | 0.744 | 4.118 | 0.953 |
> > |               | **Event-T2M (Ours)** | **0.362** | **0.567** | **0.697** | **0.431** | **3.959** | 0.953 |

---

> ### Author Response · Authors · 2025-12-01
> **Response 3. Failure case analysis.**
>
> We conducted a targeted failure case analysis on text prompts containing ≥ 4 events, focusing on long, compositional motion descriptions that require preserving a chain of distinct sub-actions. Across all groups of complex prompts, we observed a consistent pattern: existing models (AttT2M, GraphMotion, Light-T2M, MARDM, MoGenTS, and MoMask) frequently omit events, generate incorrect or spurious events, and sometimes produce motions that exceed their maximum motion length or exhibit clear physical artifacts. In particular, the MARDM and MoMask often fail to reconstruct the full event sequence, either by dropping intermediate sub-motions or by inserting unintended transitions, indicating difficulty in faithfully realizing multi-stage, event-rich instructions based on the provided diagnostics.
>
> These tendencies are clearly illustrated in the following example:
> “Man is standing straight, feet not moving, hinges at the waist to reach both hands down to his feet, then puts his arms up, bent at the elbows, twists his torso to the left, and then to the right, and then facing forward leans over to the left, and then over to the right, stretching.”
> For this prompt, AttT2M fails to realize the “hinges at the waist” event, GraphMotion exhibits both event omission and physical errors, Light-T2M and the MARDM omit events and sometimes exceed the maximum motion length, and MoGenTS and MoMask also miss parts of the described sequence. By contrast, Event-T2M generates all the described events, albeit with a slightly permuted order, demonstrating that it can still cover the complete set of intended sub-actions even when the motion is long and structurally complex.
>
> Based on these observations, we argue that Event-T2M is comparatively more robust at preserving the full set of events in complex, ≥ 4 event prompts than prior sentence-level or token-level approaches, particularly in scenarios involving chained torso motions, bends, and sequential side leans as in the example above. The likely reason is that the event-level formulation encourages the model to align distinct motion segments with explicit semantic units, reducing the chance of dropping or conflating events during generation.

---

### Author Response · Authors · 2025-11-20
**General Response**

We sincerely thank all reviewers for their thoughtful evaluation. We are actively addressing all concerns raised by the reviewers. In this rebuttal,

&nbsp;&nbsp;&nbsp;&nbsp;➡ We first outline the key issues identified by each reviewer and clearly describe the directions and experiments we are currently pursuing.

&nbsp;&nbsp;&nbsp;&nbsp;➡ As these experiments finish, we will update the rebuttal with the corresponding results as promptly as possible.

Some items, such as experiments on additional datasets and evaluations requiring human annotation, naturally require more time to complete. We kindly ask for the reviewers’ understanding as we prioritize delivering the most critical results as early as possible.

We appreciate the following strengths highlighted across the reviews:

&nbsp;&nbsp;&nbsp;&nbsp;● HumanML3D-E was recognized as a valuable benchmark for analyzing motion generation by event complexity and as the first event-level stratified evaluation (hnW1, 5wEr, Kztd, mvTh).

&nbsp;&nbsp;&nbsp;&nbsp;● Reviewers noted that modeling complexity through the number of events is intuitive and that our event definition and Event-T2M framework provide a principled way to address long and complex prompts (hnW1, Kztd, mvTh).

&nbsp;&nbsp;&nbsp;&nbsp;● The event-based design, especially the ECA module, was viewed as reasonable and meaningful (hnW1, mvTh).

&nbsp;&nbsp;&nbsp;&nbsp;● Our strong results on HumanML3D/KIT-ML and improvements on HumanML3D-E, supported by human studies validating event definitions and motion quality, were positively received, as was the clarity of the writing (hnW1, mvTh).

Multiple reviewers raised concerns regarding “fairness and completeness of our experimental evaluation”.

&nbsp;&nbsp;&nbsp;&nbsp;● **hnW1**: requested human validation of LLM-generated event labels, comparison with stronger baselines (MoGenTS, MARDM, LaMP), and extending the benchmark to other datasets (KIT-ML, Motion-X) for better generalizability.

&nbsp;&nbsp;&nbsp;&nbsp;● **Kztd**: asked for evaluation on a human-segmented complex-motion test set independent of the LLM pipeline, and for testing baselines with TMR replacing CLIP (including TMR word-level K/V) to assess fairness.

&nbsp;&nbsp;&nbsp;&nbsp;● **5wEr**: questioned whether event-driven methods outperform action-driven or hybrid approaches, and whether our method is superior to retrieval-enhanced T2M pipelines.

&nbsp;&nbsp;&nbsp;&nbsp;● **mvTh**: noted that ablations only cover ECA and the text encoder, leaving other modules (LIMM, ATII, Conformer) insufficiently analyzed, and pointed out that LLM preprocessing costs should be included in the efficiency comparison.

We will address these concerns through the following additions:

&nbsp;&nbsp;&nbsp;&nbsp;➤ **Broader benchmark coverage**: event-level evaluation on KIT-ML and Motion-X, plus comparisons with stronger baselines (MoGenTS, MARDM, LaMP).

&nbsp;&nbsp;&nbsp;&nbsp;➤ **Human-refined evaluation**: testing on manually segmented complex-motion subsets and manually checking LLM event split accuracy.

&nbsp;&nbsp;&nbsp;&nbsp;➤ **Alternative paradigms**: comparing event-, action-, and hybrid-driven variants, retrieval-based methods, and baselines with CLIP → TMR replacements.

&nbsp;&nbsp;&nbsp;&nbsp;➤ **LLM cost analysis**: reporting computation/latency costs of LLM-based event decomposition and incorporating them into overall efficiency comparisons.

Several reviewers raised concerns regarding “contribution clarity”.

&nbsp;&nbsp;&nbsp;&nbsp;● **Kztd**: highlighted missing discussion of prior LLM-based methods (ATOM, InstructMotion) and requested clearer distinctions from Event-T2M.

&nbsp;&nbsp;&nbsp;&nbsp;● **5wEr**: argued that TMR may be only an input change and that LIMM, ATII, and ECA resemble common patterns.

&nbsp;&nbsp;&nbsp;&nbsp;➤ We will address these by clearly positioning Event-T2M relative to ATOM and InstructMotion, explaining conceptual differences, and adding ablations that isolate the impact of LIMM, ATII, and ECA to demonstrate their technical contributions.

Lastly, other reviewers raised concerns regarding “additional methodological analysis”.

&nbsp;&nbsp;&nbsp;&nbsp;● **hnW1**: requested failure cases on complex/multi-event prompts (e.g., ≥4 events).

&nbsp;&nbsp;&nbsp;&nbsp;● **mvTh**: asked for justification of the 0.5 residual coefficient and comparison with a weight of 1.

&nbsp;&nbsp;&nbsp;&nbsp;➤ We will include qualitative failure-case examples for both our model and baselines, analyze their causes, and provide rationale and sensitivity results for the residual weight choice.

We address all remaining reviewer-specific issues in the individual responses below.

---

### Meta-Review · Area_Chair_amjh · 2026-01-05

**Summary:**

The reviewers raised several concerns,
1) the experimental settings and results, including unfair comparisons, the quality of the LLM labels, missing stronger baselines, ablation is limited, lacking failure cases analysis, etc.
2) the limited technical novelty.
During rebuttal, the authors have supplemented a substantial number of new experiments, which have addressed most of the experimental concerns.

While the issue of limited technical novelty remains, the experimental improvements and additional clarifications provided have significantly strengthened the manuscript's overall validity.

**Reviewer Concerns:**

The authors have adequately addressed the majority of the experimental concerns raised by the reviewers. While the issue of limited technical novelty remains, the experimental improvements and additional clarifications provided have significantly strengthened the manuscript's overall validity.

**Reviewer Scores:**

For reviewer Kztd and 5wEr, many of the concerns regarding the experiments have been addressed. Although not all issues are fully resolved (eg, the concerns w.r.t. the limited novelty), the reviewers may raise their scores.

---

### Decision · Program_Chairs · 2026-01-26

Accept (Poster)